# Affinity gaps among B cells in germinal centers drive the selection of MPER precursors

Rashmi Ray[1,12], Torben Schiffner[2,3,4,5,12], Xuesong Wang[1,12], Yu Yan[1,12], Kimmo Rantalainen [2,3,4], Chang-Chun David Lee[3,4,6], Shivang Parikh [1], Raphael A. Reyes [1], Gordon A. Dale[1], Ying-Cing Lin [1], Simone Pecetta [1,11], Sophie Giguere [1], Olivia Swanson [2,3,4], Sven Kratochvil [1], Eleonora Melzi [1], Ivy Phung [4,7,8], Lisa Madungwe[1], Oleksandr Kalyuzhniy [2,3,4], John Warner[1], Stephanie R. Weldon[1], Ryan Tingle[2,3,4], Edward Lamperti[1], Kathrin H. Kirsch[1], Nicole Phelps[2,3,4], Erik Georgeson [2,3,4], Yumiko Adachi[2,3,4], Michael Kubitz[2,3,4], Usha Nair[1], Shane Crotty [4,7,8], Ian A. Wilson [3,4,6], William R. Schief [1,2,3,4,9] ✉ & Facundo D. Batista [1,10] ✉

Current prophylactic human immunodeficiency virus 1 (HIV-1) vaccine research aims to elicit broadly neutralizing antibodies (bnAbs). Membrane-proximal external region (MPER)-targeting bnAbs, such as 10E8, provide exceptionally broad neutralization, but some are autoreactive. Here, we generated humanized B cell antigen receptor knock-in mouse models to test whether a series of germline-targeting immunogens could drive MPER-specific precursors toward bnAbs. We found that recruitment of 10E8 precursors to germinal centers (GCs) required a minimum affinity for germline-targeting immunogens, but the GC residency of MPER precursors was brief due to displacement by higher-affinity endogenous B cell competitors. Higher-affinity germline-targeting immunogens extended the GC residency of MPER precursors, but robust long-term GC residency and maturation were only observed for MPER-HuGL18, an MPER precursor clonotype able to close the affinity gap with endogenous B cell competitors in the GC. Thus, germline-targeting immunogens could induce MPER-targeting antibodies, and B cell residency in the GC may be regulated by a precursor–competitor affinity gap.

Broadly neutralizing antibodies (bnAbs) targeting conserved sites on the envelope (Env) protein of human immunodeficiency virus (HIV), which is composed of the gp120 and gp41 subunits, provide templates for antibody-based HIV vaccine development[1–6]. The route to a mature HIV bnAb from B cell antigen receptor (BCR) modification after antigen activation through somatic hypermutation (SHM) and cycles of expansion and selection in the germinal center (GC)[7] is difficult due to high rates of SHM in mature bnAbs, the rarity of bnAb precursors,

long heavy chain complementarity determining region 3s (HCDR3s) and, in some cases, poly- or autoreactivity[1,4,5,8–11].

Known HIV bnAbs target several sites on the trimeric HIV Env glycoprotein, including the membrane-proximal external region (MPER) in gp41, located near the viral membrane[4,5]. MPER-class bnAbs target overlapping epitopes on the C-terminal MPER helix and have high neutralization breadth[3,6,11–14]. MPER bnAbs contain long and hydrophobic HCDR3s, which, in the case of 2F5 and 4E10, leads to binding to human

autoantigens and/or polyreactivity and to their blockade by immune checkpoints in bnAb knock-in mouse models[15–18]. The bnAb 10E8, which binds to the distal MPER, has less poly- and autoreactivity[2,12,19] and appreciable neutralization breadth[12,20]. An unmutated common ancestor (UCA) of 10E8 proposed based on next-generation sequencing (NGS) data combined with structure–function studies lacks affinity for MPER and does not neutralize even the most sensitive HIV-1 isolates[21]. To gain recognition of Env in its membrane context, extensive SHM, particularly in HCDR2 and HCDR3, would be required[21,22]. One strategy to overcome the lack of affinity of 10E8-like precursor antibodies for native Env, referred to as germline targeting (GT), relies on priming with immunogens designed for increased affinity to diverse precursors, followed by sequential boosting with increasingly native-like immunogens to shepherd SHM toward the mature bnAb sequence. The hydrophobic MPER is excluded from most soluble Env immunogens to prevent aggregation[14,23,24]. A series of scaffolds displaying engineered MPER epitopes, named 10E8-GT and displayed on multivalent nanoparticles for improved immunogenicity, can bind 10E8-like precursors in vitro[25].

Here, we generated two mouse models expressing the heavy chain sequence 10E8-UCA[21] or 10E8-NGS-04 (a 10E8-like heavy chain identified by sequencing naive B cells from healthy human donors)[25,26] in functional BCRs with mouse light chains. We also used a pre-existing line expressing a heavy chain sequenced from genuine human B cell precursors binding to 10E8-GT9 immunogens, referred to as MPER-HuGL18 ref. [25]. We found that B cells in the knock-in mice developed normally and underwent affinity maturation in GCs after immunization with 10E8-GT immunogens[25] and identified a dynamic affinity gap between precursor B cells and endogenous competitors driving GC kinetics. Furthermore, variations among B cell lineages in BCR affinity maturation capacity were of substantial importance for maintenance in GCs.

## Results

### Humanized 10E8-like BCRs are functional in mice

To address whether 10E8-like BCRs had normal physiology, expression and B cell distribution in mice, we inserted the prerearranged heavy chain sequences of 10E8-UCA (encoded by $V_H3\text{-}15^*0$, $D_H3\text{-}3^*01$ and $J_H1^*01$)[21] and 10E8-NGS-04 (encoded by $V_H3\text{-}15^*02$, $D_H3\text{-}3^*01$ and $J_H6^*02$)[25] into the *IgH* locus of C57BL/6 mice[26,27] to generate mice that were $IgH^{10E8\text{-}UCA/WT}IgK^{WT/WT}$ (hereafter 10E8-UCA[H]) and $IgH^{10E8\text{-}NGS\text{-}04/WT}$ $IgK^{WT/WT}$ (hereafter 10E8-NGS-04[H]). Proportions of bone marrow $B220^+CD43^+$ B cells (hereafter early B cells) and $B220^+CD43^-$ B cells[28] (hereafter late B cells) were similar in 8- to 10-week-old 10E8-UCA[H], 10E8-NGS-04[H] and wild-type C57BL/6 mice (Extended Data Fig. 1a,b), as were the proportions of early B cell fractions (BP-1[−]CD24[−] B cells (fraction A) and BP-1[−]CD24[+] B cells (fraction B)) and late B cell fractions (IgD[−]IgM[−] B cells (fraction D), IgD[−]IgM[−/lo] B cells (fraction E) and IgD[+]IgM[−/+] B cells (fraction F); Extended Data Fig. 1a,b). The frequency of BP-1[+]CD24[+] B cells (early fraction C) was diminished in 10E8-NGS-04[H] and 10E8-UCA[H] mice relative to in wild-type mice, but as fractions D−F were not diminished (Extended Data Fig. 1a,b), this was likely due to rapid checkpoint passage. Spleen $B220^+TCR\beta^-$ B cells (hereafter B cells) and $TCR\beta^+B220^-$ T cells were comparable in 10E8-UCA[H], 10E8-NGS-04[H] and wild-type mice (Fig. 1a,b), as were CD21[hi]CD24[lo] mature follicular B cells and CD21[hi]CD24[hi]CD23[+] T2 B cells, whereas CD21[hi]CD24[hi]CD23[−] marginal zone B cells and CD21[lo]CD24[hi] T1 B cells were slightly reduced in 10E8-UCA[H] and 10E8-NGS-04[H] mice compared to in wild-type mice (Fig. 1a,b and Extended Data Fig. 1c,d). Expression of B220, IgM and IgD in follicular B cells was similar in 10E8-UCA[H], 10E8-NGS-04[H] and wild-type mice (Extended Data Fig. 1e). B220[+]CD93[+]IgM[−/lo] T3 populations were similar in wild-type and 10E8-UCA[H] mice but were somewhat lower in 10E8-NGS-04[H] mice (Extended Data Fig. 2a–c), indicating that B cells in 10E8-UCA[H] and 10E8-NGS-04[H] mice were not anergic.

To quantify cell membrane expression of 10E8-NGS-04 and 10E8-UCA heavy chains, we sequenced BCRs of single-cell-sorted $B220^+IgM^+Live/CD4^-CD8^-Gr1^-F4/80^-$ B cells (hereafter $B220^+IgM^+$ B cells) from the blood of 10E8-UCA[H] and 10E8-NGS-04[H] mice (Extended Data Fig. 2d). The *Igh* sequences in ~94% ($n = 1,046$) and ~65% ($n = 1,260$) of $B220^+IgM^+$ B cells were identical to the original 10E8-NGS-04 and 10E8-UCA heavy chain sequences, respectively, and were associated with diverse mouse Igκ (Extended Data Fig. 2e,f), although ~80% of 10E8-NGS-04 knock-in heavy chains paired to a small set of mouse light chains (IGKV1-135, IGLV1, IGKV10-96 and IGKV1-117; Extended Data Fig. 2e).

The epitope scaffold 10E8-GT9.2 bears an MPER epitope with additional GT mutations[25] (Extended Data Fig. 2g) and binds human 10E8-UCA[H]/UCA[L] with 920 nM affinity and human 10E8-NGS-04 (heavy + light chains) with 3.5 μM affinity[25]. More than 20% of blood $B220^+CD4^-CD8^-F4/80^-Gr\text{-}1^-$ naive B cells (hereafter naive B cells) in 10E8-NGS-04[H] mice bound biotinylated 10E8-GT9.2 probes, whereas the fraction of naive B cells from wild-type mice that bound was negligible (Fig. 1c), and ~50% of blood naive B cells from 10E8-UCA[H] mice bound biotinylated 10E8-GT9.2 probes (Fig. 1d). Single-cell BCR sequencing (scBCR-seq) of sorted 10E8-GT10.2[+] blood naive B cells from 10E8-NGS-04[H] mice indicated that 100% of $B220^+10E8\text{-}GT10.2^+$ B cells ($n = 1,026$) expressed the 10E8-NGS-04 heavy chain paired with various mouse Igκ molecules (Fig. 1e); similarly, 100% of $B220^+10E8\text{-}GT10.2^+$ B cells ($n = 1,418$) expressed the 10E8-UCA heavy chain paired with various mouse light chains (Fig. 1e). Thus, knocked-in 10E8-UCA and 10E8-NGS-04 heavy chains did not inhibit B cell development, crossed immune checkpoints, were well tolerated, underwent allelic exclusion and were found in functional BCRs.

### GT immunogens prime 10E8-UCA[H]/10E8-NGS-04[H] B cells in vivo

We next tested whether 10E8-UCA[H] B cells could be primed with 10E8-GT9.2 immunogens. Human 10E8-UCA had higher affinity for 10E8-GT9.2 than 10E8-NGS-04 (Extended Data Fig. 3a). To test the affinity of 10E8-UCA[H] B cells, mouse light chains associated with 10E8-UCA[H]-derived 10E8-GT9.2[+] BCRs were expressed, purified and paired with 10E8-UCA Fab heavy chains. The 10E8-UCA[H]/mouse[L] Fabs generated this way bound to 10E8-GT9.2 12mer[25] (Extended Data Fig. 2g) with a median affinity of ~3 μM, which was approximately fourfold lower than the affinity for the human 10E8-UCA[H]/UCA[L] pair (~0.7 μM; Fig. 2a). To create a system with physiologically relevant low frequencies of precursor B cells, we adoptively transferred 50,000 $B220^+CD45.2^+$ B cells from CD45.2 10E8-UCA[H] mice into congenic CD45.1 wild-type mice, which lead to $4{:}10^5$ splenic 10E8-UCA[H] B cells (Extended Data Fig. 3b), slightly higher than the estimated frequency of ~$1.5{:}10^5$ precursor B cells that express 10E8-like heavy chains in humans[25]. Recipient mice were immunized with 50 μg of 10E8-GT9.2 12mer in Ribi adjuvant or 10E8-GT9-KO 12mer, which expresses mutated (672A, 673R, 675R, 680E and 683D; Hxb2 numbering) epitopes that do not bind to 10E8-like precursors[25]. $B220^+CD95^{hi}CD38^{lo}$ GC B cells (hereafter GC B cells) were detected in the spleens of immunized mice on day 7 post-immunization (p.i.; Fig. 2b). 10E8-GT9-KO 12mer did not recruit 10E8-UCA[H] CD45.2[+] B cells to GCs, whereas 10E8-GT9.2 12mer recruited a minor fraction (0.23%) of 10E8-UCA[H] CD45.2[+] B cells to GCs (Fig. 2b), indicating that 10E8-UCA[H] B cells were specifically activated and recruited to GCs by 10E8-GT9.2 12mer.

The frequency of precursor B cells for modest- and low-affinity immunogens is reported to affect B cell recruitment to GCs[29]. To investigate whether precursor frequency influenced 10E8-UCA[H] B cell recruitment to GCs, we adoptively transferred 10E8-UCA[H] CD45.2[+] B cells into CD45.1 wild-type mice to establish 10E8-UCA[H] B cell frequencies of $2{:}10^6$, $8{:}10^6$, $4{:}10^5$ and $4{:}10^4$ (Extended Data Fig. 3b) 1 day before immunization with 10E8-GT9.2 12mer or 10E8-GT9-KO 12mer. On day 7 p.i., some 10E8-UCA[H] CD45.2[+] GC B cells were detected in mice with initial $4{:}10^5$ and $4{:}10^4$ 10E8-UCA[H] B cell frequencies, but not mice with $2{:}10^6$ and $8{:}10^6$ initial frequencies (Extended Data Fig. 3c,d). In the 10E8-GT9.2 12mer-immunized cohort, we also detected a high proportion (~20%)

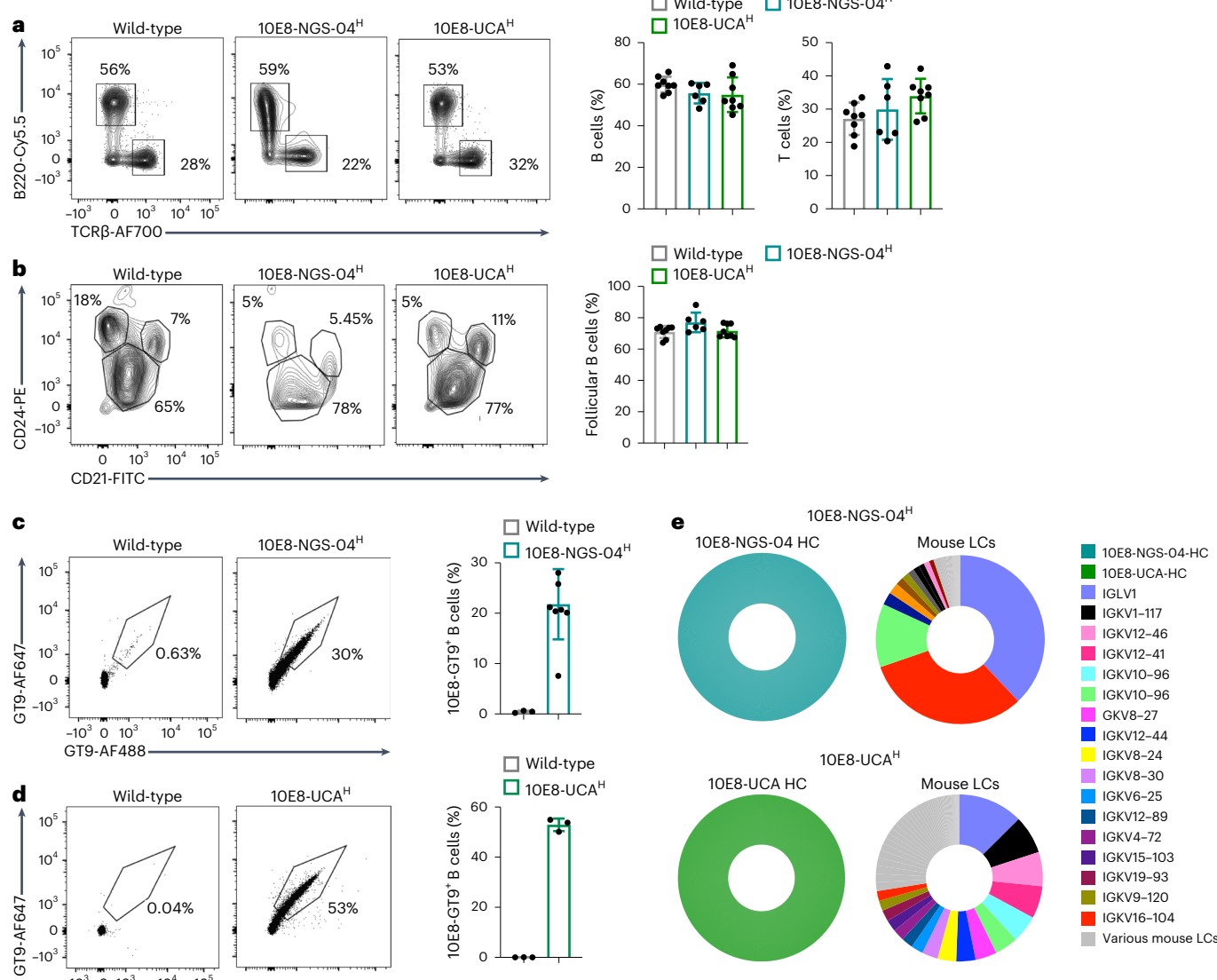

**Fig. 1 | 10E8 knock-in mice express heavy chains from human 10E8 precursor B cells. a**, Representative flow cytometry plots (left) and quantification (right) of spleen cells from 8- to 10-week-old wild-type, 10E8-NGS-04^H and 10E8-UCA^H mice and gating strategy for the quantification of splenic B220+TCRβ− B cells and TCRβ+B220− T cells. Data are pooled from two independent experiments (*n* = 3–4 mice per independent group). **b**, Representative flow cytometry plots (left) and quantification (right) of spleen cells from 8- to 10-week-old wild-type, 10E8-NGS-04^H and 10E8-UCA^H mice and gating strategy for the quantification of CD21^hiCD24^lo follicular B cells, transitional CD21^loCD24^hi T0/T1 B cells, CD21^hiCD24^hiCD23+ T2 B cells and CD21^hiCD24^hiCD23− marginal zone B cells. Data are pooled from two independent experiments (*n* = 3–4 mice per independent group). Error bars indicate mean ± s.d. from mice in pooled groups. **c**, Representative flow cytometry (left) and quantification (right) of B220+CD4−CD8−F4/80−Gr-1− naive B cells from peripheral blood binding biotinylated 10E8-GT9 in wild-type or 10E8-NGS-04^H mice. Data are pooled from two independent

experiments. Error bars indicate mean ± s.d. from mice in each group (*n* = 3–6). **d**, Representative flow cytometry (left) and quantification (right) of naive B cells binding to biotinylated 10E8-GT9 in wild-type or 10E8-UCA^H mice. Error bars indicate mean ± s.d. from mice in each group (*n* = 3). Data from a single experiment are presented and are representative of three independent experiments. **e**, Top, 10x Genomics scBCR-seq data from 10E8-GT10.2-specific naive B cells from 10E8-NGS-04^H mice (*n* = 1,026 pairs amplified) showing human 10E8-NGS-04 *IGH-V* gene frequency (left) and mouse *IGK-V* genes (right). Cells from three mice were sequenced, and representative data from one mouse are presented. Bottom, 10x Genomics scBCR-seq data from 10E8-GT10.2-specific naive B cells from 10E8-UCA^H mice (*n* = 1,418 pairs amplified) showing the frequency of the human 10E8-UCA *IGH-V* gene (left) and mouse *IGK-V* genes (right). Cells from three mice were sequenced, and representative data from one mouse are presented; HC, heavy chain; LC, light chain.

of host wild-type CD45.1+ GC B cells that bound to 10E8-GT9 (Fig. 2b,c), indicating that they may compete with 10E8-UCA^H CD45.2+ GC B cells.

To address the poor recruitment of 10E8-UCA^H B cells to GCs, we added a PADRE epitope, which increases immunogenicity by enhancing CD4+ T cell help, to the linker between the T298 scaffold and nanoparticle (10E8-GT9.3 12mer; Extended Data Fig. 4a). In parallel, directed evolution to select for mutations to reduce competitor binding recovered two predominant mutations, N101I and T95A. Constructs that

contained T95A, whether alone or in combination with other recovered mutations, failed to express as nanoparticles, whereas the addition of N101I resulted in correctly assembled particles (10E8-GT9.4 12mer; Extended Data Fig. 4a). We combined the PADRE epitope and the N101I mutation to generate 10E8-GT9.5 12mer (Extended Data Fig. 4a). A second round of directed evolution on 10E8-GT9.5 enriched for mutations E31A and E55D in the scaffold (10E8-GT9.6 12mer; Fig. 2d and Extended Data Fig. 4b,c). The affinity of 10E8-GT9.3, 10E8-GT9.4, 10E8-GT9.5 and

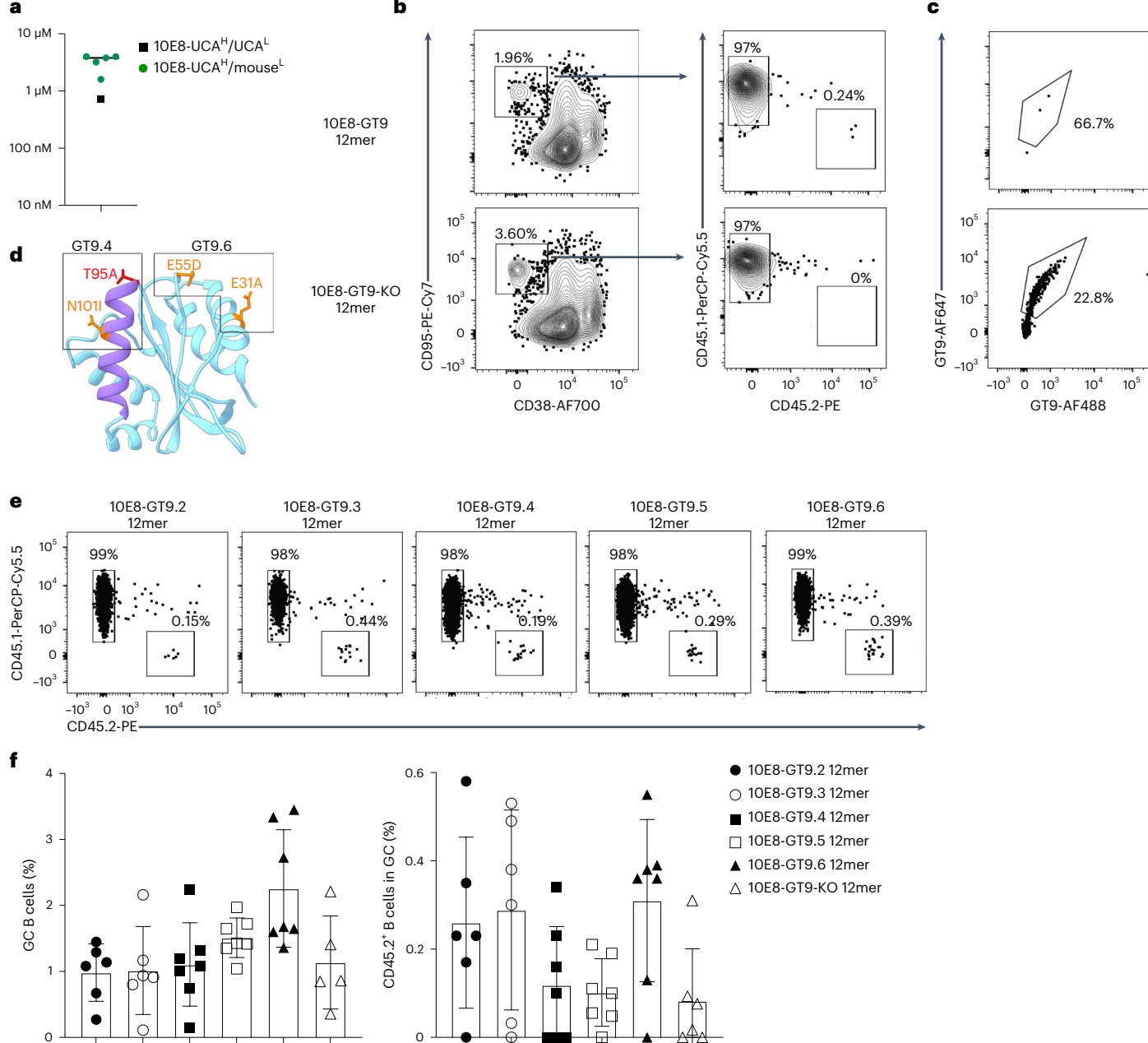

**Fig. 2 | 10E8-UCA[H] B cells are primed and recruited to GCs. a,** Affinities of human 10E8-UCA[H]/UCA[L] or 10E8-UCA[H]/mouse[L] Fabs to 10E8-GT9.2 12mer in a nanoparticle SPR assay. Symbols represent individual antibodies, and the line denotes the median 10E8-UCA[H]/mouse[L] value. **b,** Representative flow cytometry plots of CD95[hi]CD38[lo]B220[+] GC B cells and their distribution into CD45.2[+] and CD45.1[+] B cells at day 7 p.i. in CD45.1 wild-type mice transferred with 50,000 CD45.2[+] 10E8-UCA[H] B cells (to reach a frequency of 4:10[5] CD45.2[+] 10E8-UCA[H] B cells) 1 day before immunization with 50 µg of 10E8-GT9 12mer or 10E8-GT9-KO and Ribi adjuvant. **c,** Representative flow cytometry plots showing the percentage of CD45.2[+]10E8-GT9[++] (top) CD45.1[+]10E8-GT9[++] (bottom) GC B cells at day 7 p.i. with 50 µg of 10E8-GT9.2 12mer in recipients as in **b. d,** Ribbon diagrams showing the modifications in immunogens 10E8-GT9.4 and

10E8-GT9.6 derived through the reduction of off-target effects (10E8-GT9.4) plus the addition of the PADRE T cell help epitope (10E8-GT9.6). Mutation T95A (red) was removed, as candidates containing this mutation failed to express as 12mers. **e,f,** Representative flow cytometry plots of CD45.2[+] GC B cells and CD45.1[+] GC B cells (**e**) and quantification of GC B cells and CD45.2[+] GC B cells (**f**) at day 7 p.i. in CD45.1 wild-type mice transferred with CD45.2[+] 10E8-UCA[H] B cells (frequency of 6:10[5] CD45.2[+] 10E8-UCA[H] B cells) and immunized 1 day later with 50 µg of 10E8-GT9.2 12mer, 10E8-GT9.3 12mer, 10E8-GT9.4 12mer, 10E8-GT9.5 12mer, 10E8-GT9.6 12mer or 10E8-GT9-KO 12mer with Ribi adjuvant. Error bars indicate mean ± s.d. from mice in each group (n = 6). Data from one representative experiment are presented, and two independent experiments were performed.

10E8-GT9.6 variants to human 10E8-UCA[H]/UCA[L] was between 280 nM and 1 µM; the affinities of this set of immunogens to 10E8-UCA[H]/mouse[L] antibodies were all within a narrow range (median: 3.2–4.4 µM; Extended Data Fig. 4d). To determine the effects of these modifications on 10E8-UCA[H] B cell recruitment to the GC, we intravenously transferred 10E8-UCA[H]CD45.2[+] cells into CD45.1 wild-type mice to establish

a frequency of 6:10[5] 10E8-UCA[H] B cells and immunized with 10E8-GT9.2, 10E8-GT9.3, 10E8-GT9.4, 10E8-GT9.5, 10E8-GT9.6 or 10E8-GT9-KO 12mer 1 day after transfer. Mice immunized with 10E8-GT9.6 12mer showed a twofold increase in GC B cells (mean: 2.3%) on day 7 p.i. relative to those immunized with any other nonknockout variant (means: ~0.9–1.5% GC B cells; Fig. 2f). However, all nonknockout variants

induced means of ~0.1–0.3 CD45.2[+] GC B cells (Fig. 2e,f), suggesting that none of them enhanced CD45.2[+] B cell recruitment compared to 10E8-GT9.2 12mer. Thus, although 10E8-UCA[H] B cells were activated and recruited to GCs in response to 10E8-GT9 12mers, recruitment was not robust and was lower than that achieved in other HIV bnAb precursor models using immunogens of similar affinities[26,29–32].

## Immunogen formulation enhances 10E8-specific responses

We sought to overcome activation barriers using increased immunogen affinity for precursor BCRs, multimerization to enhance avidity and GC trafficking[33,34] and alternate dosage and adjuvants. To determine whether increasing antigen affinity improved 10E8-UCA[H] B cell recruitment in GCs, we used 10E8-GT10.2 12mer, which includes additional GT mutations along with additional mutations to reduce off-target binding and additional glycans to shield the 10E8-GT scaffold from generating competitive or off-target immune responses[25]. Relative to 10E8-GT9.2, 10E8-GT10.2 12mer exhibited improved affinity (~100 nM) to the human 10E8-UCA[H]/UCA[L] and higher affinities for 10E8-UCA[H]/mouse[L] antibodies[25] (median: ~1.3 µM; Fig. 3a). To confirm that 10E8-GT10.2 could bind diverse 10E8-class precursor heavy chains, we incubated peripheral blood mononuclear cells from 10E8-UCA[H] and 10E8-NGS-04[H] mice with 10E8-GT10.2 12mer. The dissociation rate of 10E8-NGS-04[H] B cells from the 10E8-GT10.2 probe was more rapid than 10E8-UCA[H] B cells, indicating present, but weaker, binding by 10E8-NGS-04[H] B cells to 10E8-GT10.2 12mer (Extended Data Fig. 5a). To examine immunogenicity in vivo, CD45.2[+]10E8-UCA[H] B220[+] B cells were transferred into CD45.1 wild-type mice to achieve a frequency of 1:10[4] precursor B cells 1 day before immunization with 10E8-GT10.2 12mer, 10E8-GT9.6 12mer or a 10E8-GT9-KO 12mer control (Extended Data Fig. 5b). Both 10E8-GT9.6 12mer and 10E8-GT10.2 12mer induced comparable GC formation in the spleen (Fig. 3c), whereas 10E8-GT10.2 12mer induced a greater than twofold increase in the recruitment of 10E8-UCA[H] CD45.2[+] GC B cells (mean: 0.85%) compared to 10E8-GT9.6 12mer (mean: 0.43%; Fig. 3b,c), which was low relative to the ranges achieved in other bnAb elicitation models[26,29–32].

To determine the effect of 10E8-GT multimerization, CD45.2 10E8-UCA[H] B cells were adoptively transferred into wild-type mice (frequency of ~1:10[4]) 1 day before immunization with 10E8-GT10.2 60mer (Extended Data Fig. 4a) (25 µg, 6.25 µg or 1.25 µg), 10E8-GT10.2 12mer (5 µg) or the 10E8-GT9-KO 12mer control (5 µg). On day 7 p.i., 10E8-GT10.2 60mer at any dose did not improve CD45.2[+] GC B cell recruitment (means of 0.98% for 25 µg, 0.26% for 2.5 µg and 0.29% for 1.25 µg) relative to 10E8-GT10.2 12mer (1.38%; Extended Data Fig. 6a,b). However, serum-binding analyses indicated that 10E8-GT10.2 60mer induced stronger total IgG responses than 10E8-GT10.2 12mer (Extended Data Fig. 6c).

To examine the effect of antigen formulations, CD45.1 wild-type mice transferred with CD45.2 10E8-UCA[H] B cells at a frequency of ~1:10[4] were immunized with different doses of 10E8-GT10.2 12mers in Ribi (40 µg, 10 µg and 2.5 µg) or alhydrogel (20 µg, 5 µg and 1.25 µg). On day 7 p.i. all Ribi-adjuvanted 10E8-GT10.2 12mer doses induced strong GC formation (Extended Data Fig. 6d), and recruitment of 10E8-UCA[H] CD45.2[+] B cells to splenic GCs was dose independent (2.78% for 40 µg, 0.97% for 10 µg and 2.26% for 2.5 µg; Fig. 3d and Extended Data Fig. 6d). Alhydrogel-adjuvanted 10E8-GT10.2 12mers recruited more 10E8-UCA[H] CD45.2[+] B cells to GCs than Ribi-adjuvanted 10E8-GT10.2 12mers, with the highest mean responses (18.86% in spleen and 33% in lymph nodes) induced by 1.25 µg of 10E8-GT10.2 12mer (Fig. 3d and Extended Data Fig. 6e,f). Most GCs in the spleens of 10E8-GT10.2 12mer + alhydrogel-immunized mice contained CD45.2[+]GL-7[+]B220[+] B cells (Fig. 3e). CD45.1 wild-type mice adoptively transferred with CD45.2 10E8-UCA[H] B cells (frequency of 1:10[4]) before immunization with 5 µg of 10E8-GT9 (9.2, 9.3, 9.4 and 9.6) 12mers formulated with alhydrogel (Extended Data Fig. 7a), but not when immunized with 10E8-GT9-KO 12mer with alhydrogel, showed strong recruitment of

CD45.2[+] GC B cells in spleens and lymph nodes at day 7 p.i. (Extended Data Fig. 7b,c). To determine the immunogenicity of the alhydrogel formulations with diverse 10E8 heavy chains, we adoptively transferred CD45.1 wild-type mice with CD45.2 10E8-NGS-04[H] B cells to establish a frequency of 1:10[4] and immunized 1 day later with 5 µg of 10E8-GT10.2 12mer or 10E8-GT9-KO 12mer with alhydrogel (Extended Data Fig. 5c). The recruitment of 10E8-NGS-04[H] CD45.2[+] GC B cells was low but present at day 7 p.i. (mean: 0.08%) but was almost absent at day 14 (mean: 0.03%; Extended Data Fig. 5d,e). Thus, multimerization of 10E8-GT immunogens was not effective in the recruitment of 10E8 precursors to the GCs, in contrast to other engineered multimeric immunogens (eOD-GT8 60mer) that drive the activation of rare precursors[30,32,35–39], while increased affinity somewhat, and alhydrogel formulation substantially, improved the GC recruitment of 10E8-UCA[H] B cells, indicating the importance of class-specific approaches to bnAb elicitation.

## 10E8-GT10.2 does not sustain 10E8-UCA[H] B cell responses

To characterize responses and affinity maturation over time, we investigated the GC kinetics of 10E8-UCA[H] CD45.2[+] B cells transferred into CD45.1 wild-type mice after immunization with 5 µg of 10E8-GT10.2 12mer or 10E8-GT9-KO 12mer with alhydrogel up to day 21 p.i. (Extended Data Fig. 8a). Immune responses to 10E8-GT10.2 12mer in spleen showed similar GC B cells at day 7 (mean: 1.7%) and day 21 (mean: 1.7%) p.i. (Extended Data Fig. 8b), but CD45.2[+] GC B cells, while abundant at day 7 (mean: 8.91%), were almost undetectable at day 14 (mean: ~0.08%) and day 21 (mean: 0.09%) p.i. (Fig. 4a,b); epitope-specific 10E8-UCA[H] GT10[++]KO[−]CD45.2[+] GC B cells, which bind the GT10 probe but not the GT10.2-KO probe, were detected at days 7 and 14, but not at day 21 (Extended Data Fig. 8c). Total serum IgG increased from day 7 to day 21 p.i. but was not generally epitope specific (Extended Data Fig. 8d).

A substantial number of wild-type CD45.1[+] GC B cells bound to the 10E8-GT10.2 probe at day 7 (14.1%), day 14 (~16.8%) and day 21 (~19.1%) p.i. (Fig. 4b), indicating that host CD45.1[+] B cells competed for antigen. We therefore performed scBCR-seq on epitope-specific wild-type CD45.1[+]GT10[++]KO[−] GC B cells at days 7, 14 and 21 p.i. and 10E8-UCA[H] GT10[++]KO[−] GC B cells on days 7 and 14 p.i. Compared to pretransfer (day −1, naive) 10E8-UCA[H] GT10[++]KO[−] B cell and wild-type GT10[++]KO[−] B cell IGHV sequences, IGHV sequences in 10E8-UCA[H] GT10[++]KO[−] GC B cells had acquired a median of 3.5 amino acid mutations, similar to IGHV sequences in wild-type CD45.1[+]GT10[++]KO[−] GC B cells (median of 3 amino acid mutations) on day 7 p.i. (Fig. 4c). The remaining 10E8-UCA[H] GT10[++]KO[−] GC B cells had acquired a median of five amino acid mutations, less than wild-type CD45.1[+]GT10[++]KO[−] GC B cells (median of seven amino acid mutations) on day 14 p.i. (Fig. 4c), indicating that 10E8-UCA[H] B cells gained, on average, fewer mutations over time than wild-type B cells. To investigate affinity maturation, we randomly selected BCRs from epitope-specific wild-type CD45.1[+]GT10[++]KO[−] GC B cells and 10E8-UCA[H] CD45.2[+]GT10[++]KO[−] GC B cells at days 7, 14 and 21 p.i. for expression as monoclonal antibodies (mAbs; hereafter GT10.2-10E8-UCA[H] or GT10.2 wild-type (GT10.2-WT) mAbs) and analyzed their affinity for the 10E8-GT10.2 probe using surface plasmon resonance (SPR). Pretransfer (day −1) wild-type or 10E8-UCA[H] GT10[++]KO[−] naive B cells identified by scBCR-seq were expressed as 'naive' wild-type or 10E8-UCA[H] control mAbs, respectively. The median affinity of naive 10E8-UCA[H] mAbs was ~660 nM, whereas that of GT10.2-10E8-UCA[H] mAbs at day 7 or 14 p.i. was 565 nM and 1.5 µM, respectively (Fig. 4d). Most naive wild-type mAbs had undetectable (>50 µM) affinity, but GT10.2-WT mAbs reached median affinities of ~20 nM at day 7 p.i., ~30-fold greater than GT10.2-10E8-UCA[H] at day 7 p.i. (Fig. 4d). The affinity of GT10.2-WT mAbs further improved at day 14 (17.6 nM) and day 21 (17 nM) p.i. (Fig. 4d), indicating a sizable median affinity gap between GT10.2-10E8-UCA[H] and GT10.2-WT mAbs starting day 7 p.i.

To investigate whether an increase in initial precursor B cell frequency extended the presence of 10E8-UCA[H] B cells in the GC, we transferred 5 times ($1 \times 10^6$) or 25 times ($25 \times 10^6$) more CD45.2 10E8-UCA[H] B

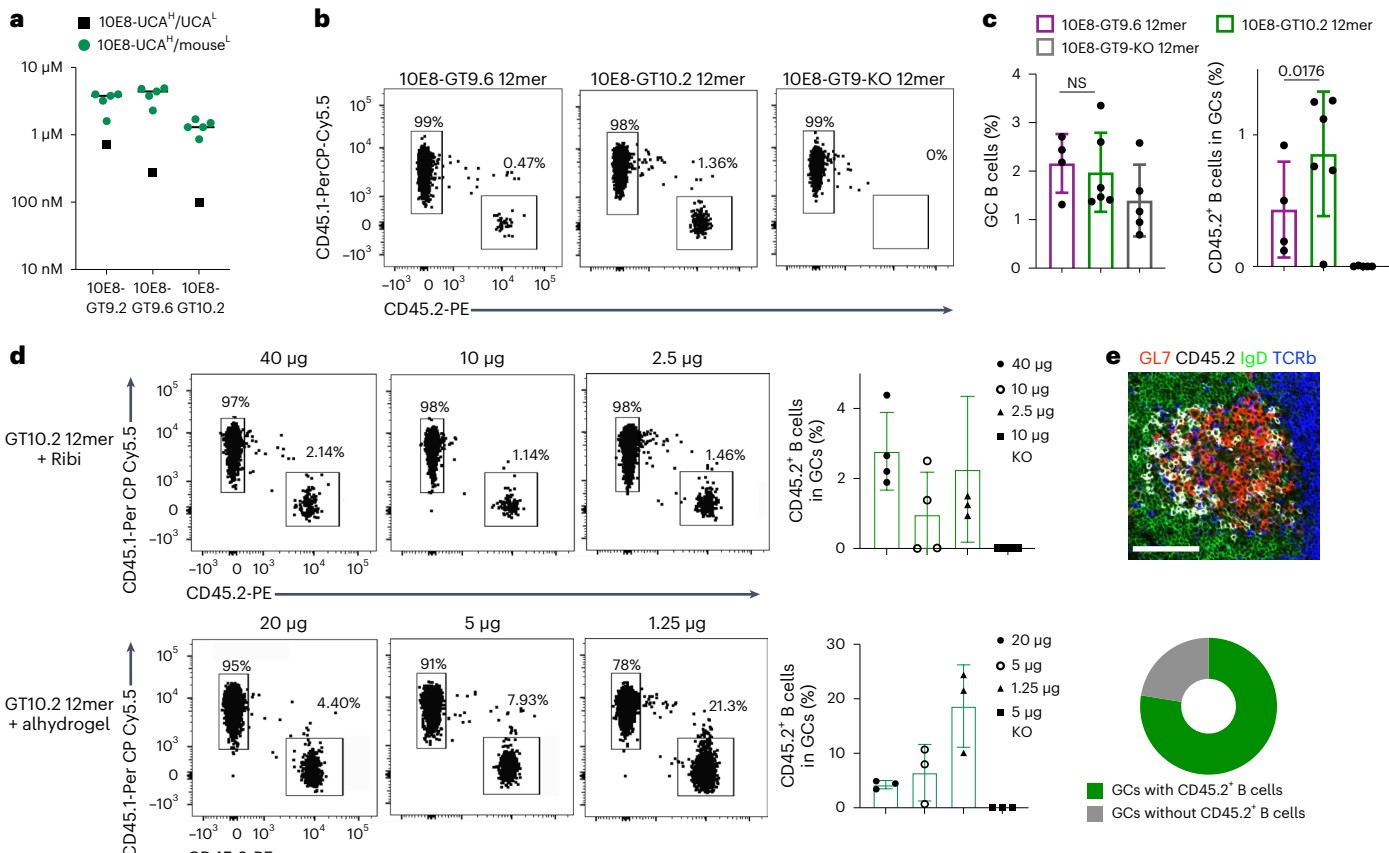

**Fig. 3 | Antigen formulation significantly enhances epitope-specific 10E8 B cells in GCs. a**, Affinities of 10E8-UCA$^H$/mouse$^L$ Fabs or human 10E8-UCA$^H$/UCA$^L$ for 10E8-GT9.2, 10E8-GT9.6 and 10E8-GT10.2 12mers in nanoparticle SPR assays. Symbols represent individual antibodies, and lines represent 10E8-UCA$^H$/mouse$^L$ median values. **b,c**, Representative flow cytometry plots of CD45.2$^+$ and CD45.1$^+$ GC B cells (**b**) and percentage of GC B cells and CD45.2$^+$ GC B cells (**c**) at day 7 p.i. in the spleens of CD45.1 wild-type mice transferred with CD45.2 10E8-UCA$^H$ B cells (frequency of 1:10$^4$ CD45.2$^+$10E8-UCA$^H$ B cells) and immunized 1 day later with 50 μg of 10E8-GT9.6 12mer, 10E8-GT10.2 12mer and 10E8-GT9-KO 12mer with Ribi adjuvant. Error bars indicate mean ± s.d. from mice in each group (n = 4–6). Significance was calculated with a two-tailed, paired t-test. Data shown are from one of two independent experiments; NS, not significant. **d**, Representative flow

cytometry plots (left) and quantifications (right) of 10E8-UCA$^H$ CD45.2$^+$ B cells in GC B cells at day 7 p.i. in the spleens of CD45.1 wild-type mice transferred with CD45.2 10E8-UCA$^H$ B cells (frequency of 1:10$^4$ CD45.2$^+$10E8-UCA$^H$ B cells) immunized with 10E8-GT10.2 12mer (40 μg, 10 μg and 2.5 μg) or 10 μg of 10E8-GT9-KO 12mer with Ribi adjuvant (n = 4; top) or with 10E8-GT10.2 12mer (20 μg, 5 μg and 1.25 μg) or 5 μg of 10E8-GT9-KO 12mer with alhydrogel adjuvant (bottom). Error bars indicate mean ± s.d. from mice in each group (n = 3). Alhydrogel data are representative of three independent experiments, and Ribi data are representative of two independent experiments, with one quantified. **e**, Representative immunohistochemistry image of the spleen (top) and percentage (bottom) of 10E8-UCA$^H$ CD45.2$^+$ B cells in GCs in recipients at day 7 p.i., as in **d**. Scale bar, 50 μm. The image is representative of images obtained from four mice.

cells into CD45.1 wild-type mice to establish frequencies of 5:10$^4$ and 25:10$^4$, respectively, 1 day before immunization with 10E8-GT10.2-12mer and used frequencies of 1:10$^4$ CD45.2 10E8-UCA$^H$ B cells as a control. A frequency of 5:10$^4$ did not increase the recruitment of CD45.2$^+$ GC B cells (~3%) relative to 1:10$^4$ (~6.9%) at day 7 p.i., whereas an initial frequency of 25:10$^4$ increased CD45.2$^+$ GC B cell (~27%) recruitment at day 7 p.i. (Extended Data Fig. 8f,g), but CD45.2$^+$ GC B cells were competed out at day 14 p.i. (Extended Data Fig. 8f,g). As such, competition from higher-affinity endogenous CD45.1$^+$ B cells might have interfered with the maintenance of 10E8-UCA$^H$ precursor B cells in GCs, even when initial precursor recruitment to the GC was high.

**Off-target endogenous antibodies block 10E8 responses**

Next, we determined the crystal structures of 10E8-GT10.2 complexed with Fabs of one GT10.2-WT mAb and one GT10.2-10E8-UCA$^H$ mAb isolated at day 14 p.i. The tip of GT10.2-10E8-UCA$^H$ mAb bound to the designed binding pocket of 10E8-GT10.2 in a manner consistent with the mature 10E8 mAb (Fig. 5a). The GT10.2-WT mAb bound to an epitope adjacent to, but partially overlapping with, the MPER helix graft of the immunogen (Fig. 5a), suggesting a site of competing antibody

responses. Although GT10.2-WT mAb does not sterically clash with the GT10.2-10E8-UCA$^H$ mAb (Fig. 5b), it does substantially overlap with the mature 10E8 binding mode (Fig. 5b).

To test whether endogenous CD45.1 wild-type antibodies inhibited the binding of CD45.2 10E8-UCA$^H$ mAbs, we incubated biotinylated 10E8-GT10.2 conjugated to streptavidin beads with the GT10.2-10E8-UCA$^H$ or GT10.2-WT mAbs for which the structures were determined and used 10E8-GT10.2 conjugated to streptavidin beads as a control, followed by incubation with a fluorescently labeled GT10.2-10E8-UCA$^H$-PE mAb (Fig. 5c). We observed complete blockade of GT10.2-10E8-UCA$^H$-PE mAb binding to 10E8-GT10.2 in competition with unlabeled GT10.2-10E8-UCA$^H$ mAb (Fig. 5c), indicating the validity of the inhibition assay. When used to compete for 10E8-GT10.2 binding, GT10.2-WT mAb substantially inhibited the binding of GT10.2-10E8-UCA$^H$-PE mAb (Fig. 5c and Extended Data Fig. 8e), indicating that GT10.2-WT mAb competed for binding on 10E8-GT10.2. To analyze the competitive effect of wild-type mAbs in vivo, we injected GT10.2-WT mAb intraperitoneally (i.p.) in CD45.1 wild-type mice and concurrently transferred CD45.2 10E8-UCA$^H$ B cells (frequency 1:10$^4$) 1 day before immunization with 10E8-GT10.2 12mer[40] (Fig. 5d). We found

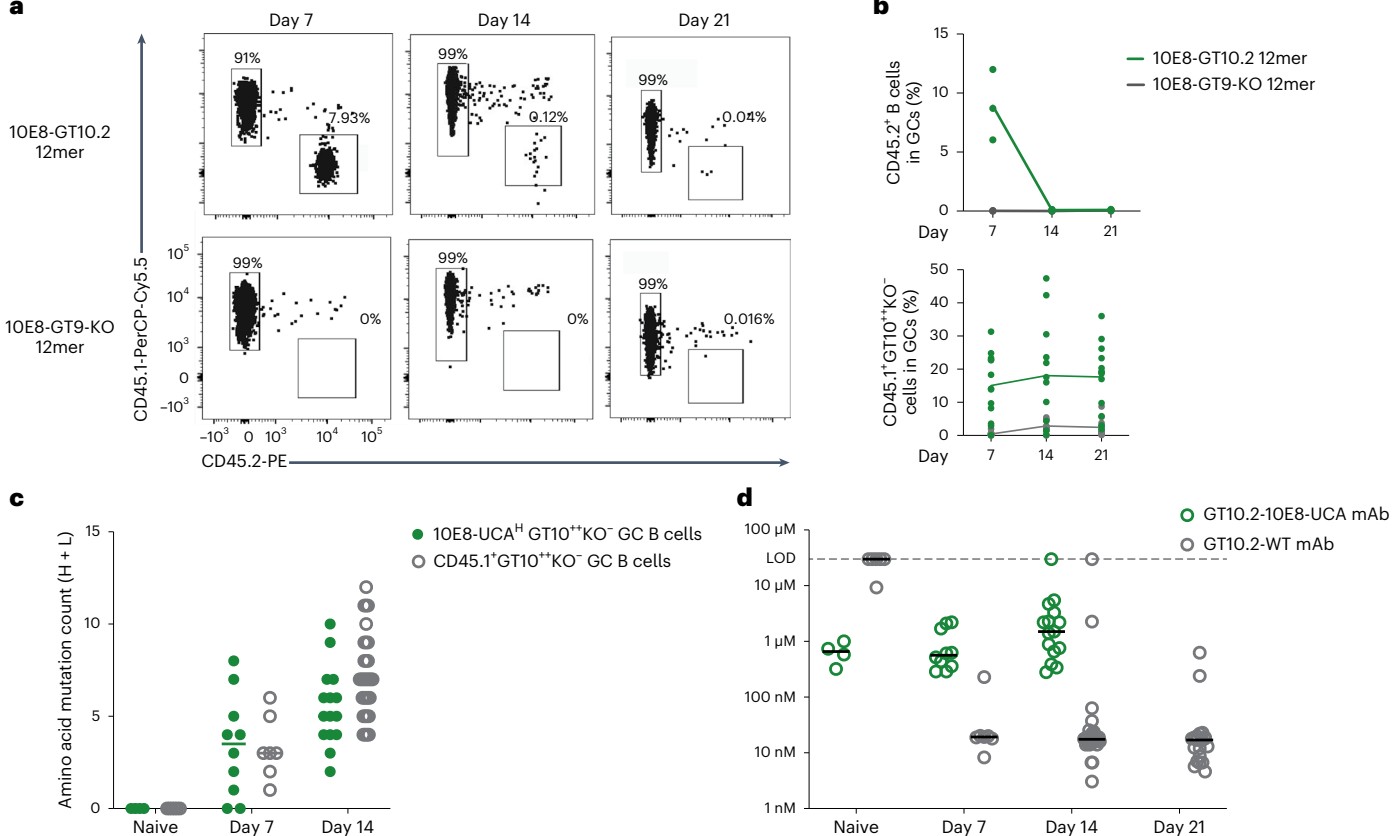

**Fig. 4 | 10E8 B cells do not persist in GCs. a**, Flow cytometry plots of CD45.2+ GC B and CD45.1+ GC B cells at days 7, 14 and 21 p.i. in the spleens of wild-type CD45.1 mice adoptively transferred (intravenously) with 200,000 CD45.2 10E8-UCA^H B cells to establish a frequency of 1:10^4 10E8-UCA^H B precursors and immunized 1 day later with 5 µg of 10E8-GT10.2 12mer (n = 3 mice) or 5 µg of 10E8-GT9-KO 12mer (n = 3) with alhydrogel. Representative data from one of three independent experiments are shown. **b**, Percentage of CD45.2+ GC B cells in splenic GCs (top) and percentage of CD45.1+GT10++KO− GC B cells in splenic GCs (bottom) at days 7, 14 and 21 p.i. as in **a** (n = 3). Each circle represents one mouse. Top, representative data from one of three independent experiments; lines mark the means. Bottom, data are from pooled groups from three independent experiments; lines mark

the respective means. **c**, Amino acid SHM of 10E8-UCA^H CD45.2+GT10++KO− B cells isolated from spleen GCs at days 7 and 14 p.i. with 10E8-GT10.2 12mer, endogenous CD45.1+GT10++KO− B cells isolated from spleen GCs at day 14 p.i. with 10E8-GT10.2 60mer and pretransfer (day −1, naive) 10E8-UCA^H GT10++KO− and wild-type GT10++KO− B cells as a control. Circles represent individual sequences, and lines indicate median values. **d**, Affinities of monomeric mAbs expressed from 10E8-UCA^H CD45.2+GT10++KO− B cells (GT10.2-10E8-UCA^H), endogenous CD45.1+GT10++KO− B cells (GT10.2-WT) and pretransfer (day −1) 10E8-UCA^H GT10++KO− B cells and wild-type GT10++KO− B cells (naive), as in **c**, determined by monomer SPR assay. Circles represent individual antibodies, and lines represent median values; LOD, limit of detection.

almost no GC recruitment of 10E8-UCA^H CD45.2+ B cells at day 7 p.i. in mice injected with GT10.2-WT mAb (mean: 0.08%), in contrast to mice that did not receive GT10.2-WT mAbs (mean: 0.65%; Fig. 5d,e). These observations indicated that endogenous mouse antibodies could sterically impede the binding of 10E8-UCA^H to 10E8-GT10 immunogens and outcompete the mature 10E8 bnAb binding mode during antibody maturation.

**10E8-GT10.3 12mer sustains GC responses**

Next, we used directed evolution to improve the affinity of 10E8-GT immunogens for 10E8-UCA^H B cells. In contrast to 10E8-GT10.2, 10E8-GT10.3 contained N34D, which removed a predicted (although unoccupied in 10E8-GT10.2)[25] N-linked glycosylation site close to the MPER as well as I106L, a GT mutation (I682L in Hxb2 numbering) rarely found in natural isolates (<0.1%, hiv.lanl.gov; Fig. 6a). 10E8-GT10.3 had an approximately threefold higher affinity for human 10E8-UCA^H/UCA^L than 10E8-GT10.2 (Fig. 6b) but retained similar affinity as 10E8-GT10.2 for 10E8-UCA^H/mouse^L antibodies (Fig. 6b). Immunization of CD45.1 wild-type mice adoptively transferred with CD45.2 10E8-UCA^H B cells (frequency of 1:10^4) with 5 µg of 10E8-GT10.3 12mer adjuvanted with alhydrogel resulted in similar GC sizes (mean: 1.3%) as after immunization with 10E8-GT10.2 12mer (mean: 1.16%; Extended Data Fig. 9a,b) but

higher frequencies of CD45.2+ GC B cells in splenic GCs (7.17%) than 10E8-GT10.2 (3.01%; Fig. 6c,d) at day 7 p.i. Approximately 80% of the CD45.2+ GC B cells bound 10E8-GT10.2 12mer, but not 10E8-GT10.2-KO 12mer (Extended Data Fig. 9c), suggesting that the response was epitope specific. Fewer epitope-specific CD45.1+GT10++KO− GC B cells were detected on day 21 after immunization with 10E8-GT10.3 12mer compared to 10E8-GT10.2 12mer immunization (Extended Data Fig. 9d).

To determine the effect of 10E8-GT10.3 12mer immunization on the affinity maturation of 10E8-UCA^H and wild-type mAbs, we performed scBCR-seq of epitope-specific wild-type CD45.1+GT10++KO− GC B cells and 10E8-UCA^H CD45.2+GT10++KO− GC B cells at days 7, 14 and 21 p.i. and inferred the germline sequences (iGL sequences) of these BCRs. A random selection of these sequences, including iGLs, were expressed as GT10.3-10E8-UCA^H and GT10.3-WT mAbs, and their binding to 10E8-GT10.2 was assessed by SPR analysis (as 10E8-GT10.3 showed a propensity to dimerize). We did not observe an increase in affinity for GT10.3-10E8-UCA^H mAbs at days 7 and 14 p.i. relative to iGL (Fig. 6e), but the gap between the median affinity of GT10.3-10E8-UCA^H and GT10.3-WT mAbs from days 7 and 14 p.i. was smaller than that previously observed between GT10.2-10E8-UCA^H and GT10.2-WT mAbs after GT10.2 12mer immunization. In contrast to GT10.2 12mer immunization, GT10.3-10E8-UCA^H mAbs were recovered at day 21 p.i., but only

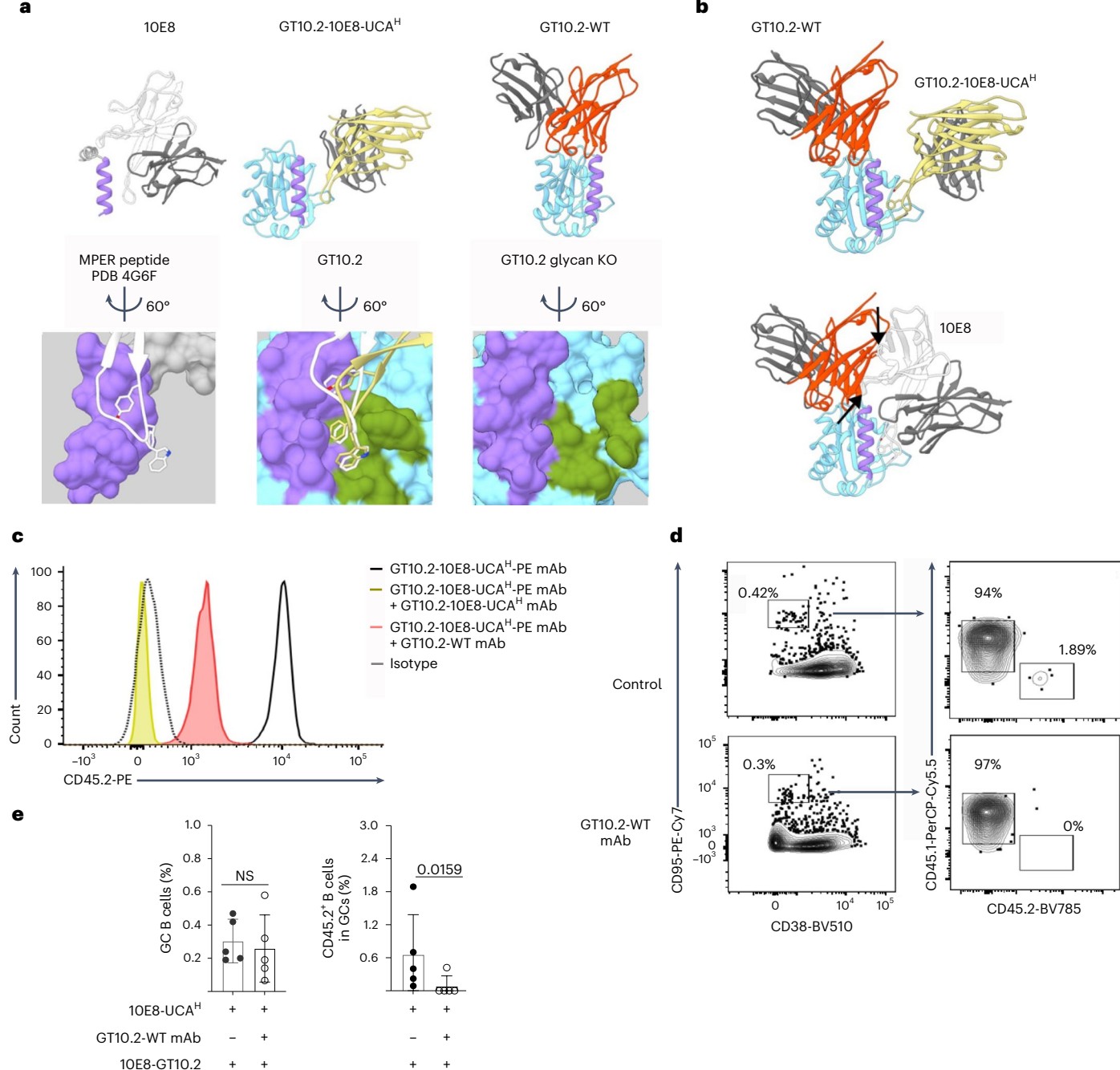

**Fig. 5 | Antibodies produced p.i. by 10E8-UCA$^H$ B cells face competition from wild-type antibodies. a**, Orientation of mature 10E8 Fab in relation to the MPER peptide in the published Protein Data Bank (PDB) 4G6F crystal structure[12] (heavy chain, white; light chain, dark gray; MPER peptide, purple; top left) and surface rendering and positioning of the critical YxFW residues of HCDR3 (bottom left), representative GT10.2-10E8-UCA$^H$ mAb in complex with a glycan-knockout (KO) version of 10E8-GT10.2 (heavy chain, yellow; light chain, dark gray; targeted MPER graft with GT mutations, purple; top center; structure aligned and oriented to the MPER peptide as in the top left) and designed binding pocket (green) and engagement of the GT10.2-10E8-UCA$^H$ mAb HCDR3 (yellow) compared to mature 10E8 HCDR3 (white; bottom center) and representative GT10.2-WT mAb in complex with 10E8-GT10.2 (heavy chain, red; light chain, dark gray; targeted MPER graft with GT mutations, purple; top right) and binding pocket (green; bottom right). **b**, Overlay of the GT10.2-10E8-UCA$^H$ and GT10.2-WT mAbs described in **a**, showing the representative competitor response in relation to

on-target response (top) and overlay of the GT10.2-WT mAb with mature 10E8 mAb from **a** (bottom). Arrows indicate clashes of the antibodies. **c**, Representative flow cytometry histograms showing the mean fluorescence intensity of fluorescently labeled GT10.2-10E8-UCA$^H$-PE mAb to 10E8-GT10.2-coated streptavidin beads in the presence of GT10.2-10E8-UCA$^H$ mAb or GT10.2-WT mAb or in the absence of other mAbs. Data are derived from one of two experiments. **d,e**, Representative flow cytometry plots (**d**) and quantifications (**e**) of GC B cells and CD45.2$^+$ B cells in GC B cells in the spleen at day 7 p.i. in CD45.1 wild-type mice adoptively transferred with CD45.2 10E8-UCA$^H$ B cells to reach a frequency of 1:10$^4$ CD45.2 10E8-UCA$^H$ B cells 1 day before i.p. injection with or without 10 μg of GT10.2-WT mAb, followed by immunization with 5 μg of 10E8-GT10.2 12mer with alhydrogel 12 h later. Error bars indicate mean ± s.d. from mice in each group (*n* = 5). Data were analyzed with a two-tailed Mann–Whitney test and are derived from one experiment.

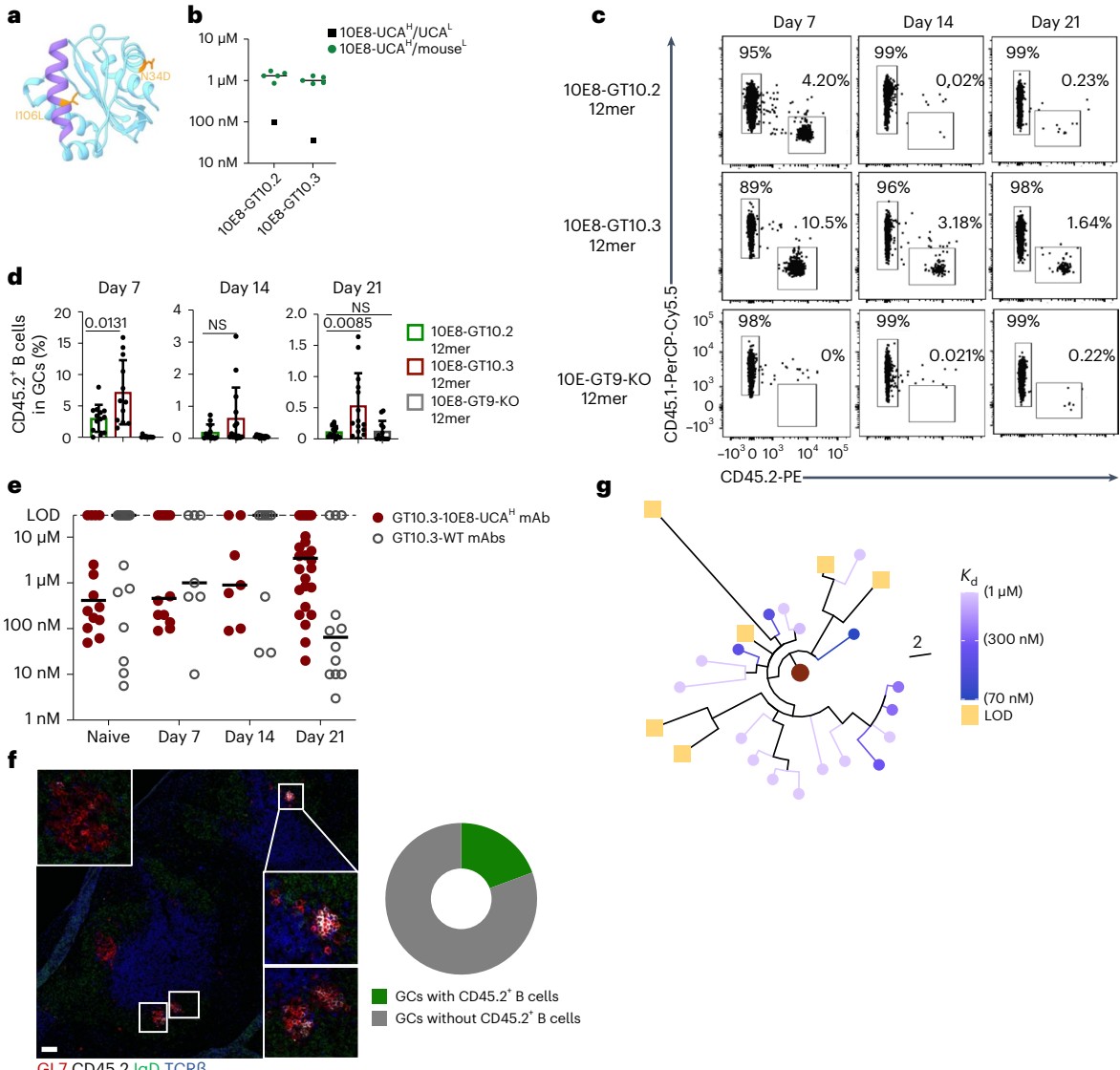

**Fig. 6 | 10E8-GT10.3 improves sustenance of 10E8-UCA^H B cells in GCs.**
**a**, Location of mutations introduced in 10E8-GT10.2 to generate 10E8-GT10.3 (MPER, purple; scaffold backbone, cyan; mutations, orange). **b**, Affinities of human 10E8-UCA^H/UCA^L Fab and Fabs of 10E8-UCA^H/mouse^L antibodies (sorted based on B220^+IgM^+ expression on B cells) to 10E8-GT10.2 12mer and 10E8-GT10.3 12mer in nanoparticle SPR assays. Symbols represent individual antibodies, and lines represent 10E8-UCA^H/mouse^L medians. **c,d**, Representative flow cytometry plots of CD45.2^+ and CD45.1^+ B cells (**c**) and percentages of CD45.2^+ B cells (**d**) in spleen GCs at days 7, 14 and 21 p.i. in CD45.1 wild-type mice transferred with CD45.2 10E8-UCA^H B cells to reach 1:10^4 10E8-UCA^H B cells and immunized 1 day later with 5 μg of 10E8-GT10.2 12mer, 10E8-GT10.3 12mer or 10E8-GT9-KO 12mer with alhydrogel. Data in **d** are pooled from two independent experiments (n = 5–8). Error bars indicate mean ± s.d. from mice

in pooled groups. Significance was calculated with a Student's t-test (unpaired, two tailed). **e**, Affinities of mAbs expressed from epitope-specific 10E8-UCA^H CD45.2^+GT10^++KO^− GC B cells (GT10.3-10E8-UCA^H mAbs) and endogenous CD45.1^+GT10^++KO^− GC B cells (GT10.3-WT mAbs) at days 7, 14 and 21 p.i. with 10E8-GT10.3 12mer as in **c** and the affinity for corresponding iGL sequences (naive), as measured by monomer SPR assay. Circles represent individual antibodies, and lines indicate median values. **f**, Representative immunohistochemistry image of a spleen section at day 21 p.i. from a mouse immunized with 10E8-GT10.3 12mer and alhydrogel, as in **c** (left), and percentage of total GCs containing CD45.2^+ B cells (right). White boxes show GCs with CD45.2^+ B cells. Scale bar, 200 μm (left). **g**, Phylogenetic tree of IGHV sequences from 10E8-UCA^H B cells, colorized on the basis of affinities at day 21 p.i., as in **e**.

a few of the day 21 GT10.3-10E8-UCA^H mAbs showed an increase in affinity relative to iGL (Fig. 6e). Immunofluorescence imaging of spleen sections taken on day 21 p.i. from the 10E8-GT10.3 12mer-immunized cohort indicated that 10E8-UCA^H CD45.2^+GL7^+ B cells were clustered in ~20% of GCs (Fig. 6f).

IGHV sequences from 10E8-UCA^H GT10^++KO^− GC B cells sorted at days 7, 14 and 21 p.i. with 10E8-GT10.3 12mer from CD45.1 wild-type mice transferred with 1:10^4 CD45.2 10E8-UCA^H B cells showed ongoing SHM and diversification (Extended Data Fig. 9e,f). BCRs from epitope-specific CD45.2^+GT10^++KO^− GC B cells acquired a median of

less than two amino acid (maximum of six amino acid) mutations by day 7 p.i., which increased to a median of two amino acid mutations (maximum of six) at day 14 p.i. and a median of two amino acids (seven maximum) at day 21 p.i. (Extended Data Fig. 9e). However, there was no correlation ($R^2 = 0.01$, $P > 0.05$) between affinity and the number of amino acid mutations in the heavy chain of CD45.2 10E8-UCA^H B cells on day 21 p.i. with 10E8-GT10.3 (Extended Data Fig. 9g). Furthermore, phylogenetic analysis of sequences from day 21 p.i. with 10E8-GT10.3 suggested that lower-affinity CD45.2 10E8-UCA^H B cells were the clonal descendants of higher-affinity CD45.2 10E8-UCA^H B

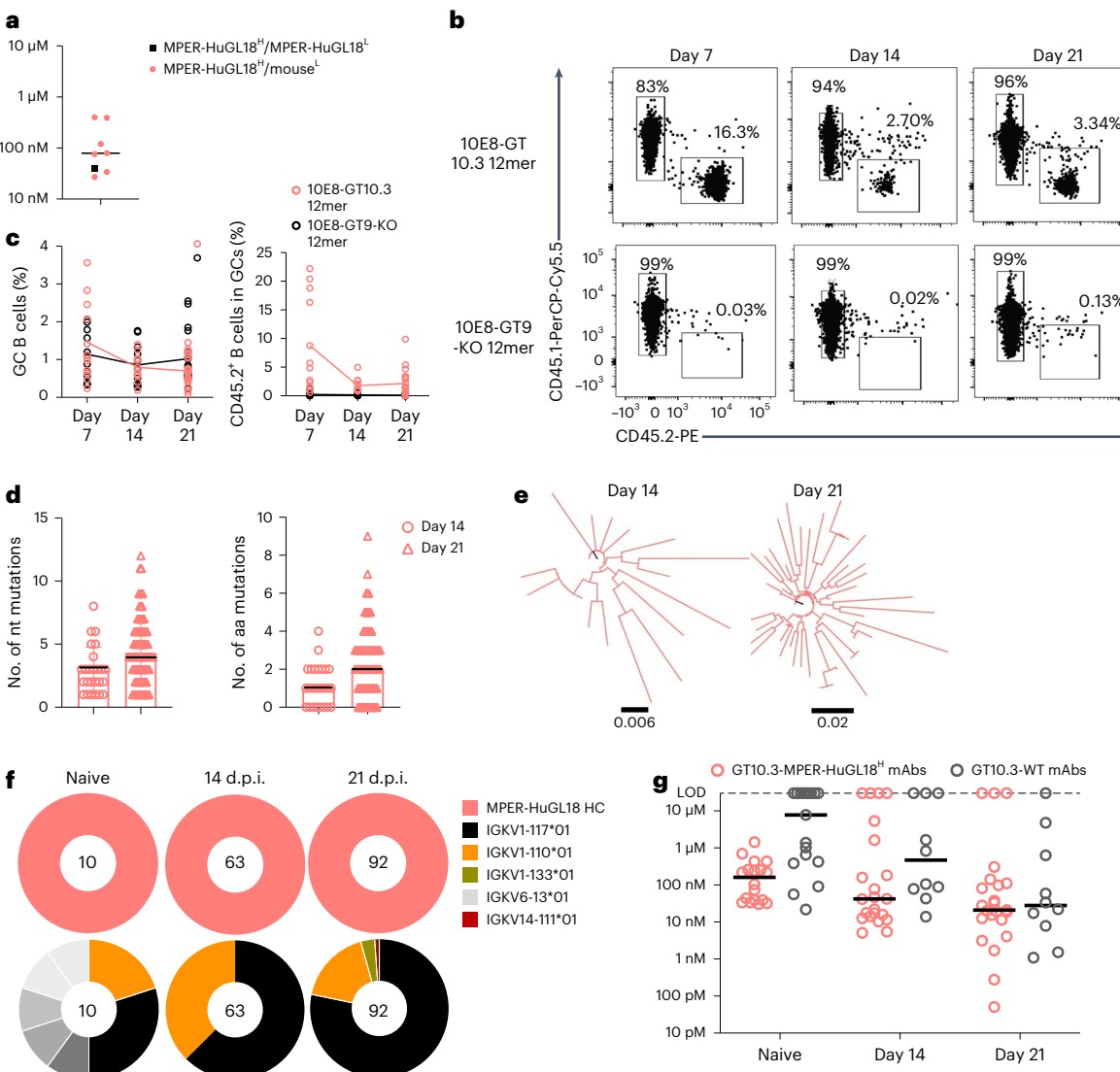

**Fig. 7 | MPER-HuGL18H mice show a sustained GC response to 10E8-GT10.3 12mer. a**, Affinities of human MPER-HuGL18H/MPER-HuGL18L Fabs and MPER-HuGL18H/mouseL Fabs to 10E8-GT10.3 12mer in the nanoparticle SPR assay. Symbols represent individual antibodies, and lines represent median MPER-HuGL18H/mouseL values. **b,c**, Representative flow cytometry plots showing CD45.2+ and CD45.1+ B cells (**b**) and quantification of the percentage of GC B cells and CD45.2+ B cells (**c**) in spleen GCs at days 7, 14 and 21 p.i. in CD45.1 wild-type mice transferred with CD45.2 MPER-HuGL18H B cells (1:10⁴ MPER-HuGL18H B cells) 1 day before immunization with 5 μg of 10E8-GT10.3 12mer or 5 μg of 10E8-GT9-KO 12mer with alhydrogel. Data were pooled from two to three independent experiments (n = 4–10 per treatment), and lines show respective means. **d**, Number of nucleotide (nt; left) and amino acid (aa; right) mutations in IGHV of epitope-specific MPER-HuGL18H CD45.2+GT10++KO− GC B cells at days

14 and 21 p.i., as in **b**. The black line indicates the median number of mutations. **e**, Phylograms showing diversification of IGHV sequences from MPER-HuGL18 CD45.2+GT10++KO− GC B cells at days 14 and 21 p.i., as in **d**. **f**, Pie plots showing MPER-HuGL18 heavy chains (top) and associated mouse light chains (bottom) sequenced from epitope-specific MPER-HuGL18H CD45.2+GT10++KO− GC B cells at days 14 and 21 p.i. with 10E8-GT10.3 12mer and alhydrogel as in **b** and pretransfer (day −1, naive) MPER-HuGL18 GT10++KO− B cells as a control. **g**, Affinities of mAbs expressed from MPER-HuGL18H CD45.2+GT10++KO− GC B cells (GT10.3-MPER-HuGL18H) and endogenous CD45.1+GT10++KO− GC B cells (GT10.3-WT) at days 14 and 21 p.i. with 10E8-GT10.3 12mer as in **b** and affinities of iGL and pretransfer (day −1) MPER-HuGL18H, with iGL wild-type mAbs for monomeric 10E8-GT10.2 (naive) as a control, as measured by monomer SPR assay. Circles represent individual antibodies, and lines mark median values.

cells (Fig. 6g), which might have lost monovalent affinity through detrimental mutation(s)[41]. Several mouse light chains were commonly found paired with the heavy chains of 10E8-UCAH CD45.2+GT10++KO− GC B cells isolated at day 21 p.i., and there was substantial affinity variation within specific heavy chain–light chain sequence pairings (Extended Data Fig. 9h,i), indicating that SHM-generated diversity in BCRs might have caused occasional affinity loss or, alternatively, that low affinities at later time points might have resulted from ongoing naive B cell entry to GCs[42]. These observations indicated that immunization with 10E8-GT10.3 12mer enabled some 10E8-UCAH B cells to compete and mature in GCs up to day 21 p.i.

## MPER-HuGL18H B cells are sustained and mature in GCs

To investigate whether 10E8-GT10 immunogens activated diverse bnAb precursors, we used C57BL/6 MPER-HuGL18H mice, which use a human germline heavy chain (MPER-HuGL18) sequenced from 10E8-GT9.2-binding naive B cells isolated from HIV-seronegative donors but are otherwise similar to 10E8-UCAH and 10E8-NGS-04H mice[25]. MPER-HuGL18 bound 10E8-GT10.3 with high affinity (~40 nM) when paired with the light chain sequenced from the same human BCR from which the heavy chain was first identified (MPER-HuGL18H/MPER-HuGL18L) and with median affinities of ~80 nM when paired with mouse light chains (MPER-HuGL18H/mouseL; Fig. 7a). MPER-HuGL18H

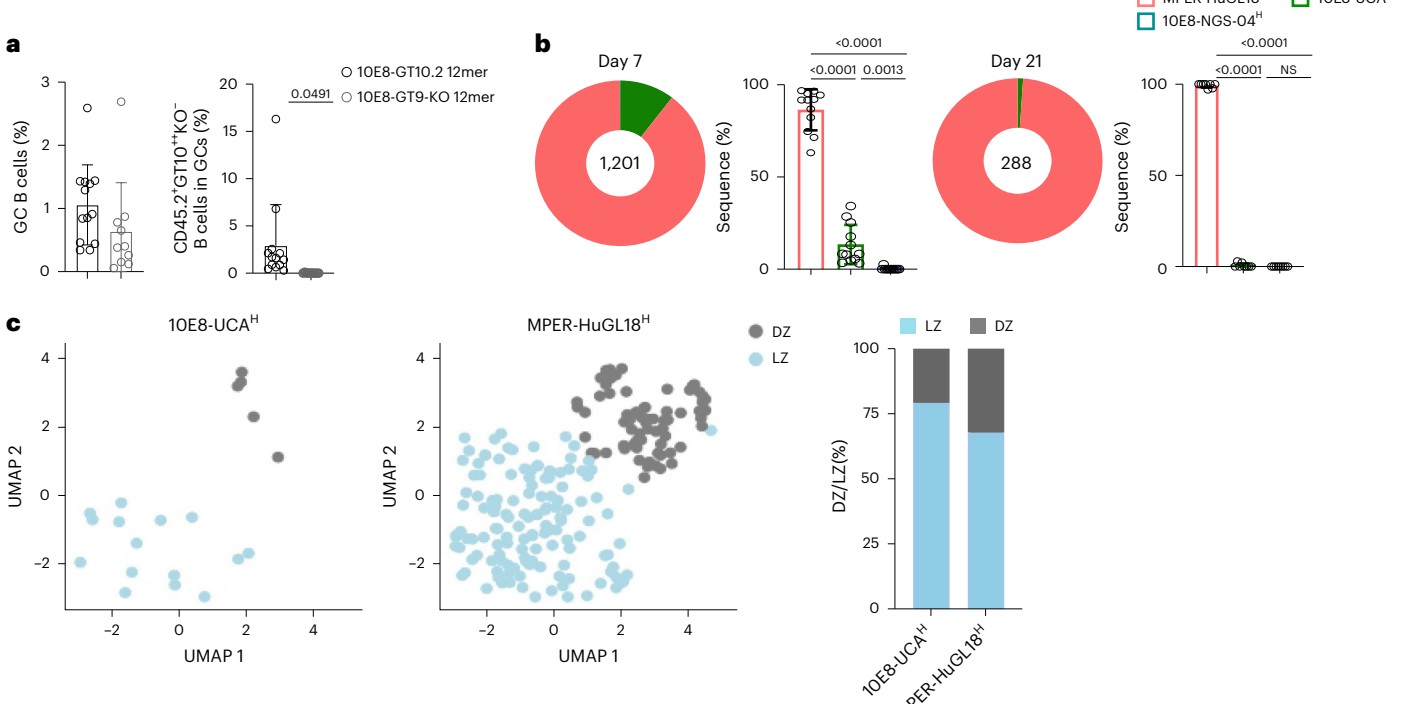

**Fig. 8 | Survival of 10E8 B cells in GCs after immunization with a clinically relevant candidate immunogen varies by precursor clonotype.**
**a**, Quantifications of the percentages of GC B cells (left) and epitope-specific CD45.2⁺GT10⁺⁺KO⁻ GC B cells (right) in spleens at day 7 p.i. in CD45.1 wild-type mice adoptively transferred with 10E8-NGS-04ᴴ, 10E8-UCAᴴ and MPER-HuGL18ᴴ CD45.2 B cells to reach a frequency of 1:10⁴ for each precursor lineage 1 day before immunization with 5 µg of 10E8-GT10.2 12mer or 10E8-GT9-KO 12mer with alhydrogel. Data are pooled from two independent experiments (*n* = 5–7 per treatment). Error bars indicate mean ± s.d. from mice in the pooled group. Significance was calculated by using a two-tailed unpaired *t*-test. **b**, Pie plots

and bar graphs showing the frequencies of MPER-HuGL18, 10E8-UCA and 10E8-NGS-04 heavy chains sequenced from CD45.2⁺GT10⁺⁺KO⁻ GC B cells at days 7 and 21 p.i. as in **a** (*n* = 12 mice). Significance was calculated by using an unpaired *t*-test with Welch's correction. **c**, Uniform manifold approximation and projection embedding of 10E8-UCAᴴ and MPER-HuGL18ᴴ CD45.2⁺GT10⁺⁺KO⁻ GC B cells from the dark zone (DZ) and light zone (LZ) of spleen GCs at day 7 p.i., as in **b** (left), and stacked bar plot showing cluster composition of 10E8-UCAᴴ and MPER-HuGL18ᴴ light zone and dark zone B cells. Bars represent the fraction of cells in dark zone/light zone clusters in each cell line.

mice demonstrated normal B cell development (Extended Data Fig. 1a–e) and expressed the MPER-HuGL18 heavy chain in 60% of peripheral blood B cells[25]. When CD45.2 MPER-HuGL18ᴴ B cells were transferred into CD45.1 wild-type mice (1:10⁴) 1 day before immunization with 10E8-GT10.3 12mer or 10E8-GT9-KO 12mer with alhydrogel (Extended Data Fig. 10a), about 8% (mean) of the B cells in spleen GCs were CD45.2⁺ GC B cells at day 7 p.i., with 1.78% at day 14 and 2.14% at day 21 (Fig. 7b,c and Extended Data Fig. 10d). At day 7 p.i., 92.6% of CD45.2⁺ GC B cells were epitope specific, as were 86% at day 14 and 67% at day 21 (Extended Data Fig. 10b). Immunofluorescence analysis of spleens demonstrated that ~20% of GCs contained MPER-HuGL18ᴴCD45.2⁺B220⁺GL-7⁺ B cells at day 21 p.i. (Extended Data Fig. 10c). MPER-HuGL18ᴴ B cells class switched and formed CD45.2⁺GT10⁺⁺B220⁺IgDˡᵒ memory B cells at day 36 p.i. (Extended Data Fig. 10e,f). scBCR-seq analysis indicated increasing mutations in the heavy chains of MPER-HuGL18ᴴCD45.2⁺GT10⁺⁺KO⁻ GC B cells sorted at days 14 (medians: nucleotides = 3, amino acids = 1) and 21 p.i. (medians: nucleotides = 4, amino acids = 2) with 10E8-GT10.3 12mer (Fig. 7d), which drove mutational diversity (Fig. 7e). A single mouse light chain (IGKV1-117) was associated with the MPER-HuGL18 heavy chains in ~80% of the CD45.2⁺GT10⁺⁺KO⁻ GC B cells found in GCs at days 14 and 21 p.i. (Fig. 7f), suggesting that light chain use might have contributed to MPER-HuGL18ᴴ B cell maintenance in the GC.

To follow the affinity maturation of MPER-HuGL18ᴴ B cells, we sequenced BCRs from epitope-specific MPER-HuGL18ᴴ CD45.2⁺GT10⁺⁺KO⁻ GC B cells and wild-type CD45.1⁺GT10⁺⁺KO⁻ GC B cells at days 14 and 21 p.i. with 10E8-GT10.3 12mer, expressed them as

GT10.3-MPER-HuGL18ᴴ and GT10.3-WT mAbs and used SPR to determine their affinities to 10E8-GT10.2 monomers. GT10.3-MPER-HuGL18ᴴ mAbs reached a median affinity of 20 nM on day 21 p.i., whereas the naive MPER-HuGL18ᴴ mAbs, which consisted of both iGL and pretransfer (day −1) BCR sequences, had a median affinity of 161 nM (Fig. 7g). The median affinity of GT10.3-WT mAbs from day 21 p.i. (30 nM) also increased relative to naive wild-type mAbs (iGLs, ~8 µM); however, GT10.3-MPER-HuGL18ᴴ mAbs showed a better median affinity than GT10.3-WT at days 14 and 21 p.i. (Fig. 7g). MPER-HuGL18ᴴ B cells using the prevalent mouse light chain IGKV1-117 had a high median affinity (~1.71 nM) but also encompassed a broad affinity range when examined on day 21 p.i. (Extended Data Fig. 10g), whereas heavy chain SHM within this specific pairing did generally increase with improved affinities, but the relationship was not significant ($R^2$ = 0.01, $P > 0.05$; Extended Data Fig. 10h). When we plotted affinity on a phylogenetic tree of 23 heavy chains paired with IGKV1-11 and other light chains, we observed that, while some highly mutated lineages were high in affinity, other highly mutated branches lost affinity (Extended Data Fig. 10i). These observations suggested that MPER-HuGL18ᴴ B cells might differ from 10E8-UCAᴴ B cells in their capacity to gain sufficient affinity to overcome endogenous competition and to remain in GCs.

### MPER-HuGL18ᴴ B cells are more competitive in GCs
To assess the importance of initial antigen affinity of the BCR for entry to the GC, we adoptively transferred CD45.2 10E8-NGS-04ᴴ, CD45.2 10E8-UCAᴴ and CD45.2 MPER-HuGL18ᴴ B cells into CD45.1 wild-type

mice at a frequency of 1:10⁴ for each precursor lineage 1 day before immunization with 10E8-GT10.2 12mer or the 10E8-GT9-KO 12mer control[25] and found that CD45.2⁺GT10⁺⁺KO⁻ B cells were recruited to GCs only after 10E8-GT10.2 12mer immunization (Fig. 8a). scBCR-seq of epitope-specific CD45.2⁺GT10⁺⁺KO⁻ GC B cells indicated that 10E8-UCA^H CD45.2⁺ B cells represented 10.01% and MPER-HuGL18^H CD45.2⁺ B cells represented 89.88% of epitope-specific CD45.2⁺GT10⁺⁺KO⁻ GC B cells at day 7 p.i., whereas the frequency of 10E8-NGS-04^H CD45.2⁺ B cells was negligible (0.11%; Fig. 8b). MPER-HuGL18^H CD45.2⁺ B cells represented >98% of the epitope-specific CD45.2⁺GT10⁺⁺KO⁻ GC B cells at day 21, whereas 10E8-UCA^H CD45.2⁺ B cells were limited to ~1%, and no 10E8-NGS-04^H CD45.2⁺ B cells were detected (Fig. 8b). At day 7 p.i., a larger fraction of MPER-HuGL18^H CD45.2⁺ B cells (32.23%) were observed in GC dark zones than 10E8-UCA^H CD45.2⁺ B cells (20.83%; Fig. 8c). Thus, MPER-HuGL18^H B cells had a greater ability to remain and proliferate in GCs.

## Discussion

Here, we showed that knock-in mice expressing heavy chain sequences derived from 10E8 precursor B cells isolated from individuals infected with HIV-1 (10E8-UCA^H) or healthy individuals (10E8-NGS-04^H) and mice expressing a heavy chain sequenced from genuine human B cell precursors binding to 10E8-GT9 immunogens (MPER-HuGL18^H)[25] had normal B cell development, in contrast to previously reported 2F5^H and 4E10^H knock-in mice[16–18], and that B cells from these mice underwent activation in response to 10E8-GT immunogens. We furthermore identified off-target sites of immunogenicity through structural analysis, and we also found that the relative affinities of 10E8 precursor B cells and endogenous precursor B cells over time drive intra-GC competition.

Endogenous competition, while observed in other models[26,29–32,43,44], was particularly critical to 10E8 immune responses in mice. We found that the 10E8-GT9 immunogen variants with affinities ranging from ~1 to 3 µM provided limited activation of 10E8 B cell precursors, although immunogens of similar or lower affinities induce sustained GC responses in models for other classes of bnAbs[29,30,32]. Crystal structures of 10E8-GT10.2 complexes showed that a wild-type mouse antibody (GT10.2-WT) isolated at day 14 p.i. bound a nearby epitope, inhibited the binding of GT10.2-10E8-UCA^H mAbs in vitro and the activation of 10E8-UCA^H B cells in vivo. Recruitment and maturation of B cells in GCs are controlled by competition and not absolute affinity[44,45], and low-affinity B cells can remain in GCs in the absence of high-affinity competitors[46]. We suspect, therefore, that the increase in baseline affinity to 10E8-UCA between 10E8-GT10.2 and 10E8-GT10.3 allowed some maintenance of 10E8-UCA B cell precursors in the GC by narrowing the affinity gap with epitope-specific endogenous B cells during competition for antigen in the GC. However, reasonable starting affinities alone were insufficient to overcome endogenous competition and ensure retention in GCs.

A major goal of GT design is the activation of diverse bnAb precursors[10,26,47,48]. The responses of MPER-HuGL18^H B cells, which use a different *IGH-V* gene than 10E8-UCA^H B cells, demonstrated that GC retention of B cells bearing 10E8-like HCDR3s after immunization with 10E8-GT10.3 12mers is possible. Higher initial recruitment of precursors to the GC did not sufficiently explain the difference in residence time between 10E8-UCA^H and MPER-HuGL18^H B cells, as 10E8-UCA^H B cells transferred at 25-fold higher frequencies and thus present in day 7 GCs at higher frequencies were still lost at day 14. The fraction of p.i. MPER-HuGL18^H Fabs achieving affinity gains compared to 10E8-UCA^H B cell-derived Fabs suggests interclonal variation in the likelihood that SHM will produce affinity maturation, perhaps due to probabilistic variation in some types of mutations or the favorability of certain codons[49,50], which may underpin the distinctive GC dynamics of 10E8-UCA^H and MPER-HuGL18^H B cells. Whatever the underlying cause, the fact that MPER-HuGL18^H B cells were substantially favored in the GCs against 10E8-UCA^H and 10E8-NGS-04^H after 10E8-GT10.2 immunization highlighted the importance of GT immunogens engaging multiple B cell precursor lineages.

Affinity is only one factor in GC recruitment and maintenance. Delivery with alhydrogel enhanced epitope-specific 10E8-UCA^H B cell recruitment, in line with reports that adjuvants strongly affect Env trimer immunogenicity[51]. Adjuvant activity is dependent on many factors, some of which (for example, Toll-like receptor activation[52–54]) vary across species; nonetheless, this model confirmed the integrity of 10E8-GT epitopes in formulations with alhydrogel. High valency, effective for the eOD-GT8 60mer, which induces VRC01-bnAb-class responses in mice[55] and in clinical trials in humans[37], did not improve 10E8-GT10.2. 10E8-GT10.2 60mers induced higher serum binding titers than 10E8-GT10.2 12mers but lower epitope-specific 10E8-UCA^H B cell recruitment to GCs, potentially because the increased differentiation into short-lived antibody-secreting cells that is associated with multimerization[55] limited GC residency.

Our study highlights the need for bnAb-class-specific approaches to GT immunogen design and offers a route toward inducing MPER-class bnAbs through the generation of new 10E8 precursor models without autoreactivity, the characterization of lineage-specific acquisition of affinity-enhancing mutations and the identification of the dynamic gap between competing BCR affinities over time as a determinant of GC survival.

## Online content

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

¹The Ragon Institute of Mass General, MIT and Harvard, Cambridge, MA, USA. ²Department of Immunology and Microbiology, The Scripps Research Institute, La Jolla, CA, USA. ³IAVI Neutralizing Antibody Center, The Scripps Research Institute, La Jolla, CA, USA. ⁴Center for HIV/AIDS Vaccine Immunology and Immunogen Discovery, The Scripps Research Institute, La Jolla, CA, USA. ⁵Institute for Drug Discovery, Leipzig University Medical Faculty, Leipzig, Germany. ⁶Department of Integrative Structural and Computational Biology, The Scripps Research Institute, La Jolla, CA, USA. ⁷Center for Infectious Disease and Vaccine Research, La Jolla Institute for Immunology (LJI), La Jolla, CA, USA. ⁸Department of Medicine, University of California, San Diego, La Jolla, CA, USA. ⁹Moderna, Inc., Cambridge, MA, USA. ¹⁰Department of Biology, Massachusetts Institute of Technology, Cambridge, MA, USA. ¹¹Present address: Moderna, Inc., Cambridge, MA, USA. ¹²These authors contributed equally: Rashmi Ray, Torben Schiffner, Xuesong Wang, Yu Yan. ✉e-mail: schief@scripps.edu; fbatista1@mgh.harvard.edu

## Methods

### Ethics statement

All animal experiments were performed under the approval of the Institutional Animal Care and Use Committee of Harvard University and the Massachusetts General Hospital (animal study protocols 2016N000022 and 2016N000286) and were conducted in accordance with the regulations of the Association for Assessment and Accreditation of Laboratory Animal Care International.

### Mice

For experiments, male B6.SJL-$Ptprc^aPepc^b$/BoyJ mice (CD45.1$^{+/+}$; 8–12 weeks of age) were purchased from The Jackson Laboratory. F0 mice from the 10E8-UCA$^H$-knock-in mouse (CD45.2$^{+/+}$) colony were bred at the animal facility of the Gene Modification Facility (Harvard University), and breeding for colony expansion and experimental procedures was subsequently performed at the Ragon Institute of Mass General, MIT and Harvard in a facility where the ambient temperature was maintained at 20 °C with 40% humidity and a 12-h light/12-dark light cycle. Ear or tail snips from 10E8-UCA$^H$-knock-in mice were genotyped by TaqMan assay under a fee-for-service agreement (TransnetYX). TaqMan probes for the genotyping assay were developed by TransnetYX. CD45.2$^+$ B cells from male or female 10E8-UCA$^H$, 10E8-NGS-04$^H$ or MPER-HuGL18$^H$ donor knock-in mice (8–16 weeks old) were enriched using a Pan B Cell Isolation kit II (Miltenyi Biotec), counted, diluted to desired cell numbers in PBS and adoptively transferred into CD45.1$^+$ recipient mice, as reported previously[29]. Further details on mouse strains as well as all reagents and tools described below can be found in Supplementary Table 3.

### Generation of 10E8-UCA$^H$-knock-in mice

10E8-UCA$^H$- and 10E8-NGS-04$^H$-knock-in mice were generated following published protocols[27,32]. In brief, the 4E10 targeting vector[56] was modified by the incorporation of human rearranged 10E8-UCA or 10E8-NGS-04 V(D)J (heavy chain construct) sequences downstream of the promoter region and by elongation of the 5′ and 3′ homology regions using the Gibson assembly method (New England Biolabs). The targeting vector DNA was confirmed by Sanger sequencing (Eton Bioscience). Next, fertilized mouse oocytes were microinjected with a donor plasmid containing the prearranged 10E8-UCA or 10E8-NGS-04 $Igh$ with the mouse VHJ558 promoter, two pairs of single guide RNAs (25 ng ml$^{-1}$) targeting the $Igh$ locus and AltR-Cas9 protein (50 ng ml$^{-1}$) and injection buffer[32]. Following culture, the resulting zygotes were implanted into the uteri of pseudopregnant surrogate C57BL/6J mothers.

### Immunizations

Both male and female donor mice were used for adoptive transfers. All transfers and immunizations were performed in male CD45.1$^{+/+}$ mice due to risk of rejection of cells from male donor mice to female recipient mice. Preparations of immunogens (10E8-GT9 12mer and 10E8-GT10 12mer nanoparticles at 5 µg per mouse (or as specified in the text)) were diluted in PBS at a volume of 100 µl per mouse for i.p. injection and then mixed at a 1:1 ratio with 2% alhydrogel (Invitrogen) or Sigma adjuvant system (Ribi) for at least 20 min. The final formulation was injected i.p. (total volume of 200 µl per mouse).

### Immunogen and flow cytometry probe production

All proteins were expressed in Freestyle 293F cells as previously described[25,48]. In brief, nanoparticles were purified by Galanthus Nivalis lectin affinity purification, followed by size-exclusion chromatography. Antibodies and Fabs were purified by protein A affinity chromatography and CaptureSelect CH1-XL affinity chromatography, respectively, and buffer exchanged into TBS. Monomeric proteins contained C-terminal His tags and were purified by immobilized metal affinity chromatography, followed by size-exclusion chromatography.

For flow cytometric probe binding, Avi-tagged 10E8-GT9 or 10E8-GT10 was biotinylated by BirA enzymatic reaction (Avidity) according to the manufacturer's protocol. Biotinylated 10E8-GT9 or 10E8-GT10 probes and respective knockouts were prereacted in independent tubes for at least 30 min in a 4:1 molar ratio with fluorescently labeled streptavidin (streptavidin-AF488 and/or streptavidin-AF647). Reagents were then combined with fluorescently labeled antibodies for fluorescence-activated cell sorting (FACS) staining.

### Enzyme-linked immunosorbent assay

Antigen-specific antibody titers were detected by enzyme-linked immunosorbent assay using anti-His (2 mg ml$^{-1}$) to capture GT10 or GT10-KO antigen (2 mg ml$^{-1}$) on 96-well plates. Plates were washed five times with 0.05% Tween 20 in PBS, blocked with 100 µl of 3% bovine serum albumin (BSA) in PBS for 1 h at room temperature and washed again before incubation with 1:2 or 1:5 serially diluted mouse serum samples for 1 h at RT. Wells were washed and incubated with Alkaline Phosphatase AffiniPure Goat Anti-Mouse IgG (Jackson ImmunoResearch, 115-055-071) at 1:1,000 in PBS with 0.5% BSA for 1 h at RT. $p$-Nitrophenyl phosphate (Sigma, N2770) dissolved in double distilled water (50 µl per well, room temperature, 25 min) was used for detection. Absorbance at 405 nm was determined with a plate reader (Synergy Neo2, BioTek). Enzyme-linked immunosorbent assay curves were calculated and analyzed using GraphPad Prism 9.5.1 (GraphPad).

### Flow cytometry

At selected time points following immunization, whole spleens were mechanically dissociated to generate single-cell suspensions. ACK lysis buffer was used to remove red blood cells, and splenocytes were then resuspended in FACS buffer (2% fetal bovine serum/PBS), Fc blocked (clone 2.4G2, BD Biosciences) and stained for viability with Live/Dead Blue (Thermo Fisher Scientific) for 20 min at 4 °C. For surface staining, GT9 or GT10 probes (described above) and antibodies (at a dilution of 1:100) to CD4-APC-eF780, CD8-APC-eF780, Gr-1-APC-eF780, F4/80-APC-eF780, B220-B510, CD95-PE-Cy7, CD38-A700, CD45.1-PerCPCy5.5, CD45.2-PE, IgD-BV786, IgM-BUV395 and IgG1-BV421 were used. Cells were acquired using a BD LSRFortessa (BD Biosciences) for flow cytometric analysis and sorted using a BD FACS Aria II instrument (BD Biosciences) and BD FACSDiva Software. Data were analyzed using FlowJo software (TreeStar). B cells were single-cell dry sorted into 96-well PCR plates, rapidly frozen on dry ice and stored at −80 °C until processing.

### BCR probe binding and dissociation measurement by flow cytometry

Peripheral blood mononuclear cells containing naive B cells from 10E8-UCA$^H$ and 10E8-NGS-04$^H$ mice were surface stained using 10E8-GT10.2 probe conjugated with streptavidin-BV421. After final washing, stained cells were incubated at room temperature for the indicated lengths of time to allow dissociation before being subjected to FACS analysis.

### 10x gene expression analysis

The Seurat R package (v4.3.0) was used to perform 10x gene expression analysis[57]. Droplets containing more than 500 unique molecular identifiers, more than 300 unique genes and a mitochondrial ratio of less than 20% were considered of poor quality and were filtered out before subsequent analysis (log$_{10}$ GenesPerUMI > 0.80 and housekeeping genes > 50 were also applied). The top 2,000 highly variable genes were selected using the 'FindVariableFeatures' function on the normalized dataset before scaling and principal-component analysis. The top 40 principal-component analysis vectors were used as input to the 'RunUMAP' and 'FindNeighbors' function. A range of resolutions were tested for the 'FindClusters' function, and a final resolution of 0.5

was used to classify cells into the dark zone and light zone, with the help of expression levels of signature genes and cell scoring using known dark zone/light zone gene lists. The 'Findmarkers' function was used to define differentially expressed genes between different groups.

### In vitro bead assay

mAb-GT10.2-CD45.2-10E8[H]-14DPI-PE was labeled using an Alexa Fluor Antibody Labeling kit (Thermo Fisher) following the manufacturer's instructions. Streptavidin-coated beads (Spherotech) were washed with PBS and incubated with biotinylated 10E8-GT10.2 at a concentration of 24 µg of protein per 0.5 mg of beads for 30 min on ice. Beads were washed three times with PBS. Competing antibody mAb-GT10.2-CD45.2-10E8[H]-14DPI or mAb-GT10.2-CD45.1-14DPI was incubated with 10E8-GT10.2-conjugated beads on ice for 30 min. Next, mAb-GT10.2-CD45.2-10E8[H]-14DPI-PE was added and incubated on ice for 30 min. Beads were washed and acquired for FACS analysis.

### scBCR-seq

Two types of scBCR-seq experiments were performed in this study: single-cell Sanger sequencing and single-cell NGS.

**Single-cell Sanger sequencing.** Following single-cell sorting of antigen-specific B cells, the genes encoding the variable region of the heavy and light chains of IgG were amplified by reverse transcription PCR. In brief, first-strand cDNA synthesis was performed using SuperScript III Reverse Transcriptase (Invitrogen) according to manufacturer's instructions. Nested PCR reactions consisting of PCR-1 and PCR-2 were performed as 25-µl reactions using HotStarTaq enzyme (Qiagen), 10 mM dNTPs (Thermo Fisher Scientific) and cocktails of *Igg*- and *Igk*-specific primers and thermocycling conditions described previously[58]. PCR products were run on precast E-Gel 96 (2%) gels with SYBR Safe (Thermo Fisher Scientific), and wells with bands of the correct size were submitted to GENEWIZ for Sanger sequencing. Heavy chain products were sequenced using the heavy chain reverse primer from PCR-2 (5′-GCTCAGGGAARTAGCCCTTGAC-3′), and the light chain was sequenced using the light chain reverse primer (5′-TGGGAAGATGGATACAGTT-3′) from PCR-2. Reads were quality checked, trimmed, aligned and analyzed using Geneious software. IMGT/V-QUEST[59,60] (http://www.imgt.org) was used for mouse/human immunoglobulin gene assignments.

**Single-cell NGS assay.** Following first-strand cDNA synthesis, nested PCR reactions consisting of PCR-1 reaction were performed in 12.5-µl reactions using HotStar Taq enzyme (Qiagen), 10 mM dNTPs (Thermo Fisher Scientific) and betaine (200 µM) with a cocktail of *Igg*-specific or *Igk*-specific primers with inline index barcodes (gene-specific targets are as described previously[58]). The following thermocycling conditions were used: initial denaturation for 5 min at 94 °C, followed by 35 cycles of 94 °C for 15 s, 65 °C for 40 s and 72 °C for 60 s, with a final extension at 72 °C for 5 min. The PCR-2 reaction further amplified the target product, and Illumina adapters were added by using a reaction mix consisting of 12.5 µl of KAPA HiFi HotStart ReadyMix (Roche, KK2602), 9.5 µl of prime adapter matrix (1 µM; Supplementary Table 1) and 3 µl of PCR-1 product for a total reaction volume of 25 µl. The following thermocycling conditions were used: initial denaturation for 5 min at 98 °C, followed by 5 cycles of 98 °C for 15 s, 60 °C for 90 s and 72 °C for 90 s, followed by 35 cycles of 98 °C for 15 s, 65 °C for 40 s and 72 °C for 60 s, with a final extension of 72 °C for 5 min. At this point, all products were pooled into a single 1.7-µl Eppendorf tube, where a two-sided bead cleanup was performed using SPRIselect (Beckman Coulter, B23318) at a 1:2 ratio followed by a 1:5 ratio. Pooled libraries were then checked for size using a Tapestation (Agilent) and quantified using Qubit (Thermo Fisher Scientific). Completed libraries were pooled at equimolar concentrations and sent for sequencing on an Illumina MiSeq using a 600-cycle kit (Genewiz).

### Contig assembly and demultiplexing

Full-length contigs were assembled from the paired-end reads using FLASH (https://ccb.jhu.edu/software/FLASH)[61]. Following assembly, a custom Python script was used to demultiplex the resulting contigs into their respective plates using the inline index barcode. The contigs were further demultiplexed to individual wells (cells) using Illumina index pairs (Supplementary Table 1). Consensus sequences were then annotated using mixCR (https://mixcr.com)[62,63] and a modified IMGT/V-QUEST (http://www.imgt.org) reference.

### Immunofluorescence

Spleens were frozen in Tissue-Tek OCT compound (VWR) and cryosectioned. Sections were fixed with 4% paraformaldehyde for 10 min, washed three times in PBS, blocked for 45 min with 10% goat serum in PBS and stained for 1 h with 100-fold diluted antibodies (Pacific Blue anti-mouse B220, AlexaFluor 488 anti-mouse CD45.2, AlexaFluor 594 anti-mouse CD3/TCRβ and AlexaFluor 647 anti-mouse GL7). After washing three times with PBS, stained sections were mounted with ProLong Diamond Antifade Mountant (Thermo Fisher). Images were acquired on a Zeiss Elyra PS.1 confocal microscope and analyzed in ImageJ 2.0.0.

### Immunogen design

Immunogens were optimized using directed evolution as described previously[10,25,64]. Error-prone PCR libraries of the respective previous best 10E8-GT immunogens were created using a Random Mutagenesis kit and transformed into yeast cells. Induced cells were stained with serum from wild-type mice immunized with 10E8-GT9.2 12mer and 10E8-UCA, followed by labeling with PE-conjugated anti-mouse IgG and AlexaFluor 647-conjugated anti-human IgG and FITC-conjugated anti-cMyc. FITC-positive, PE-low and AlexaFluor 647-high cells were sorted on a BD Influx and regrown. After three rounds of enrichment, cells were plated on SD-Trp-Ura agar plates directly after sorting and grown at 30 °C for 3 days. Colonies were picked, inserts were amplified using a Phire Plant Direct PCR kit, and unpurified PCR products were sequenced by Sanger sequencing (Azenta Life Sciences).

### SPR

The following two types of SPR experiments were used in this study: the nanoparticle SPR assay and the monomer SPR assay.

The nanoparticle SPR assay was performed on a Biacore 8K instrument using CM3 sensor chips (Cytiva) and 1× HBS-EP+ (pH 7.4) running buffer (20× stock from Teknova, H8022) supplemented with BSA at 1 mg ml[−1]. In a typical experiment, approximately 4,000 response units of capture antibody (SA684_Scaf_10E8-GT10_W6-01) in 10 mM sodium acetate (pH 4.5) was amine coupled to the sensor surface using N-hydroxysuccinimide and 1-ethyl-3-(3-dimethylaminopropyl) carbodiimidehydrochloride from an Amine Coupling kit (GE Healthcare, BR-1000-50). In each cycle, 10E8-GT 12mers were captured on the surface for 30 s at 1 µg ml[−1], followed by injection of a dilution series of Fab for 60 s with a 600-s dissociation time. Phosphoric acid (0.85%) was used as the regeneration solution with 30 s contact time and was injected three times per each cycle. Raw sensograms were analyzed using Biacore Evaluation software (Cytiva), reference spot and blank double referencing, fitting to a Langmuir model.

A monomer SPR assay was used for experiments that included large numbers of antibodies. Experiments were performed on a ProteOn XPR36 (Bio-Rad) using a GLC Sensor Chip (Bio-Rad; Fig. 4) or a Carterra LSA instrument using HC30M sensor chips (Carterra; Figs. 6 and 7). In these experiments, anti-human IgG Fc capture antibody was immobilized onto the surface, as described previously. In each cycle, antibodies were captured, and titration series of monomeric 10E8-GT immunogens were flowed over the surface for 60 s, followed by a 600-s dissociation time.

## Crystallization and structure determination

The CD45.1 Fab–GT10.2 complex and CD45.2 Fab–GT10.2 glycan-knockout complex were adjusted to 12.1 and 10.0 mg ml$^{-1}$ in TBS buffer (20 mM Tris and 150 mM sodium chloride; pH 7.6). The complexes were screened for crystallization with the JCSG Core Suite (Qiagen) as precipitant on a robotic high-throughput CrystalMation system (Rigaku) at The Scripps Research Institute. Crystal screening was set up by vapor diffusion in sitting drops with 0.1 µl of protein and 0.1 µl of reservoir solution. The optimized crystallization conditions were 2 M ammonium sulfate for the antibody CD45.1 Fab–GT10.2 complex and 0.2 M sodium chloride, 0.1 M phosphate citrate and 20% PEG8000 for the antibody CD45.2 Fab–GT10.2 glycan-knockout complex, respectively. Crystals were incubated at 20 °C, collected within 21 days and soaked in reservoir solution containing 15% (vol/vol) ethylene glycol for the antibody CD45.1 Fab–GT10.2 complex and 20% (vol/vol) glycerol for the antibody CD45.2 Fab–GT10.2 glycan-knockout complex as cryoprotectant, respectively. The collected crystals were then flash-cooled and stored in liquid nitrogen until data collection. Diffraction data were collected at cryogenic temperature (100 K) at Advanced Light Source (at Lawrence Berkeley National Laboratory) beamline 5.0.1 with a beam wavelength of 0.97741 Å for the antibody CD45.1 Fab–GT10.2 complex and at the Argonne Photon Source beamline 23-ID-B with a beam wavelength of 1.0337 Å for the antibody CD45.2 Fab–GT10.2 glycan-knockout complex. The diffraction data were processed with HKL2000 (ref. 65). The complex structures were solved by molecular replacement using Phaser[66] with the $V_H$–$V_L$ model generated by Repertoire Builder (https://sysimm.org/rep_builder/) for both antibody CD45.1 and antibody CD45.2 Fabs and T117v2 (PDB 5T80)[67] for GT10.2 with or without glycans. Iterative model building and refinement were performed in Coot[68] and PHENIX[69], respectively. Further details are provided in Supplementary Table 2.

## Statistics and reproducibility

For immunization studies, statistical analysis was performed in Prism 9.01 (GraphPad) using a two-tailed Student's $t$-test or as specified in the figure legends. $P$ values less than 0.05 were considered significant (*$P < 0.05$; **$P < 0.01$; ***$P < 0.001$; ****$P < 0.0001$), as indicated in the figure legends. Group size was determined based on prior work in the field with similar immunogens[31,32]; no statistical method was used to predetermine sample size. No data were excluded from the analysis, and mice were randomly assigned to groups. The investigators were not blinded to allocation during experiments and outcome assessment. Data distribution was assumed to be normal, but this was not formally tested.

## Reporting summary

Further information on research design is available in the Nature Portfolio Reporting Summary linked to this article.

## Data availability

BCR sequences are available at NCBI Genbank under accession numbers PP617372–PP617659 and PP617660–PP618659, and crystal structures are available at the Protein Data Bank (PDB) under accession IDs 9BDH (for CD45.1 in complex with GT10.2) and 9BDI (for CD45.2 in complex with GT10.2). Owing to the extent and complexity of the remaining underlying datasets, all additional information required for reanalysis is available from the lead contact upon request.

## Code availability

No original code is reported in this manuscript.

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

## Acknowledgements

We thank all members of the laboratory of F.D.B. for their assistance with the immunization experiments. Support for this work was provided by the Collaboration for AIDS Vaccine Discovery grants OPP1084519 and OPP1147787 (to W.R.S.), INV-034657 (to W.R.S. and I.A.W.), OPP1196345/INV-008813 (to W.R.S.) and INV009585 and INV046626 (to F.D.B.) funded by the Bill and Melinda Gates Foundation. Funding was also provided by the National Institute of Allergy and Infectious Diseases UM1 AI144462 (Scripps Consortium for HIV/AIDS Vaccine Development; to W.R.S., I.A.W. and F.D.B.) and R01 AI147826 (to W.R.S., I.A.W. and F.D.B.) and by the Ragon Institute of Mass General, MIT and Harvard (to W.R.S. and F.D.B.). The funders had no role in study design, data collection and analysis, decision to publish or preparation of the manuscript.

## Author contributions

R.R. planned, designed and performed in vivo experiments, serology, FACS, single-cell sorting and BCR-seq. T.S. provided immunogens, probes, elicited antibodies and antibody sequences for BCR knock-ins and helped plan experiments. X.W., Y.Y. and E.L. helped in mouse immunization experiments. I.P. and S.C. provided human antibody sequences. C.-C.D.L. conducted crystallography studies. S.G. helped with immunofluorescence staining and imaging. T.S., K.R., O.S. and O.K. performed SPR analyses. J.W. performed single-cell NGS sequencing. S. Parikh helped with spleen and bone marrow staining. R.A.R. helped with the bead assays.

G.A.D. helped with sequence analysis. U.N., Y.-C.L. and L.M. generated mouse models. Y.-C.L., S. Pecetta, S.K. and E.M. helped with in vivo experiments. K.H.K. supervised mouse colony management. N.P., R.T., E.G., Y.A. and M.K. produced and purified immunogens and antibodies. S.R.W. contributed to manuscript writing/editing. I.A.W. supervised the structural studies. R.R., T.S., W.R.S. and F.D.B. interpreted data. W.R.S. and F.D.B. conceived and supervised the studies. R.R., T.S. and F.D.B. wrote the paper. O.S., K.R., I.A.W. and W.R.S. edited the paper.

## Competing interests

T.S., O.S. and W.R.S. are named inventors on patent applications filed by Scripps and IAVI regarding 10E8-GT immunogens and nanoparticles in this manuscript. F.D.B. has consultancy relationships with Adimab, Third Rock Ventures and *The EMBO Journal*. W.R.S. is an employee of Moderna, Inc. All other authors declare no competing interests.

## Additional information

**Extended data** is available for this paper at https://doi.org/10.1038/s41590-024-01844-7.

**Correspondence and requests for materials** should be addressed to William R. Schief or Facundo D. Batista.

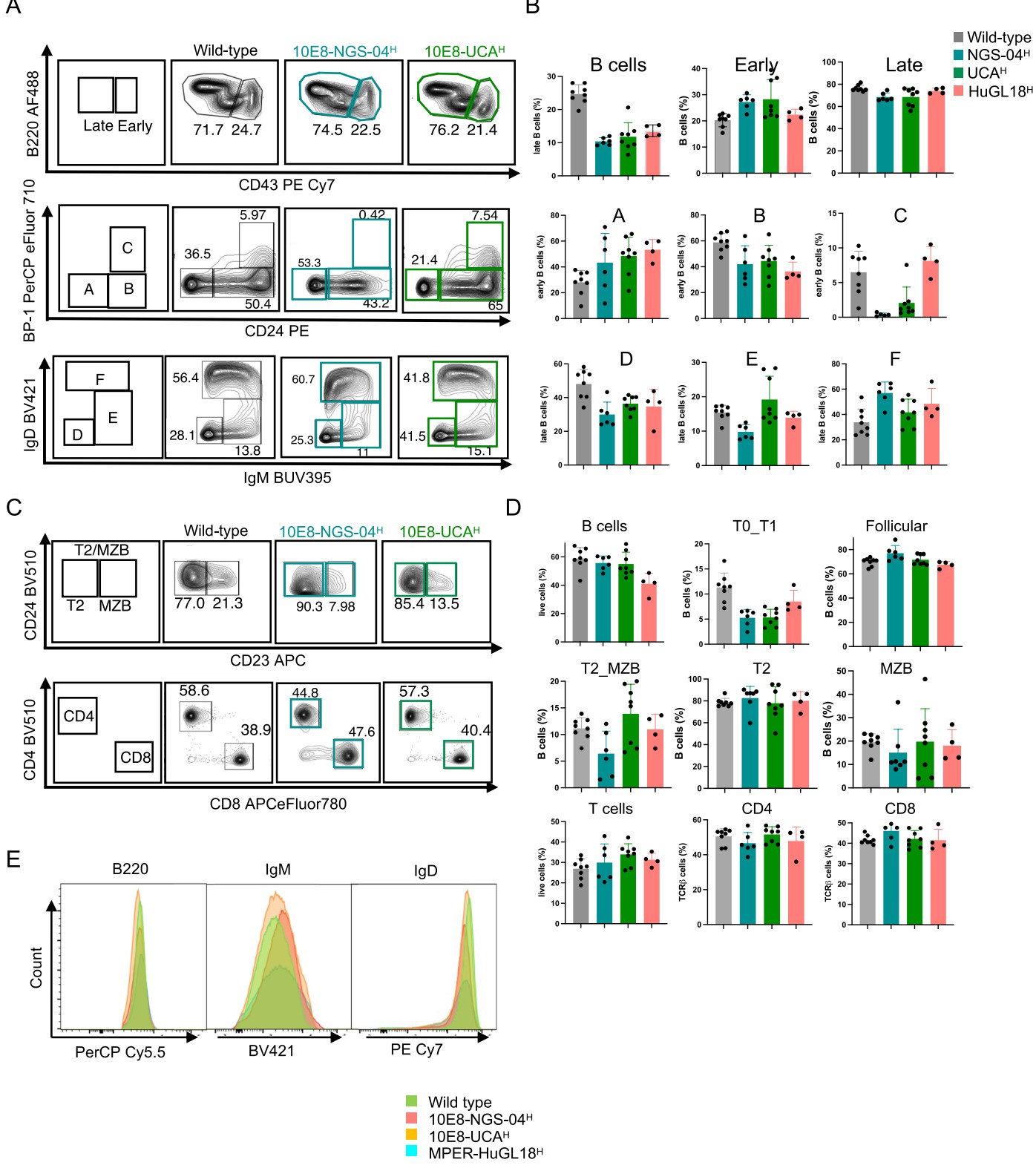

**Extended Data Fig. 1 | See next page for caption.**

**Extended Data Fig. 1 | Generation of preclinical mouse models expressing the heavy chains of NGS-04 and 10E8UCA, related to Fig. 1. a**, Representative flow cytometry of bone marrow progenitors isolated from 8–10-week-old wild-type, 10E8-NGS-04[H] and 10E8-UCA[H] mice separated by Hardy classification into: early (B220[+]CD43[+]) and late (B220[+]CD43[−]) B cells (top panel); early (A–C) B cell subfractions (middle panel); and the late (D–F) subfractions in the bone marrow (bottom panel). Data is representative of two independent experiments (n = 3–4 mice/independent group). **b**, Quantification of flow cytometry of bone marrow progenitors, as in a. Error bars indicate mean ± SD from mice in each group. Data are pooled from two independent experiments (n = 3–4 per treatment) for wild-type, 10E8-NGS-04[H] and 10E8-UCA[H] mice; MPER-HuGL18[H] data presented for comparative purposes (n = 4 mice from one experiment). **c**, Representative flow cytometry of CD4[+]/CD8[+] T cells and T2/MZB (marginal zone B cells) cells isolated from spleen samples of 8–10-week-old wild-type, 10E8-NGS-04[H] and 10E8-UCA[H] mice, as in Fig. 1a,b. **d**, Quantification of T and B cell subfractions, as in c. Error bars indicate mean ± SD from mice in each group. Data are pooled from two independent experiments (n = 3–4 per treatment) for wild-type, 10E8-NGS-04[H] and 10E8-UCA[H] mice, with some values repeated from Fig. 1a, b for comparison; MPER-HuGL18[H] data also presented for comparative purposes (n = 4 mice from one experiment). **e**, Representative flow cytometry histograms showing expression of B220, IgM and IgD in follicular B cells in spleen (green, WT; red, 10E8-NGS-04; yellow, 10E8-UCA and blue, MPER-HuGL18). Data representative of two independent experiments (n = 3–4 mice per treatment) for wild-type, 10E8-NGS-04[H] and 10E8-UCA[H] mice; MPER-HuGL18[H] data presented for comparative purposes (n = 4 mice from one experiment).

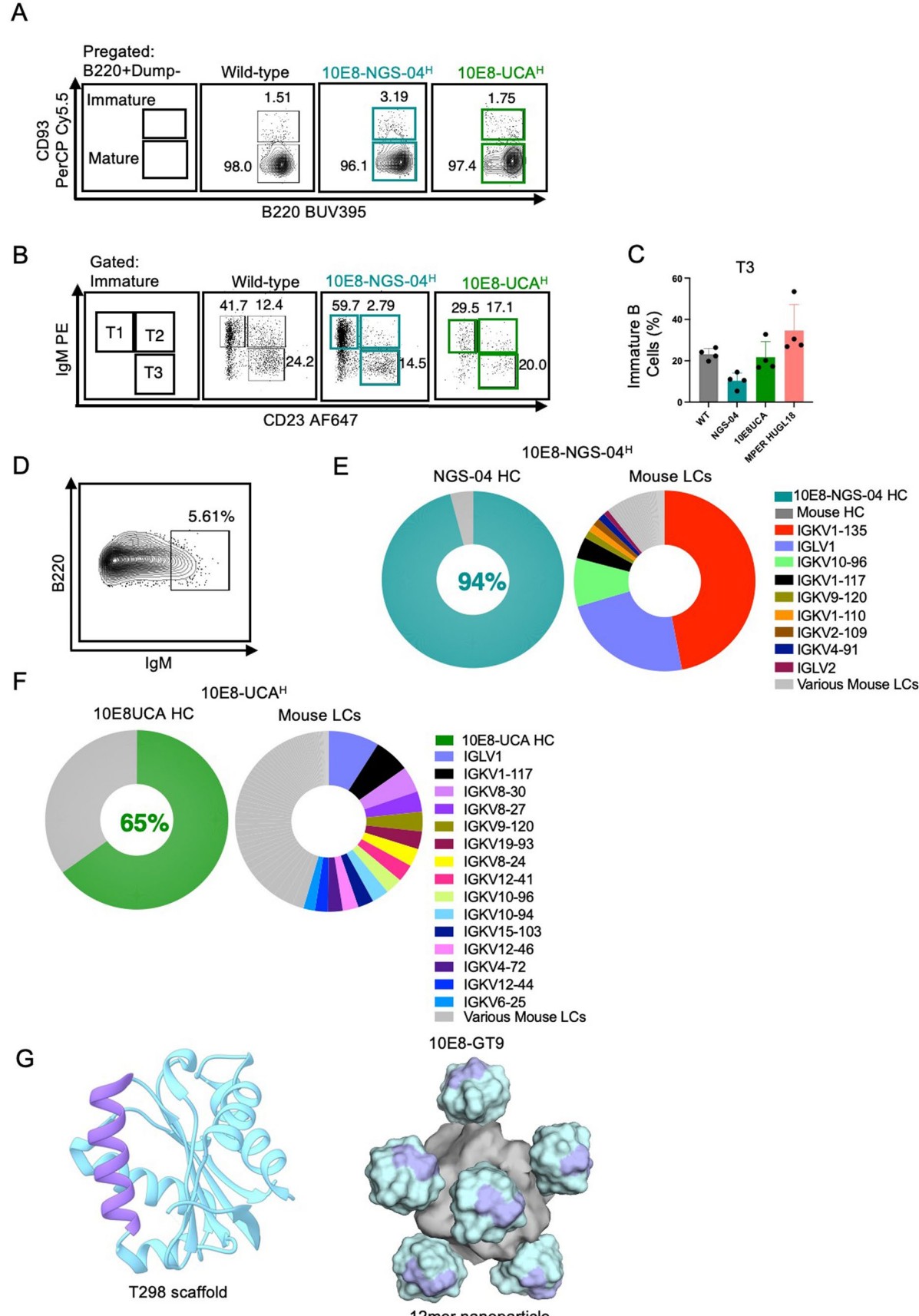

**Extended Data Fig. 2 | See next page for caption.**

**Extended Data Fig. 2 | B cell development in 10E8 mice, related to Fig. 1.**
**a**, Representative flow cytometry of immature (B220$^+$CD93$^+$) and mature (B220$^+$CD93$^-$) B cells isolated from spleen samples of 8–10-week-old wild-type, 10E8-NGS-04$^H$ and 10E8-UCA$^H$. **b**, Representative flow cytometry of T1, T2 and T3 subfractions from immature B cells isolated from spleen samples of wild-type, 10E8-NGS-04$^H$ and 10E8-UCA$^H$. **c**, Quantification of transitional cells subset T3 as in b, with MPER-HuGL18$^H$ shown for comparison. Error bars indicate mean ± SD from mice in each group (n = 4). Data representative of two independent experiments (n = 3 mice per treatment) for wild-type, 10E8-NGS-04$^H$ and 10E8-UCA$^H$ mice with one presented; MPER-HuGL18$^H$ data from one experiment presented for comparative purposes (n = 4 mice from one experiment). **d**, Gating strategy for single-cell sorting of naïve B cells from blood of 10E8-NGS-04$^H$ and 10E8-UCA$^H$ mice. **e, f**, 10x Genomics single-cell BCR sequences from B220$^+$IgM$^+$ naïve B cells sorted from 10E8-NGS-04$^H$ (e) and 10E8-UCA$^H$ (f) mice. Human NGS-04 IGHV gene frequency in teal (94%), human 10E8UCA IGHV gene frequency in green (65%), and grey murine HC. Pies on the right shows the respective murine IGKV genes in various colors; IGKV families used in key. n=pairs amplified, NGS-04 (n = 1046) and 10E8-UCA (n = 1260). **g**, Left, crystal structure of the T298 scaffold (PDB 3T43) that served as a starting point for the iterative vaccine design of 10E8-GT9 and 10E8-GT10 immunogens. Purple: MPER; Cyan: scaffold backbone. Right, schematic illustration of the multimerization of the 10E8-GT immunogens into self-assembling 12mer nanoparticles.

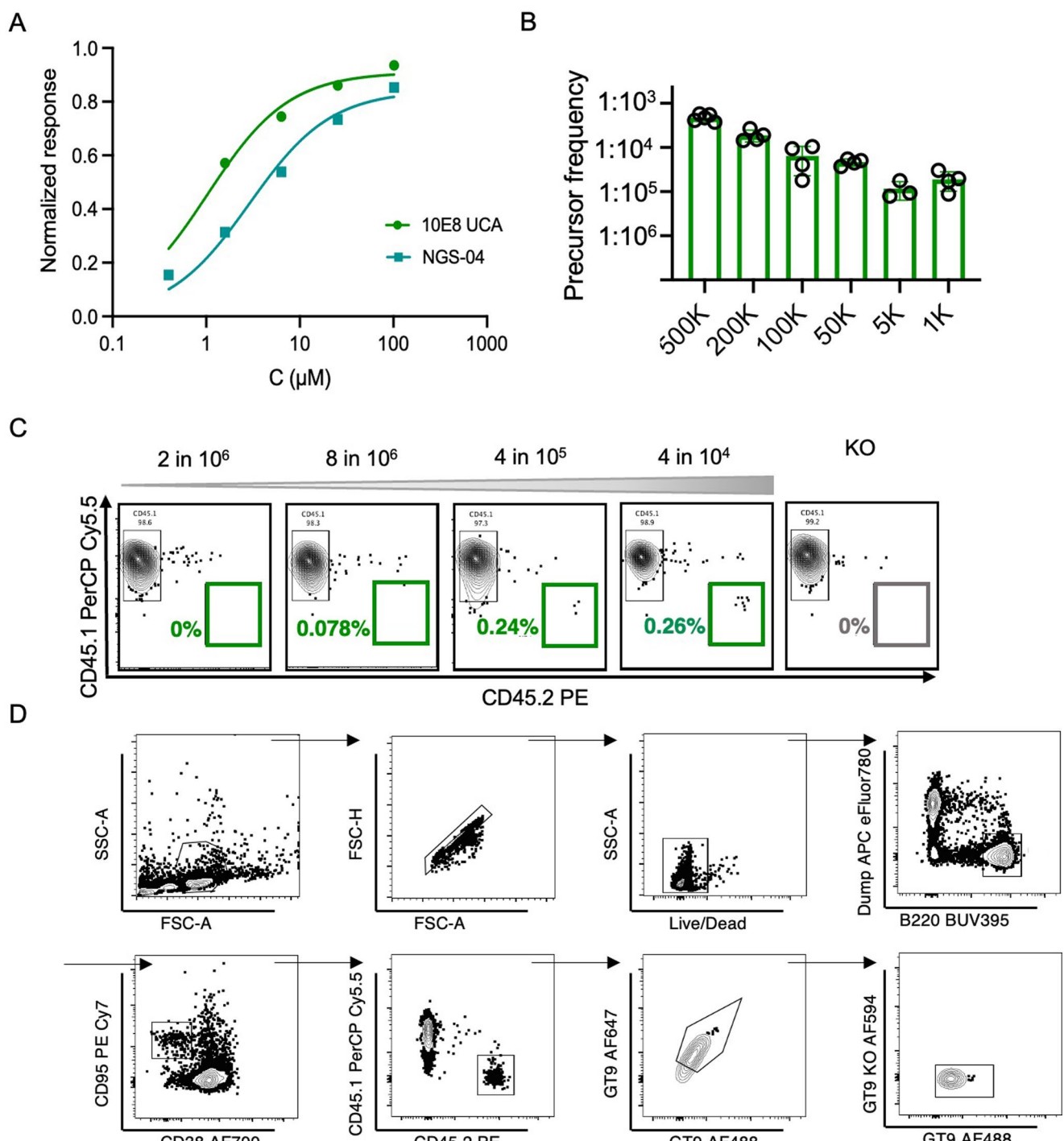

**Extended Data Fig. 3 | 10E8 UCA B cells can be activated *in vivo*, related to Fig. 2. a**, Representative equilibrium binding curves of 10E8-GT9.2 interacting with human 10E8-UCA[H]/UCA[L] and 10E8-NGS-04[H]/10E8-NGS-04[L] as measured by the monomer SPR assay. **b**, Quantification of precursor frequencies corresponding to transfer of 5,000, 50,000, 100,000, 200,000 and 500,000 CD45.2[+]10E8-UCA[H] B cells from donor mice to the host wild-type CD45.1[+] mice.

(n = 4). Error bars indicate mean ± SD from mice in each group. Experiment performed once. **c**, Flow cytometry of CD45.2[+] cells in splenic GCs of mice transferred with varying numbers of CD45.2[+]10E8-UCA[H] precursors and immunized with 50 μg of 10E8-GT9 12mer and the Ribi adjuvant system 7 DPI. **d**, Representative gating strategy used for the GC response in adoptive transfer experiments.

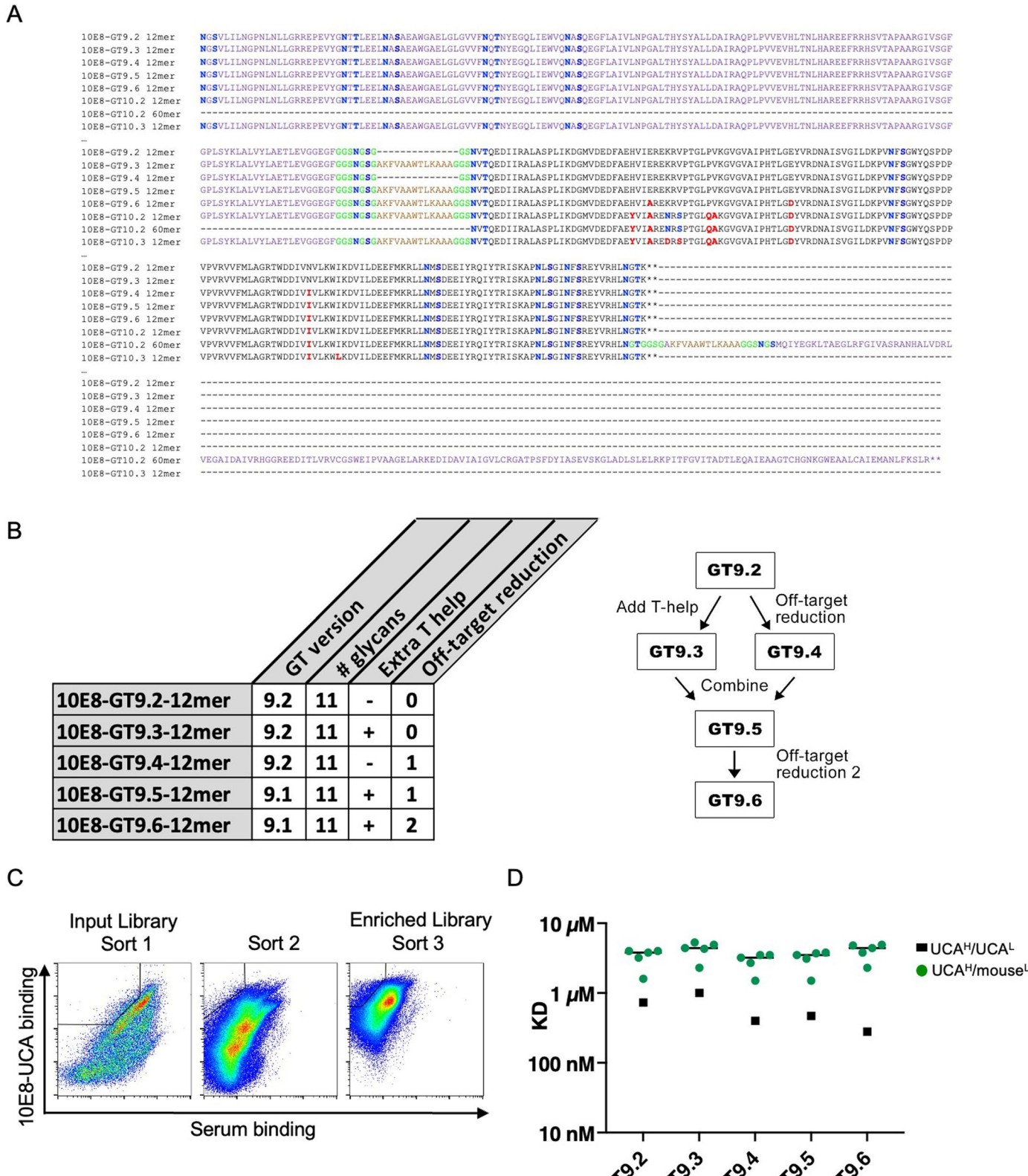

**Extended Data Fig. 4 | New immunogen sequences and binding, related to Fig. 2. a**, Sequences of new immunogens. Mutations relative to 10E8-GT9.2 12mer are red; nanoparticles are purple; the PADRE epitope is in brown; N-linked glycosylation sites are blue; flexible linkers are green. Sequences are wrapped over multiple lines. **b**, Table and schematic describing the 10E8-GT9 nanoparticle design elements including GT version, added glycan sites, presence of PADRE T-help epitope, and off-target reduction. **c**, Representative FACS gating strategy

for 10E8-GT9 libraries designed to reduce off-target responses. A selection strategy was employed to maintain high binding to 10E8-UCA (y-axis) while reducing binding of mouse competitors (x-axis). **d**, Graph showing affinities of human 10E8-UCA[H]/ UCA[L] Fab and Fabs of 10E8-UCA[H]/mouse[L] to 10E8-GT9.2 12mer, 10E8-GT9.3 12mer, 10E8-GT9.4 12mer, 10E8-GT9.5 12mer and 10E8-GT9.6 12mer, using the nanoparticle SPR assay. Symbols represent individual antibodies; lines represent median 10E8-UCA[H]/ mouse[L] values.

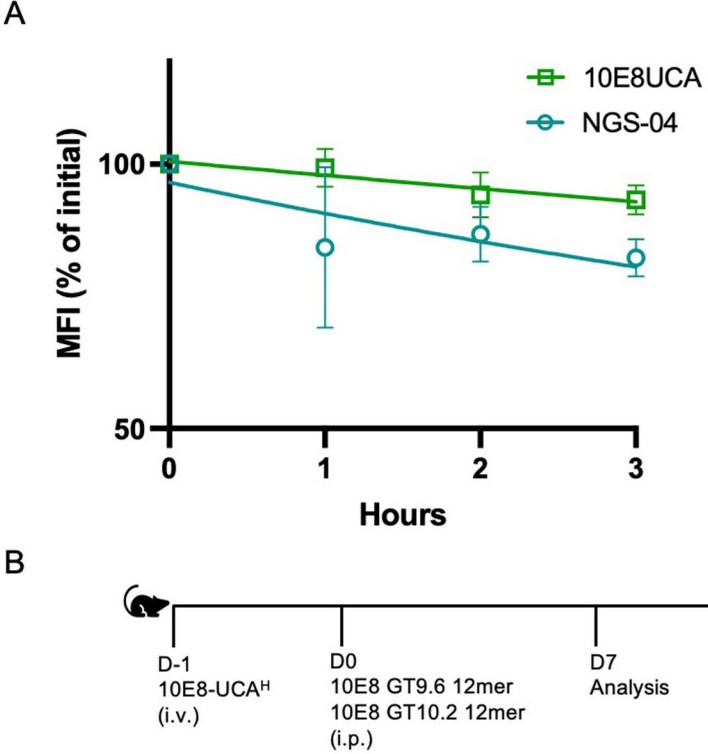

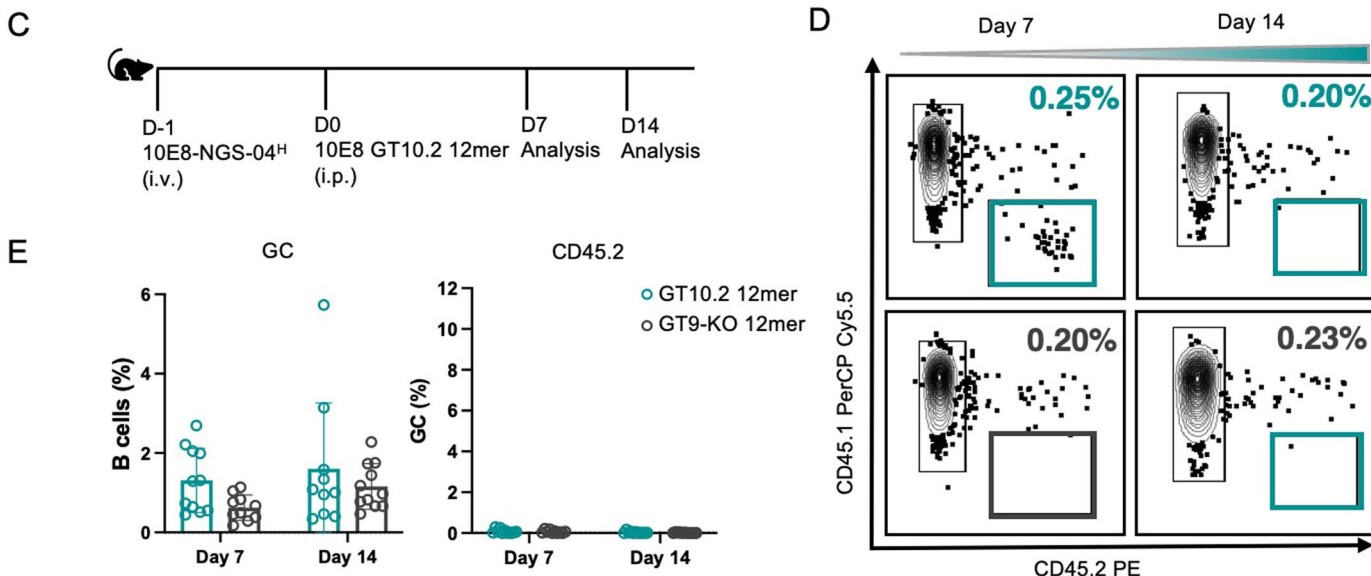

**Extended Data Fig. 5 | NGS-04[H] mice show weak immune responses to 10E8-GT10.2, related to Fig. 3. a**, Graph showing mean fluorescence intensity (MFI, % of initial) of 10E8-GT10.2 dissociation from 10E8-UCA[H] (green) and 10E8-NGS-04[H] BCRs (teal). Error bars indicate SD (n = 4). Experiment performed once. **b**, Schematic showing wild-type mice adoptively transferred with 10E8-NGS-04[H] precursors and immunized with either 50 µg of 10E8-GT9.6 12mer, 10E8-GT10.2 12mer or 10E8-GT9-KO 12mer with ribi adjuvant (control), as in Fig. 3b, c. **c**, Schematic showing wild-type mice adoptively transferred with 10E8-NGS-04[H]

precursors and immunized with either 5 µg 10E8-GT10.2 12mer and alhydrogel or 5 µg 10E8-GT9-KO 12mer control and alhydrogel. **d**, Representative flow cytometric plots of CD45.2[+]10E8-NGS-04[H] B cells recruitment to GCs 7 and 14 DPI in spleens of mice immunized as in c. **e**, Quantification of GC size and CD45.2 in GC in spleen of mice immunized in c,d. Data are pooled from two independent experiments (n = 5–6 per treatment). Error bars indicate mean ± SD from mice in each group.

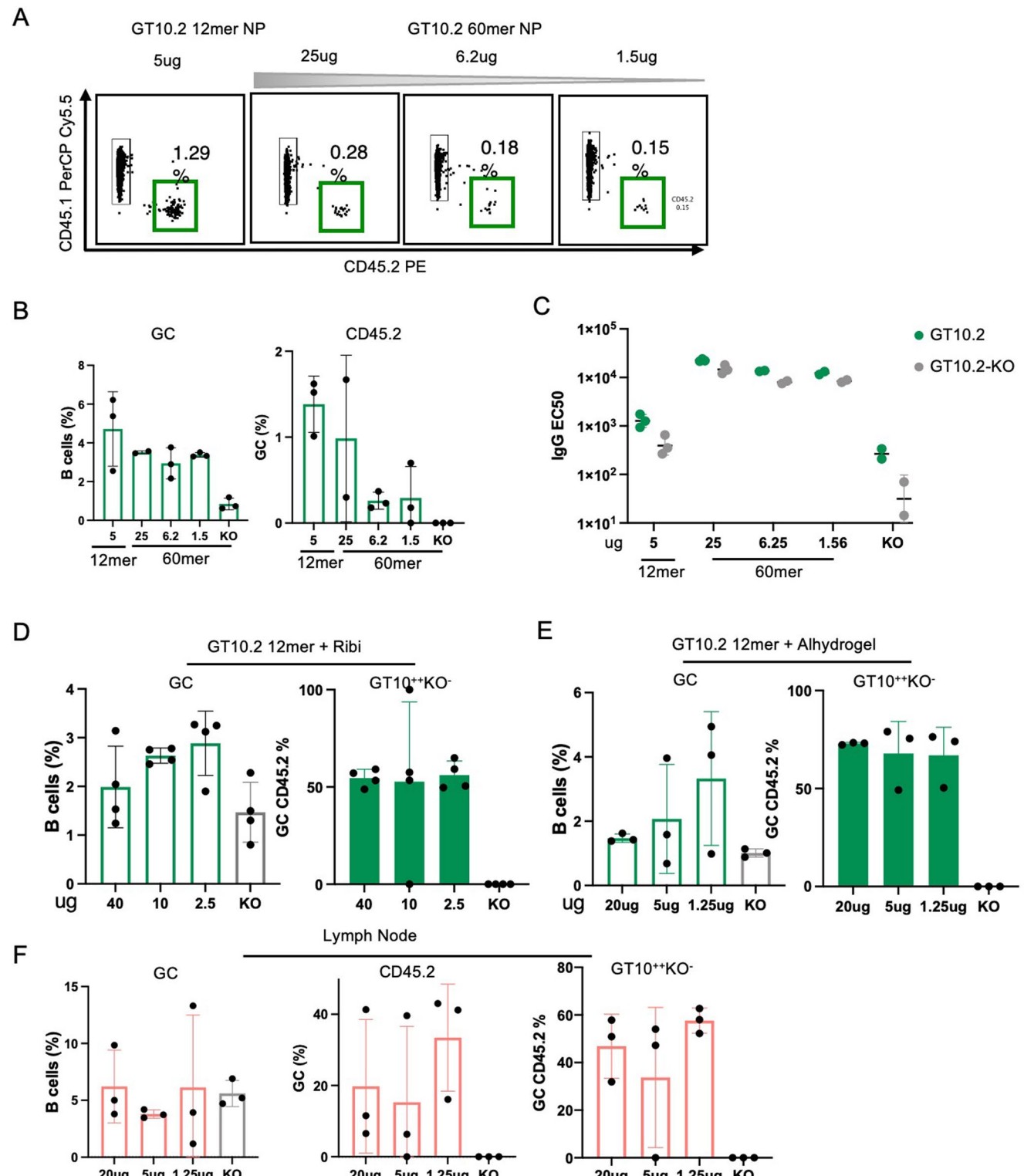

**Extended Data Fig. 6 | See next page for caption.**

**Extended Data Fig. 6 | Antigen multimerization does not impact the GC response, related to Fig. 3. a**, Representative flow cytometry plots of CD45.2$^+$10E8-UCA$^H$ recruitment to GCs 7 DPI in spleens of mice immunized with 5 µg of 10E8-GT10.2 12mer or with 25 µg, 6.2 µg or 1.5 µg of GT10.2 60mer and Ribi adjuvant. **b**, Quantification of a: (Left) GC size; (right) CD45.2$^+$10E8-UCA$^H$ cells in GCs. Error bars indicate mean ± SD from mice in each group (n = 3). Two independent experiments were performed; data from one representative shown. **c**, ELISA of Ab response to immunizations as in a. Green dots represent IgG level specific to 10E8-GT10.2 probe and gray dots represent IgG level to 10E8-GT10-KO probe. Each symbol represents a different mouse. Bars indicate geometric mean and geometric SD from mice in pooled groups. n = 2–3 mice in each group. **d**, Quantification of GC size and epitope specific (GT10$^{++}$KO$^-$) CD45.2$^+$10E8-UCA$^H$ in GC from mice immunized with different doses of 10E8-GT10.2 12mer with Ribi adjuvant, as shown in Fig. 3d. Error bars indicate mean ± SD from mice in each group (n = 4). Two independent experiments performed, one shown. **e**, Quantification of GC size and epitope specific (GT10$^{++}$KO$^-$) CD45.2$^+$10E8-UCA$^H$ in GC from mice immunized with different doses of 10E8-GT10.2 12mer with alhydrogel, as shown in Fig. 3d. Error bars indicate mean ± SD from mice in each group (n = 3). Three independent experiments were performed; data from one representative shown. **f**, Quantification of lymph node response in mice. GC size, CD45.2$^+$10E8-UCA$^H$ B cells in GCs and epitope specific CD45.2$^+$ in GCs from 7 DPI with 10E8-GT10.2 12mer and alhydrogel, from mice immunized as in Fig. 3d. Error bars indicate mean ± SD from mice in each group (n = 3).

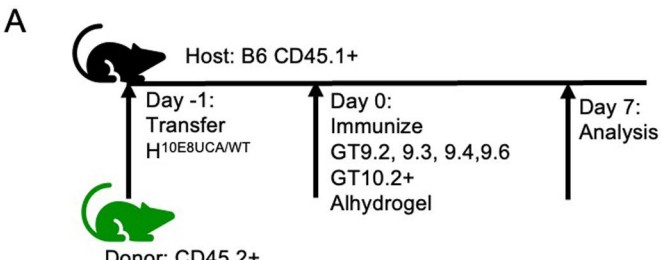

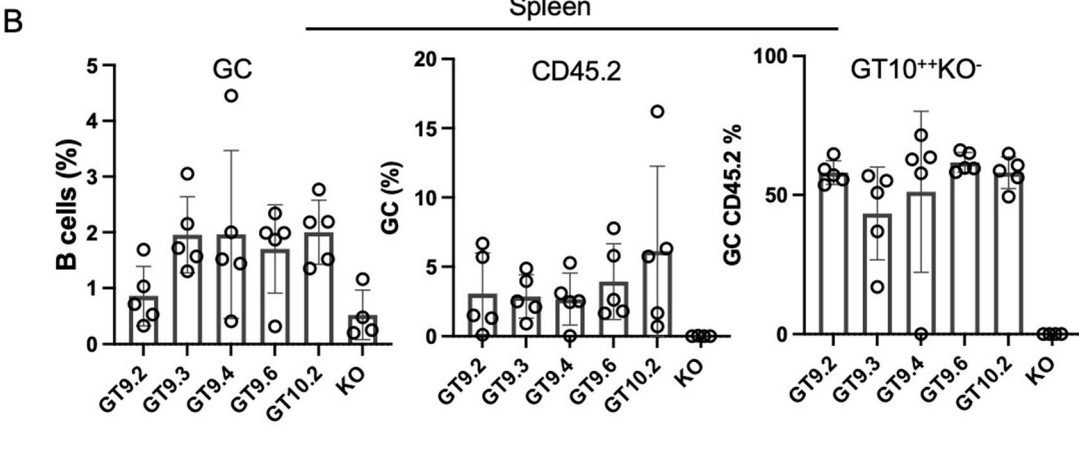

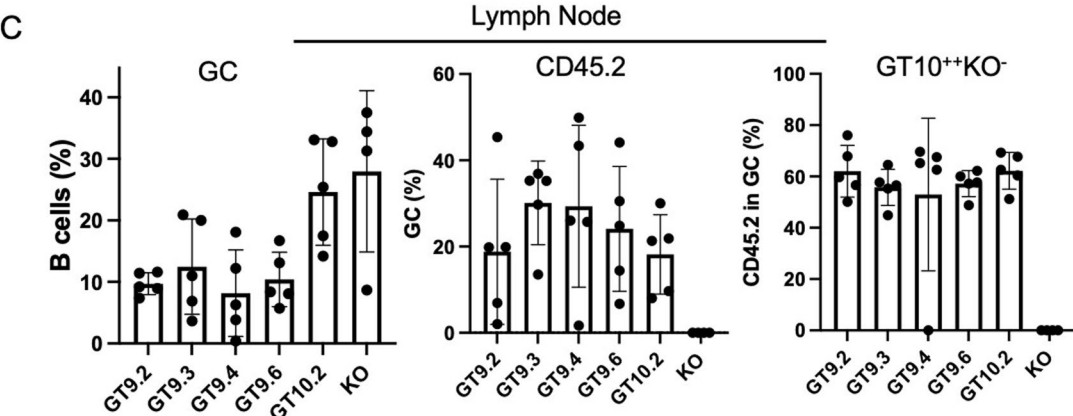

**Extended Data Fig. 7 | Antigens with a minimum affinity can recruit 10E8-UCA$^H$ B cells to germinal centers, related to Fig. 3. a**, Schematic showing mice adoptively transferred to reach ~1:10$^4$ 10E8-UCA$^H$ precursors and immunized with 5 μg either different variants of 10E8-GT9 12mer and GT10.2 12mer with alhydrogel or 10E8-GT9-KO 12mer with alhydrogel (control); experiment performed once. **b**, Quantification of GC size, CD45.2$^+$10E8-UCA$^H$ B cells in GC and epitope specific CD45.2$^+$10E8-UCA$^H$ in GC in spleen of mice immunized as in a. Error bars indicate mean ± SD from mice in each group (n = 5). **c**, Quantification of GC size, CD45.2 in GC and epitope specific CD45.2 in GC in lymph nodes of mice immunized in a. Error bars indicate mean ± SD from mice in each group (n = 5).

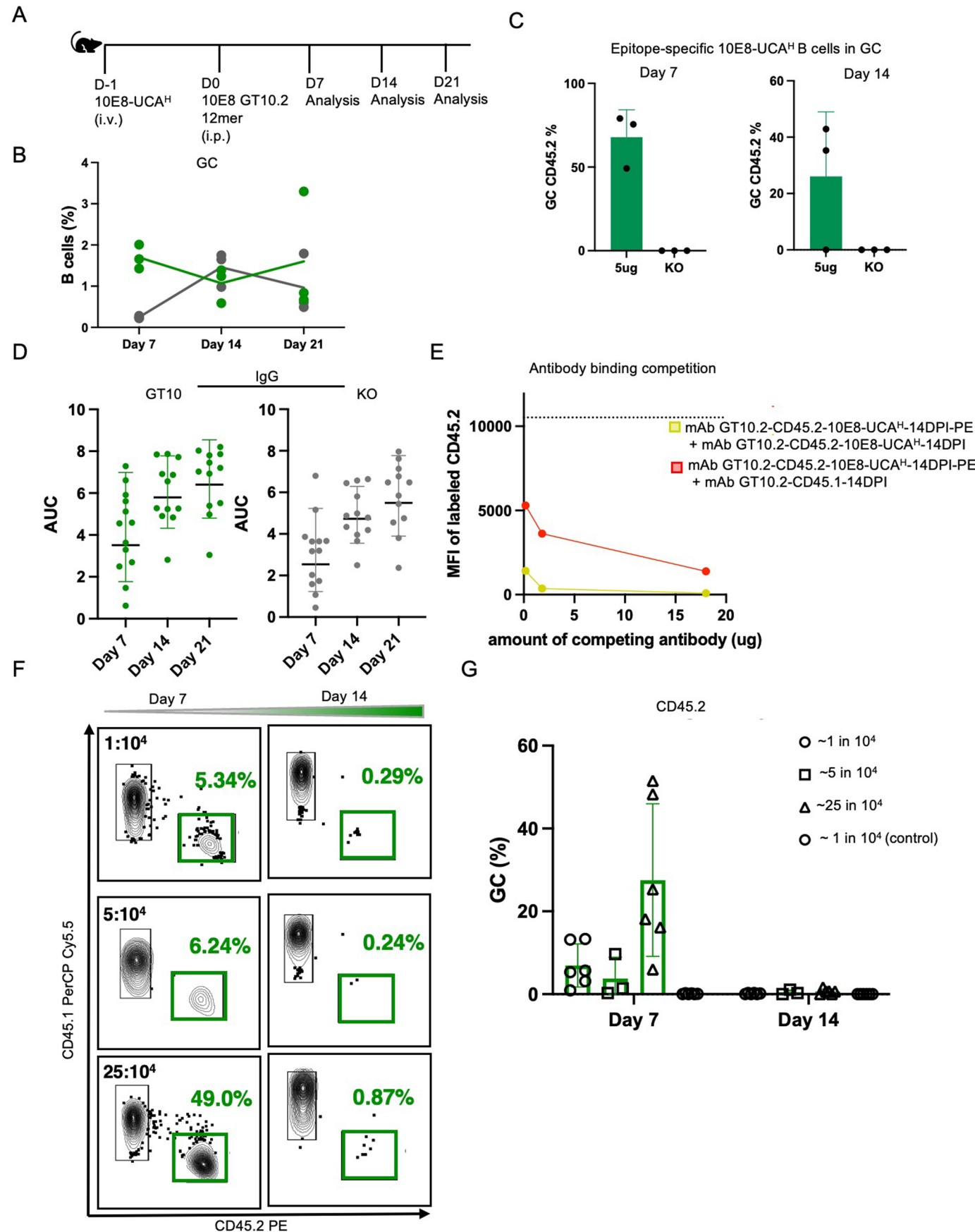

**Extended Data Fig. 8 | See next page for caption.**

**Extended Data Fig. 8 | Epitope-specific 10E8 B cells are outcompeted by endogenous WT B cells in GCs, related to Figs. 4 and 5. a**, Schematic of adoptive transfer of CD45.2$^+$ 10E8-UCA$^H$ B cells into CD45.1$^+$ wild-type mice followed by immunization with 10E8-GT10.2 12mer with alhydrogel and analysis until 21 DPI. **b**, Quantification of percent of B cells in GC of spleen from mice immunized with 10E8-GT10.2 12mer at 7, 14, and 21 DPI, as in a. Green lines show the mean percentage of 10E8-UCA$^H$ GC B cells in mice immunized with 10E8-GT10.2 12mer and gray lines show mean percentage of cells immunized with 10E8-GT9-KO 12mer (n = 3). Data from one representative of three independent experiments shown. **c**, Quantification of epitope specific CD45.2$^+$10E8-UCA$^H$ in mice immunized with 10E8-GT10.2 12mer at different time points, as in a. Error bars indicate mean ± SD from mice in each group (n = 3). Data from one representative of three independent experiments shown. **d**, ELISA of Ab response of 10E8-UCA$^H$ to immunizations with 10E8-GT10.2 12mer, as in a.

Green dots represent IgG titer specific to 10E8-GT10.2 probe and gray dots represent IgG level to KO probe. Data pooled from three independent experiments. Each symbol represents a different mouse. Bars indicate geometric mean and geometric SD from mice in all three groups. n = 4-5 mice in each group. **e**, Graph showing the MFI of mAb-GT10.2-CD45.2 10E8-UCA$^H$-14DPI-PE with varying amounts of competing antibodies (μg) on x-axis (yellow: CD45.2 Ab; red: CD45.1 Ab). Dotted line represents the 'no competition' control. **f**, Representative flow cytometric plots of CD45.2$^+$10E8-UCA$^H$ B cells recruitment to GCs 7 and 14 DPI in spleens of mice transferred to reach frequencies of ~1:10$^4$ 5:10$^4$ and 25:10$^4$ CD45.2$^+$10E8-UCA$^H$ precursors and immunized with 5 μg of 10E8-GT10.2 12mer with alhydrogel or 10E8-GT9-KO 12mer with alhydrogel (control). **g**, Quantification of CD45.2$^+$ in GC in spleen of mice immunized in f. Data are pooled from two independent experiments. Error bars indicate mean ± SD from mice in each group (n = 3 mice/independent group).

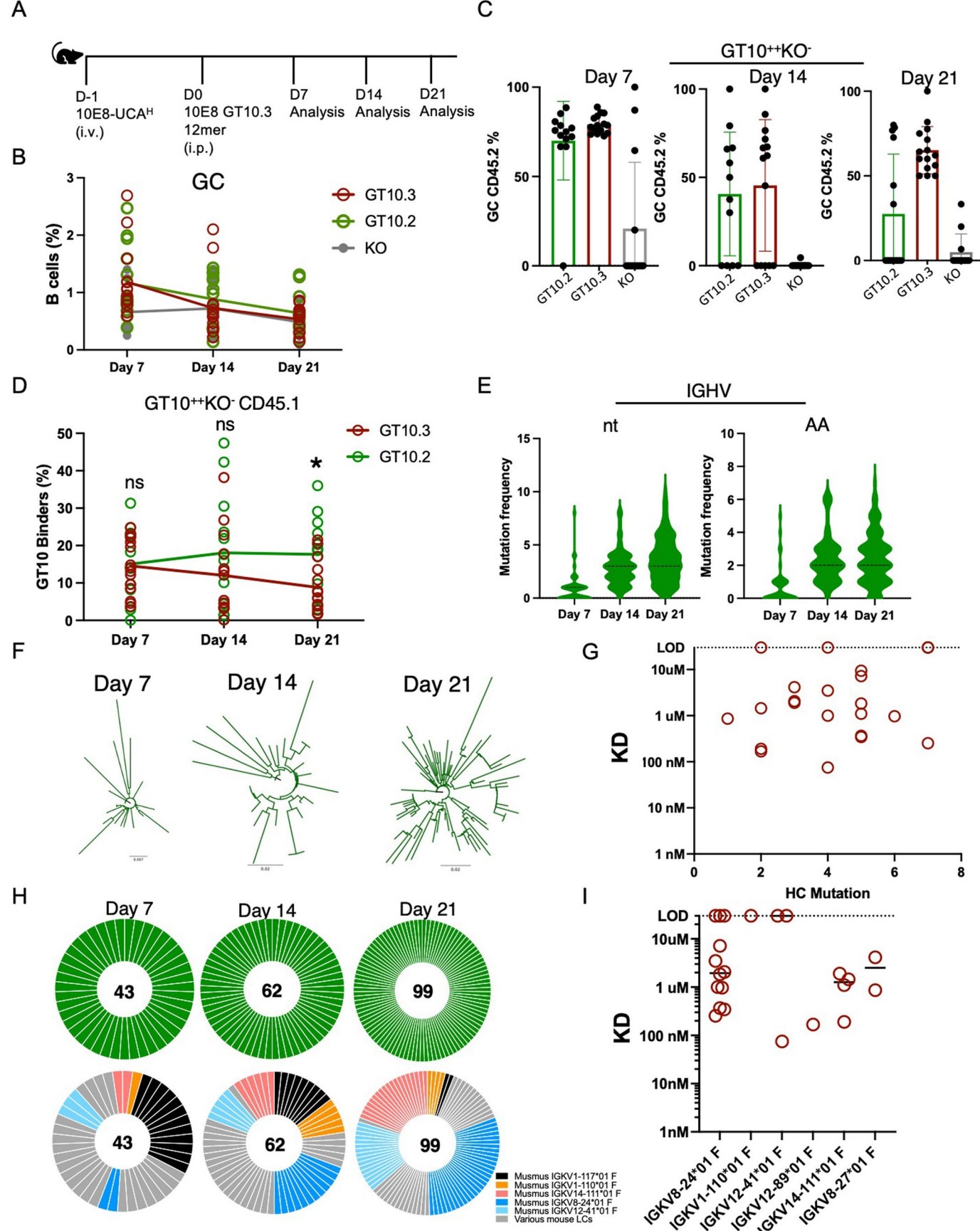

**Extended Data Fig. 9 | See next page for caption.**

**Extended Data Fig. 9 | Higher affinity 10E8-GT10.3 improves GC sustenance of 10E8 B cells, related to Fig. 6. a**, Schematic of CD45.1⁺ mice adoptively transferred to establish a frequency of 1:10⁴ CD45.2⁺10E8-UCA^H B cells and then immunized with either 5 µg of 10E8-GT10.3 12mer, 10E8-GT10.2 12mer, or a control KO immunogen with alhydrogel, as in Fig. 6c. **b**, Quantification of percentage of GC B cells post immunization with 10E8-GT10.3 12mer, 10E8-GT10.2 12mer, or 10E8-GT9-KO 12mer. Data are pooled from two independent experiments (n = 5–8) with lines marking the respective means. **c**, Quantification of epitope specific (GT10⁺⁺KO⁻) CD45.2⁺10E8-UCA^H B cells in GC in mice immunized with 10E8-GT10.2 12mer or 10E8-GT10.3 12mer at different time points, as in a. Error bars indicate mean ± SD from mice in each group (n = 5–8). Data are pooled from two independent experiments. **d**, Quantification of epitope specific CD45.1⁺ cells in GC in mice immunized as in a. Data are pooled

from two independent experiments. *p =0.0137(unpaired t test, two tailed); ns, not significant. N = 5–8 mice in each group; lines mark means. **e**, Mutations in IGHV of GT10-specific 10E8-UCA^H B cells 7, 14, and 21 DPI. (Left) Nucleotide (nt); (right) amino acid (aa). The black dashed line indicates the median number of mutations. **f**, Diversification of IGHV sequences from 10E8-UCA^H B cells isolated at 7, 14, and 21 DPI with 10E8-GT10.3 12mer as in e. **g**, Graph comparing the number of mutations in heavy chain of 10E8-UCA cells (*x*-axis) against their affinity (*y*-axis) (shown in Fig. 6e) 21 DPI with 10E8-GT10.3 12mer. **h**, HCs (top) LCs (bottom) sequenced from GT10-specific 10E8-UCA^H B cells 7, 14, and 21 DPI by 10E8-GT10.3 and alhydrogel, as in a. Numbers inside pie shows the number of cells sequenced. **i**, Graph showing the affinities of the 10E8-UCA^H 21 DPI as per in Fig. 6e with the associated mouse light chains.

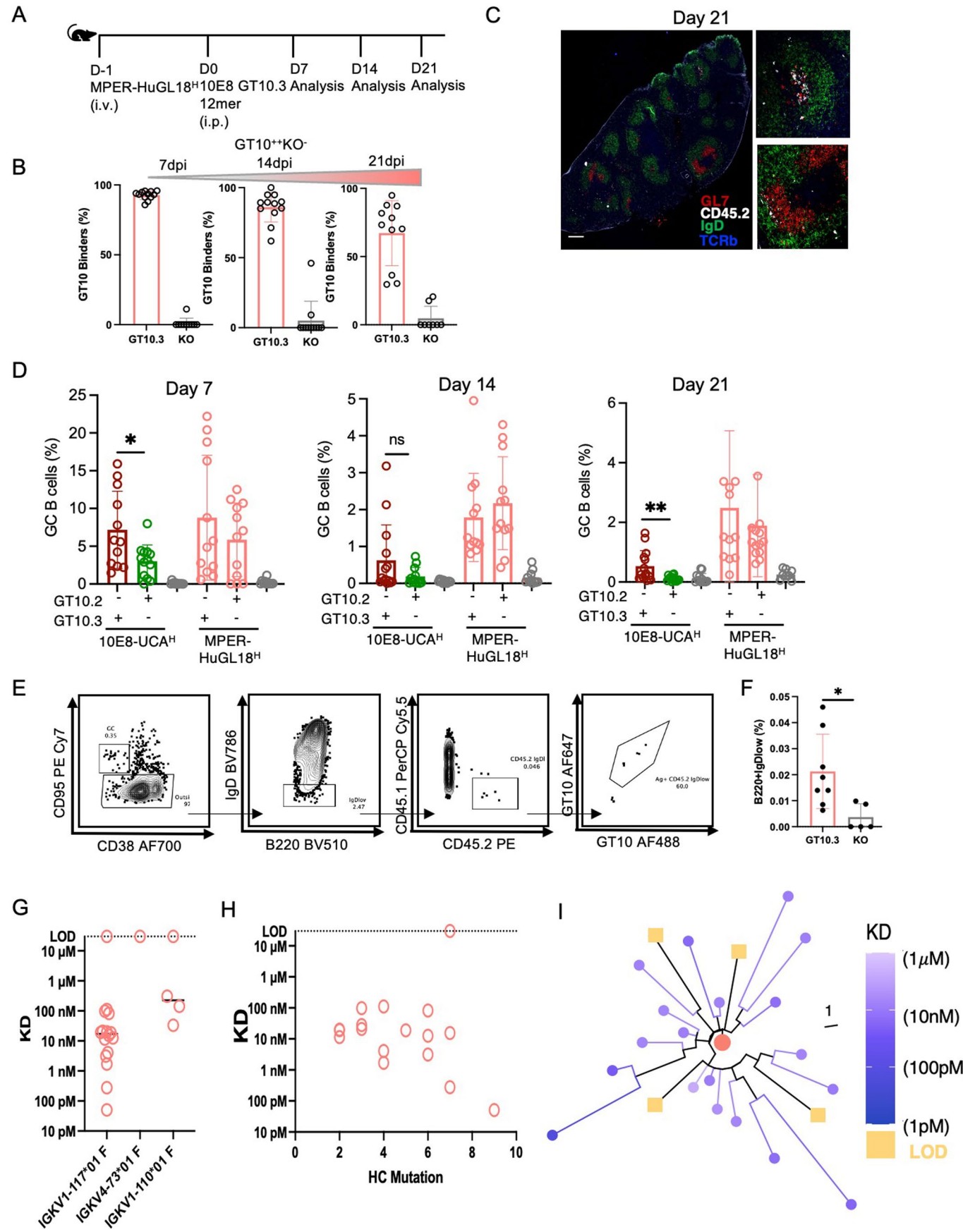

**Extended Data Fig. 10 | See next page for caption.**

**Extended Data Fig. 10 | Mice carrying native human precursor sequence show a sustained GC response and form memory, related to Fig. 7.**
**a**, Schematic of CD45.1[+] mice adoptively transferred to establish a frequency of 1:10[4] CD45.2[+]MPER-HuGL18[H] B cells and then immunized with either 5 μg of 10E8-GT10.3 12mer immunogen and alhydrogel, or 10E8-GT9-KO 12mer (control), as per Fig. 7b. **b**, Quantification of epitope specific CD45.2[+]MPER-HuGL18[H] B cells in mice immunized with 10E8-GT10.3 12mer as in a. Error bars indicate mean ± SD from mice in each group (n = 6–8). Data pooled from two independent experiments. **c**, Immunohistochemistry of spleen at 21 DPI. Red: GL7 (GC marker); White: CD45.2; Green: IgD; Blue: TCRβ. Scale, 250 um. **d**, Quantification showing CD45.2 recruitment in GCs from 10E8-UCA[H] or MPER-HuGL18[H] transferred mice immunized with 10E8-GT10.2 12mer and 10E8-GT10.3 12mer (merged data from Figs. 6d, 7c, and ref. 25). Data are pooled from 2–3 independent experiments (n = 4–6). Error bars indicate mean ± SD. (*p = 0.0131, **p = 0.0085) (unpaired

t test, two-tailed). **e**, Representative flow cytometry showing gating strategy used for the identification of memory B cells (MBC) 36 DPI with 10E8-GT10.3 12mer with alhydrogel (n = 8) or 10E8-GT9-KO 12mer with alhydrogel (n = 5). **f**, Quantification of memory B cells (MBC) 36 DPI with 10E8-GT10.3 12mer, as in e. Experiment performed once. Error bars indicate mean ± SD. (p = 0.024) (unpaired t-test, two-tailed). **g**, Graph of the affinities of MPER-HuGL18[H] mAbs 21 DPI with 10E8-GT10.3 12mer (panel with affinities) (y-axis) sorted by light chain (x-axis) LOD: limit of detection. **h**, Graph showing the affinities (y-axis) of the subset using IGKV1-117 light chains from the MPER-HuGL18[H] 21 DPI with 10E8-GT10.3 12mer (selection from Fig. 7g) vs. number of heavy chain mutations (x-axis). LOD: limit of detection. **i**, Tree showing phylogenetic relationship between MPER-HuGL18[H] B cell IGHV sequences, colorized on the basis of 21 DPI affinities shown in (Fig. 7g).

# Reporting Summary

## Statistics

For all statistical analyses, confirm that the following items are present in the figure legend, table legend, main text, or Methods section.

| n/a | Confirmed | |
|---|---|---|
| ☐ | ☒ | The exact sample size (*n*) for each experimental group/condition, given as a discrete number and unit of measurement |
| ☐ | ☒ | A statement on whether measurements were taken from distinct samples or whether the same sample was measured repeatedly |
| ☐ | ☒ | The statistical test(s) used AND whether they are one- or two-sided<br>*Only common tests should be described solely by name; describe more complex techniques in the Methods section.* |
| ☐ | ☒ | A description of all covariates tested |
| ☐ | ☒ | A description of any assumptions or corrections, such as tests of normality and adjustment for multiple comparisons |
| ☐ | ☒ | A full description of the statistical parameters including central tendency (e.g. means) or other basic estimates (e.g. regression coefficient) AND variation (e.g. standard deviation) or associated estimates of uncertainty (e.g. confidence intervals) |
| ☐ | ☒ | For null hypothesis testing, the test statistic (e.g. *F*, *t*, *r*) with confidence intervals, effect sizes, degrees of freedom and *P* value noted<br>*Give P values as exact values whenever suitable.* |
| ☒ | ☐ | For Bayesian analysis, information on the choice of priors and Markov chain Monte Carlo settings |
| ☒ | ☐ | For hierarchical and complex designs, identification of the appropriate level for tests and full reporting of outcomes |
| ☒ | ☐ | Estimates of effect sizes (e.g. Cohen's *d*, Pearson's *r*), indicating how they were calculated |

*Our web collection on statistics for biologists contains articles on many of the points above.*

## Software and code

Policy information about availability of computer code

| Data collection | BD FACSDiva™ Software, Biotek Gen5 |
|---|---|
| Data analysis | FlowJo, Prism 9.5.1, Geneious Prime, FLASh, MiXCR, Microsoft Office |

For manuscripts utilizing custom algorithms or software that are central to the research but not yet described in published literature, software must be made available to editors and reviewers. We strongly encourage code deposition in a community repository (e.g. GitHub). See the Nature Portfolio guidelines for submitting code & software for further information.

## Data

Policy information about availability of data

All manuscripts must include a data availability statement. This statement should provide the following information, where applicable:
- Accession codes, unique identifiers, or web links for publicly available datasets
- A description of any restrictions on data availability
- For clinical datasets or third party data, please ensure that the statement adheres to our policy

BCR sequences at PP617372–PP617659 and PP617660–PP618659 and crystal structures at 9BDH & 9BDI. No original code is reported in this manuscript. Any additional information required to reanalyze the data reported in this paper is available from the lead contact (F.D.B.) upon request.

# Research involving human participants, their data, or biological material

Policy information about studies with human participants or human data. See also policy information about sex, gender (identity/presentation), and sexual orientation and race, ethnicity and racism.

| | |
|---|---|
| Reporting on sex and gender | No human participants, data, or biological material are included. |
| Reporting on race, ethnicity, or other socially relevant groupings | No human participants, data, or biological material are included. |
| Population characteristics | No human participants, data, or biological material are included. |
| Recruitment | No human participants, data, or biological material are included. |
| Ethics oversight | No human participants, data, or biological material are included. |

Note that full information on the approval of the study protocol must also be provided in the manuscript.

# Field-specific reporting

Please select the one below that is the best fit for your research. If you are not sure, read the appropriate sections before making your selection.

☒ Life sciences    ☐ Behavioural & social sciences    ☐ Ecological, evolutionary & environmental sciences

For a reference copy of the document with all sections, see nature.com/documents/nr-reporting-summary-flat.pdf

# Life sciences study design

All studies must disclose on these points even when the disclosure is negative.

| | |
|---|---|
| Sample size | All experimental groups and numbers are specified in the figures legends; sample sizes were determined based on approaches used in the literature for similar immunogens/models, as described in the Methods. |
| Data exclusions | No data was excluded. |
| Replication | All major assays were repeated at least once (as noted in legends) to verify their reproducibility and were successful. |
| Randomization | Mice for experimental and control groups derive from the same colonies and were randomly assigned to groups. |
| Blinding | Blinding was not employed, as the first author's involvment from mouse development, maintenance, injection, and analysis rendered analytical blinding infeasible. |

# Reporting for specific materials, systems and methods

We require information from authors about some types of materials, experimental systems and methods used in many studies. Here, indicate whether each material, system or method listed is relevant to your study. If you are not sure if a list item applies to your research, read the appropriate section before selecting a response.

## Materials & experimental systems

| n/a | Involved in the study |
|---|---|
| ☐ | ☒ Antibodies |
| ☒ | ☐ Eukaryotic cell lines |
| ☒ | ☐ Palaeontology and archaeology |
| ☐ | ☒ Animals and other organisms |
| ☒ | ☐ Clinical data |
| ☒ | ☐ Dual use research of concern |
| ☒ | ☐ Plants |

## Methods

| n/a | Involved in the study |
|---|---|
| ☒ | ☐ ChIP-seq |
| ☐ | ☒ Flow cytometry |
| ☒ | ☐ MRI-based neuroimaging |

## Antibodies

| | |
|---|---|
| Antibodies used | All antibodies used in this study have been listed with their catalog numbers and supplier in the Reagents and Tools Table S3 in the manuscript; dilutions are described in the Methods section. |

| Validation | All antibodies deployed here have been previously used effectively in similarly-generated KI mice (e.g., Melzi et al., Immunity 2022; Wang et al., EMBO J 2021; Kratochvil et al., Immunity 2021; Tas et al., Immunity 2022). |
|---|---|

# Animals and other research organisms

Policy information about studies involving animals; ARRIVE guidelines recommended for reporting animal research, and Sex and Gender in Research

| Laboratory animals | B6.SJL-Ptprc a Pepc b /BoyJ mice (CD45.1+/+) 8–12 weeks of age and KI mice described in text housed in a facility at ambient 68F/40% humidity on a 12:12 light cycle. |
|---|---|
| Wild animals | No wild animals were used. |
| Reporting on sex | Both male and female donor mice were used for adoptive transfers. All transfers and immunizations were done in male CD45.1+/+ mice due to risk of rejection of cells from male donor to female recipient mice. |
| Field-collected samples | No field collections were performed. |
| Ethics oversight | All experiments were performed under the approval by the AALAC-accredited Institutional Animal Care and Use Committee (IACUC) of Harvard University and the Massachusetts General Hospital (MGH) (Animal Study Protocols 2016N000022 and 2016N000286). |

Note that full information on the approval of the study protocol must also be provided in the manuscript.

# Flow Cytometry

## Plots

Confirm that:

☒ The axis labels state the marker and fluorochrome used (e.g. CD4-FITC).

☒ The axis scales are clearly visible. Include numbers along axes only for bottom left plot of group (a 'group' is an analysis of identical markers).

☒ All plots are contour plots with outliers or pseudocolor plots.

☒ A numerical value for number of cells or percentage (with statistics) is provided.

## Methodology

| Sample preparation | At selected time points following immunization, whole spleens were mechanically dissociated to generate single-cell suspensions. ACK lysis buffer was used to remove red blood cells and splenocytes were then resuspended in FACS buffer (2% FBS/PBS), Fc-blocked (clone 2.4G2, BD Biosciences) and stained for viability with Live/Dead Blue (Thermo Fisher Scientific) for 20 min at 4⬚C. For surface staining GT9 or GT10 probes (described above), as well as antibodies against CD4-APCeF780, CD8-APC-eF780, Gr-1-APC-eF780, F4/80-APC-eF780, B220-B510, CD95-PE-Cy7, CD38-A700, CD45.1-PerCPCy5.5, CD45.2-PE, IgD-BV786, IgM-BUV395 and IgG1-BV421, were used. |
|---|---|
| Instrument | Cells were acquired by a BD LSRFortessa (BD Biosciences) for flow cytometric analysis and sorted using a BD FACS Aria II instrument (BD Biosciences). |
| Software | BD FACSDiva™ Software used for data collection and analysis. FlowJo used for data analysis. |
| Cell population abundance | Live dead staining was used to confirm sample quality. Population gates described in the text were then used to characterize the abundance of each cell fraction. |
| Gating strategy | The gating strategies are S3D, wherein the gating strategy shows identification of GC B cells (CD95hiCD38lo) (pregated on B220+Live/Dump-) and the proportion of CD45.1+ and CD45.2+ B cells in GCs followed by the percentage of epitope specific cells (GT9++KO-), the same gating has been used for all germinal center analysis, and S10E which shows gating strategy identification of memory B cells as CD38+CD95-IgD- B220+CD45.2+GT10++ B cells. |

☒ Tick this box to confirm that a figure exemplifying the gating strategy is provided in the Supplementary Information.

