## [Peer Review File · Nature Immunology]

Peer Review Information

Journal: Nature Immunology

Manuscript Title: Affinity gaps among B cells in germinal centers drive the selection of MPER precursors

Corresponding author name(s): Dr Facundo Batista; Dr William Schief

Reviewer Comments & Decisions:

Decision Letter, initial version:
--

19th Sep 2023

Dear Dr. Batista,

Thank you for your response to the referees' comments on your Article, "Elicitation of MPER-targeting antibodies in humanized mouse models reveals the importance of the precursor-competitor affinity gap in germinal centers". While the work is of potential interest, the reviewers have raised substantial concerns that must be addressed. As such, we cannot accept the current manuscript for publication, but would be interested in considering a revised version that addresses these serious concerns, as long as novelty is not compromised in the interim.

Please revise the manuscript to address all issues raised by the referees and according to your response. We believe it is essential to strengthen your conclusions on the role of affinity in GC entry and maintenance with new experimental data. In addition, please revise the manuscript to better emphasize the implications of your results for understanding general GC biology. At resubmission, please include a point-by-point "Response to referees" detailing how you have addressed each referee comment (please specify page and figure number where the new data can be found in the revised manuscript). This response will be sent back to the referees along with the revised manuscript.

In addition, please include a revised version of any required reporting checklist. It will be available to referees (and, potentially, statisticians) to aid in their evaluation if the manuscript goes back for peer review. A revised checklist is essential for re-review of the paper. The Reporting Summary can be found here:

When submitting the revised version of your manuscript, please pay close attention to our [href="https://www.nature.com/nature-portfolio/editorial-policies/image-integrity">Digital Image Integrity Guidelines](https://www.nature.com/nature-portfolio/editorial-policies/image-integrity). and to the following points below:

[REDACTED]

We hope to receive a suitably revised manuscript within 6 months. If you cannot send it within this time, please let us know. We will be happy to consider your revision so long as nothing similar has been accepted for publication at Nature Immunology or published elsewhere.

Nature Immunology is committed to improving transparency in authorship. As part of our efforts in this direction, we are now requesting that all authors identified as 'corresponding author' on published papers create and link their Open Researcher and Contributor Identifier (ORCID) with their account on the Manuscript Tracking System (MTS), prior to acceptance. ORCID helps the scientific community achieve unambiguous attribution of all scholarly contributions. You can create and link your ORCID from the home page of the MTS by clicking on 'Modify my Springer Nature account'. For more information please visit please visit www.springernature.com/orcid.

Thank you for the opportunity to review your work.

Sincerely,

Ioana Visan, Ph.D.
Senior Editor
Nature Immunology

Tel: 212-726-9207
Fax: 212-696-9752
www.nature.com/ni

Reviewers' Comments:

Reviewer #1:

Remarks to the Author:

In this ms, Ray et al set out to investigate how starting affinity and precursor frequency impact recruitment of MPER-binding B cells into GC, as well as their affinity maturation and maintenance. They first generate two preclinical mouse models expressing human 10E8-precursor heavy chains and validate them showing KI B cells are specific to a 10E8 germline targeting immunogen. They go on to show that 10E8 UCA B cells can be transferred into WT mice and recruited into GCs. Optimization of antigenic formulation reveals that the combination of low dose GT10.2 12mer with alum elicits a stronger GC response. Nevertheless, 10E8 UCA B cells are outcompeted by endogenous WT and fail to undergo affinity maturation. Addition of mutations to the 10E8-GT10.2 (designated 10E8-GT10.3) that increase the affinity of precursor KI B cells by 3-fold appears to lead to a somewhat more sustained response that still fails to affinity mature. Lastly, they use a different KI mouse (MPER HuGL18) carrying human germline sequences that bind to 10E8-GT9.2 with the highest affinity (80nM). Transfer of B cells from these mice into WT mice leads to a more sustained response that does affinity mature, in a manner comparable or superior to endogenous competitor cells, leading the authors to conclude that increased affinity for the antigen increases both the frequency of precursors that are recruited to GC and their maintenance. These findings, if true, constitute important advances in our understanding of how to elicit broadly neutralising Abs for HIV vaccine design. The main concern is the robustness of some of the findings which can be strengthened as described below.

Main concerns

1. The B cell phenotype of HNGS-04/WT mice appears quite different from H10E8UCA/WT mice. Mature B cells appear to express lower IgD and IgM, as well as B220. Can the authors compare the IgD, IgM and B220 MFI of mature follicular B cells between the different models and enumerate T3 B cells (only T1-T2 are enumerated). They could also add a CD93 stain to establish if the cells are generally more immature, anergic etc. These findings are interesting because they may suggest self-reactivity of this heavy chain, however, do not impact their findings since 10E8UCA B cells, which look quite comparable to the WT are used throughout the ms.

2. A key part of the analysis is the comparison between the recruitment and maintenance in GCs between GT10.3 and GT10.2 immunizations. Can the authors use statistics in Fig. 6E, to show if there is a significant difference in the maintenance of GC B cells between the two groups?

It would be important to clarify whether the frequency of CD45.2 GC B cells at the peak (day 7, which likely reflect recruitment) is the same as it appears in Fig. 6E, or different. The lines should on day 7 should start at the median values. There is a single outlier in the GT10.2 group at day 7, yet still this one value in which the line starts would not correspond to either mean or median – please check if this is an error. For this group, the median looks closer to 0.5 than 8. In the GT10.3 group, the median for day 7 would be ~12 rather than 8. This is important because it would emphasize that the main difference between the groups is not their ability to be maintained per se after the peak of the response, but rather the starting number of precursors recruited and proportion of cells at the peak of the response, which would indeed appear to be determined by the starting affinity, as the authors rightly state. Increasing the numbers of day 7 mice in this experiment may add robustness to the conclusions.

3. The authors then introduce HMPERHuGL18/WT mice as a model in which the starting KI B cells

have even higher affinity. Fig. 7H suggests the average of affinity of expressed antibodies from inferred germline sequences is $\sim 180\text{nM}$. Although the B cell maturation in this model is said to be normal and submitted as part of a different ms, IgM vs IgD, B220 vs CD93 plots and enumeration of T3 cells should be shown side-by-side to the H10E8UCA/WT B cells in case the phenotype is more like the HNGS-04/WT.

4. In this latest model, there is substantial variability in the % of cells recruited and peaking on day 7 (range of 0-24% of CD45.2 in GCs). Nevertheless, the median number is ≈ 8 , like the previous two models. If this is the case, the impact of affinity on recruitment to GCs is less clear. Can the authors comment on this once they confirm the peak CD45.2 day 7 numbers in Fig. 6E?

5. It is unclear if the rate of acquisition of somatic mutations differs between the three models, with all of them showing a median of 2-2.5 on day 14, a time point in which there has been affinity maturation when HMPERHuGL18/WT are transferred, but not when H10E8UCA/WT cells are transferred. Can the authors comment on this? Is it about the quality of the mutations rather than rate of mutation?

6. Again, a comparative analysis between the decline of CD45.2 cells in GCs between the three models (ie Fig. 7E and Fig. 6E) is important to show if there is a different propensity to survive relative to peak numbers, and if there is, if it is dependent or not on peak GC numbers.

7. The model in which there is affinity maturation and an increased fraction of KI B cells surviving through day 21 is the HMPERHuGL18/WT model, and the difference is mainly attributed to starting B cell affinity. The difference in affinity according to expressed antibodies from inferred germline sequences between HMPERHuGL18/WT ($\sim 180\text{nM}$ according to Fig. 7I) and H10E8UCA/WT ($\sim 400\text{nM}$ according to Fig. 6H) B cells immunised with GT10.3 appears to be only 2-fold. Can the authors comment if they consider this a sufficient difference in affinity to lead to the increased competitive advantage? Can there be structure-dependent factors or differing modes of epitope binding that may compound this effect?

Reviewer #2:

Remarks to the Author:

Ray

The work describes the behaviour of B cells bearing either of three germline targeted VH rearrangements derived from or related to human antibodies that are specific to the MPER region of HIV, a highly desirable target for the development of broadly neutralizing antibodies against HIV. B cell development was shown to be essentially normal, with good representation of the targeted IgH genes among the mature B cell repertoire in the different B cell compartments. These results, essentially Figures 1 and 2, are essentially background or foundation information and are uncontroversial.

The key findings of the work, relate to the response of the mice to immunization with variants of an antigen containing the MPER determinant against which the IgH gene is specific. A series of immunization strategies and variants are used to show that the outcome is not predictable in that specificity and even relatively high affinity binding of the BCR does not guarantee continuation within

the resultant germinal centers (GC). Various processes of antigen and immunization improvement are applied to increase targeted B cell involvement in the GC and participation in affinity maturation towards the desired epitope. Ultimately this is found to be successful with one of the three targeted mice in conjunction with one of the several variants of antigen plus adjuvant used.

The protein engineering is excellent, no question. The skill with which the group identifies, creates and improves the target antigen and even the responding B cells for the desired outcome is at a very, very high standard. But I was left decidedly ambiguous on the question of whether the work advanced the discipline of immunology. The processes being followed here are well described by this group and others in the field of HIV vaccine development (and appropriately cited in this work), so that is not new. This left me thinking the most important thing would be to develop an new understanding of why some of the protocols 'failed' in GC engagement and activity while others succeeded. I think the team attempted to develop explanations of why, but these were largely superficial, expected and not well investigated, certainly not enough to extend the understanding of why some B cell persist in GC and others do not, despite those B cells having several apparent advantages. I also thought there were instances where an investigation was left inconclusive on one paragraph, but later used as a fact, assigning certainty where that was probably not appropriate.

This is excellent quality work but as it currently stands, it does not sufficiently advance understanding of GC activity and clonal selection and retention to warrant publication in Nature Immunology. The novelty resides largely in the antigen and the targeted B cells and how they behave, and that is currently quite descriptive.

Minor points:

1. The number of single B cells used for assessment of targeted VH gene usage is very small (13 and 22). What is the statistical certainty of the attributed values of 70 and 77% and what is the measure of Vk usage from such a small sample?
2. Where are the data showing expression of the KI sequences among 'most' B cell populations (lines 162-164)? Which ones are absent?
3. Focussing on UCA rather than NGS-04 (lines 201-202) based on affinity of the whole original Abs is curious, as only the IgH chain is in the mice and now with a variety of IgK. Could the affinity of the serum Ab in the mice be a better measure of starting affinity? Even then, there is a difference between the surface of a B cell and a soluble IgG/IgM.
4. A major observation of the work is the displacement of the targeted B cells in GC by endogenous B cells over time. This is investigated sequencing both to assess SHM and subsequently affinity of expressed recombinant Abs (line 368) and crystal structure in binding to Ag (line 383). While endogenous B cells increased from undetected at d0 to 20nM by d7 (not d0 as stated line 376) compared to 580nM for targeted B cells. But this is not necessarily an explanation as affinity of soluble Ab need not reflect what happens to deposited Ag on FDC in the GC. The partial overlap from crystal structure is also incompletely investigated – was it a functional interference or not?
5. Line 400-401 is a point of restating an inference as a conclusion, saying affinity difference was the driver of loss of targeted B cells. Likely but not certain.
6. How do the SHM frequency and L chain usage explain persistence of the 10E8 UCAH B cells in GC (lines 426-436)? Correlation not causation.
7. Indeed, the data in lines 447-450 indicate that affinity alone is not the answer.
8. Again SHM analysis on persisting B cells does not explain their persistence in and of itself (line 487).
9. While I agree with the observation of starting affinity and persistence, the 'rule' that can be derived

form this is unclear. This remains to me as an elegant set of observations with relevance to this system.

10. And, why do they authors think that the MPER HuGL18h B cells show more substantial gains in affinity for the same number of mutations? Could this be the basis for a more generalizable conclusion?

11. The conclusions listed in lines 568-570 are completely true and important but indicate the specialized nature of the conclusions that were reached.

Author Rebuttal to Initial comments

See inserted PDF

NI-A36284 Response to Reviewers

Reviewer #1

(Remarks to the Author)

In this ms, Ray et al set out to investigate how starting affinity and precursor frequency impact recruitment of MPER-binding B cells into GC, as well as their affinity maturation and maintenance. They first generate two preclinical mouse models expressing human 10E8-precursor heavy chains and validate them showing KI B cells are specific to a 10E8 germline targeting immunogen. They go on to show that 10E8 UCA B cells can be transferred into WT mice and recruited into GCs. Optimization of antigenic formulation reveals that the combination of low dose GT10.2 12mer with alum elicits a stronger GC response. Nevertheless, 10E8 UCA B cells are outcompeted by endogenous WT and fail to undergo affinity maturation. Addition of mutations to the 10E8-GT10.2 (designated 10E8-GT10.3) that increase the affinity of precursor KI B cells by 3-fold appears to lead to a somewhat more sustained response that still fails to affinity mature. Lastly, they use a different KI mouse (MPER HuGL18) carrying human germline sequences that bind to 10E8-GT9.2 with the highest affinity (80nM). Transfer of B cells from these mice into WT mice leads to a more sustained response that does affinity mature, in a manner comparable or superior to endogenous competitor cells, leading the authors to conclude that increased affinity for the antigen increases both the frequency of precursors that are recruited to GC and their maintenance. These findings, if true, constitute important advances in our understanding of how to elicit broadly neutralising Abs for HIV vaccine design. The main concern is the robustness of some of the findings which can be strengthened as described below.

Response: We sincerely appreciate Reviewer 1's recognition of the importance of our findings to the HIV vaccine field. We furthermore appreciate their suggestions for improving our manuscript; below, we have done our best to respond to each of their requests.

Main concerns

R1.1. The B cell phenotype of HNGS-04/WT mice appears quite different from H10E8UCA/WT mice. Mature B cells appear to express lower IgD and IgM, as well as B220. Can the authors compare the IgD, IgM and B220 MFI of mature follicular B cells between the different models and enumerate T3 B cells (only T1-T2 are enumerated). They could also add a CD93 stain to establish if the cells are generally more immature, anergic etc. These findings are interesting because they may suggest self-reactivity of this heavy chain, however, do not impact their findings since 10E8UCA B cells, which look quite comparable to the WT are used throughout the ms.

R1.1 Response: We thank the reviewer for their interest in our model system, and their point that more thorough cross-model comparisons throughout would be enlightening is well-taken, and so we have generated new data and analyses in response to these concerns. We have produced new CD93 staining and T3 enumeration data, and it has now been added to **Supplementary Figure 2A–C**, and all observed variations are discussed in the text (lines 141–143). We have furthermore created a new analysis comparing the IgD, IgM and B220 mean fluorescence intensity (MFI) of the follicular B cells between the three models (NGS-04, 10E8UCA and MPER HuGL18), and we have added this data as **Supplementary Figure 1E**; we can confirm that the MFI is overall comparable to WT. In addition, we have shown immune responses in NGS-04/WT mice to 10E8-GT10.2 as **Supplementary Figure 5B–D**, demonstrating that NGS-04 can be activated and recruited to germinal centers, although those responses are weaker than those seen in 10E8UCA. This confirms that these cells are not self-reactive or anergic, as they can be specifically activated and recruited to germinal centers in response to the 10E8-GT10.2 immunogen.

R1.2. A key part of the analysis is the comparison between the recruitment and maintenance in GCs between GT10.3 and GT10.2 immunizations. Can the authors use statistics in Fig. 6E, to show if there is a significant difference in the maintenance of GC B cells between the two groups?

R1.2 Response (part 1): We agree with the reviewer that this is key to the paper, and we have added the statistical significance to panel **6E**. We have also reformatted the panel to show the difference between GT10.2 and GT10.3 more clearly.

It would be important to clarify whether the frequency of CD45.2 GC B cells at the peak (day 7, which likely reflect recruitment) is the same as it appears in Fig. 6E, or different. The lines should on day 7 should start at the median values. There is a single outlier in the GT10.2 group at day 7, yet still this one value in which the line starts would not correspond to either mean or median – please check if this is an error. For this group, the median looks closer to 0.5 than 8. In the GT10.3 group, the median for day 7 would be ~12 rather than 8. This is important because it would emphasize that the main difference between the groups is not their ability to be maintained per se after the peak of the response, but rather the starting number of precursors recruited and proportion of cells at the peak of the response, which would indeed appear to be determined by the starting affinity, as the authors rightly state. Increasing the numbers of day 7 mice in this experiment may add robustness to the conclusions.

R1.2 Response (part 2): This is an extremely helpful point, and the reviewer's comments regarding the importance of this timepoint are well-taken. As mentioned above, we have reformatted **Fig. 6E** for clarity to better show the difference in response to GT10.2 and GT10.3; the apparent outlier and value noted by the reviewer was not an error, but was potentially misleading due to the prior presentation. As in other similar quantifications, we present the means for these analyses, which were 7.17 after GT10.3 immunization, and 3.01 after GT10.2.

Furthermore, we thank the reviewer for their questions regarding initial recruitment, which inspired us to perform additional experiments to approach this question from another angle. to determine whether the starting numbers and proportion at peak are the primary driver of sustenance

in GCs, we performed additional experiments using 5-fold and 25-fold higher initially transferred precursors. We found that, when we transfer 25-fold more precursors, although the early response is substantially increased, it does not lead to better sustenance of 10E8 UCA cells: the response diminished by day 14 and became similar to that observed after the lower precursor transfer (**Supplementary Figure 8E–F**), which strengthens the evidence for a difference in the ability of 10E8UCA cells to be maintained in GCs independent of initial precursor numbers recruited.

R1.3. The authors then introduce HMPERHuGL18/WT mice as a model in which the starting KI B cells have even higher affinity. Fig. 7H suggests the average of affinity of expressed antibodies from inferred germline sequences is ~180nM. Although the B cell maturation in this model is said to be normal and submitted as part of a different ms, IgM vs IgD, B220 vs CD93 plots and enumeration of T3 cells should be shown side-by-side to the H10E8UCA/WT B cells in case the phenotype is more like the HNGS-04/WT.

R1.3 Response: We agree with the reviewer that it is important to show the model information in this manuscript. As mentioned above, we have generated new data from MPER HuGL18, which is now presented in **Supplementary Figures 1 and 2** alongside NGS-04 and 10E8UCA; this includes the requested side-by-side IgM vs IgD, B220 vs CD93 plots and enumeration of T3 cells.

R1.4. In this latest model, there is substantial variability in the % of cells recruited and peaking on day 7 (range of 0-24% of CD45.2 in GCs). Nevertheless, the median number is ≤ 8 , like the previous two models. If this is the case, the impact of affinity on recruitment to GCs is less clear. Can the authors comment on this once they confirm the peak CD45.2 day 7 numbers in Fig. 6E?

R1.4 Response: The reviewer's question is an important one, and we have tried to answer it through both a clearer comparison in **Supplemental Figure 10** and new experiments. The mean response to GT10.3 and GT10.2 in 10E8UCA is 7.17 and 3.01 at day 7. Means for GT10.3 and GT10.2 in MPER HuGL18 mice are, respectively, 8.78 and 5.85 on day 7; the divergence becomes far more substantial at days 14 and 21 (**Supplementary Figure 10C**). We suggest, therefore, that the increase in affinity has importance to initial recruitment, as indicated in the direct competition between the NGS-04, 10E8UCA and MPERHuGL18 in a single host prior to immunization; a new set of experiments we have added to **Figure 8** and discuss further below. However, as the high adoptive transfer experiments with 10E8 UCA^H above also suggest (**Supplementary Figure 8**), sustenance may depend far more heavily on each lineage's evolutionary trajectory in the GC after recruitment; we have made an effort to disentangle this question through a careful rewrite of the **Results** and the **Discussion**.

R1.5. It is unclear if the rate of acquisition of somatic mutations differs between the three models, with all of them showing a median of 2-2.5 on day 14, a time point in which there has been affinity maturation when HMPERHuGL18/WT are transferred, but not when H10E8UCA/WT cells are transferred. Can the authors comment on this? Is it about the quality of the mutations rather than rate of mutation?

R1.5 Response: This comment, and several by Reviewer 2 below, raises important points about the relationship between somatic hypermutation and affinity gain. Based on additional analyses

(Presented in **Figure 6I** and **Supplementary Figure 9F** for 10E8 UCA, and **Supplementary Figure 10G and H** for MPER HuGL18), we believe that indeed it is the quality of mutation that is more important than the rate of mutation, and we have rewritten the results throughout to clarify this point and avoid assuming a direct relationship between SHM quantity and affinity gain.

R1.6. Again, a comparative analysis between the decline of CD45.2 cells in GCs between the three models (ie Fig. 7E and Fig. 6E) is important to show if there is a different propensity to survive relative to peak numbers, and if there is, if it is dependent or not on peak GC numbers.

R1.6 Response: We do agree that this would be a useful way to present the data, and we thank the reviewer for suggesting it. We have added a side-by-side comparative analysis of the CD45.2 response to GT10.2 and GT10.3 in both 10E8UCA and MPER HuGL18 to **Supplementary Figure 10C**. While we do see that the initial recruitment after GT10.3 in MPER HuGL18 is variable, the mean in that line is 8.78, slightly higher than the 7.17 seen in 10E8UCA. However, as shown through additional experiments using higher precursor transfer (**Supplementary Figure 8E–F**), we have confirmed that the survival of these cells is not solely dependent on peak GC numbers, but rather on the gain in affinity in GC that occurs in MPER HuGL18 cells but not in 10E8UCA cells. We thank the reviewer for these important clarifying questions, and we have edited the text throughout these sections to be more precise in our distinctions.

R1.7. The model in which there is affinity maturation and an increased fraction of KI B cells surviving through day 21 is the HMPERHuGL18/WT model, and the difference is mainly attributed to starting B cell affinity. The difference in affinity according to expressed antibodies from inferred germline sequences between HMPERHuGL18/WT (~180nM according to Fig. 7I) and H10E8UCA/WT (~400nM according to Fig. 6H) B cells immunised with GT10.3 appears to be only 2-fold. Can the authors comment if they consider this a sufficient difference in affinity to lead to the increased competitive advantage? Can there be structure-dependent factors or differing modes of epitope binding that may compound this effect?

R1.7 Response: The reviewer makes a strong point, and as described above, we have now performed a direct competition experiment (**Figure 8**) wherein all three 10E8 knockin B cell precursors are adoptively transferred into the same recipient and immunized with 10E8-GT10.2, which was highly successful in NHPs in the co-submitted Schiffner et al. article and is in consideration for clinical trials. It is clear that there is a competitive advantage for MPER HuGL18 cells during initial recruitment but primarily in terms of their proliferation and selection within the GC, as demonstrated by the higher fraction of MPER HuGL18 cells in the GC dark zone (**Fig. 8E**) and by the overall affinity gains observed at day 21 (**Figure 7I**). We suspect this may be a variation in the likelihood that the IGHV genes used by 10E8 UCA and MPER HuGL18 undergo affinity-enhancing mutations, though this is speculative. We have made an effort to refocus the **Discussion** in paragraph three on gain in affinity over time rather than initial affinity.

We would like to again thank Reviewer 1 for their thoughtful and constructive recommendations, as well as their recognition of this work's potential importance to both basic immunology and HIV research.

Reviewer #2

(Remarks to the Author)

Ray

The work describes the behaviour of B cells bearing either of three germline targeted VH rearrangements derived from or related to human antibodies that are specific to the MPER region of HIV, a highly desirable target for the development of broadly neutralizing antibodies against HIV. B cell development was shown to be essentially normal, with good representation of the targeted IgH genes among the mature B cell repertoire in the different B cell compartments. These results, essentially Figures 1 and 2, are essentially background or foundation information and are uncontroversial.

Response: We thank the reviewer for their thorough discussion of our findings, but we must express a point of disagreement: the results in Figures 1 and 2 represent a genuine step forward from the perspective of MPER bnAb elicitation, as autoreactivity has previously inhibited the development of preclinical models for this entire class. We apologize for failing to appropriately convey this in the text, and we have now emphasized this in the abstract and relevant results section.

The key findings of the work, relate to the response of the mice to immunization with variants of an antigen containing the MPER determinant against which the IgH gene is specific. A series of immunization strategies and variants are used to show that the outcome is not predictable in that specificity and even relatively high affinity binding of the BCR does not guarantee continuation within the resultant germinal centers (GC). Various processes of antigen and immunization improvement are applied to increase targeted B cell involvement in the GC and participation in affinity maturation towards the desired epitope. Ultimately this is found to be successful with one of the three targeted mice in conjunction with one of the several variants of antigen plus adjuvant used.

The protein engineering is excellent, no question. The skill with which the group identifies, creates and improves the target antigen and even the responding B cells for the desired outcome is at a very, very high standard. But I was left decidedly ambiguous on the question of whether the work advanced the discipline of immunology. The processes being followed here are well described by this group and others in the field of HIV vaccine development (and appropriately cited in this work), so that is not new. This left me thinking the most important thing would be to develop an new understanding of why some of the protocols 'failed' in GC engagement and activity while others succeeded. I think the team attempted to develop explanations of why, but these were largely superficial, expected and not well investigated, certainly not enough to extend the understanding of why some B cell persist in GC and others do not, despite those B cells having several apparent advantages. I also thought there were instances where an investigation was left inconclusive on one paragraph, but later used as a fact, assigning certainty where that was probably not appropriate.

This is excellent quality work but as it currently stands, it does not sufficiently advance understanding of GC activity and clonal selection and retention to warrant publication in Nature Immunology. The novelty resides largely in the antigen and the targeted B cells and how they behave, and that is currently quite descriptive.

Response: We sincerely appreciate the reviewer's recognition of the quality of our work. However, we respectfully disagree with the reviewer's point of view that the manuscript is solely descriptive. On the contrary, the experimental work in this manuscript—generating multiple novel mouse models, studying the germinal center response against multiple antigens at a single-cell resolution, cloning BCRs for antibody expression, measuring the affinities of these antibodies and resolving the crystal structures—is all focused on increasing our mechanistic understanding of antigen competition by antibodies. We believe that this approach represents an unparalleled effort to understand the nature of competition in the GCs; furthermore, it was carried out not with traditional model antigens (e.g., HEL, NP), but rather with antigens of potential clinical relevance. Thus, these conclusions inform not only the basic biology but also translation to the clinic. However, we have nonetheless endeavored to meet the high bar this reviewer has set through the additional experiments we have performed in the interim, which include direct in vivo competition experiments and in vitro and in vivo examinations of endogenous antibody blocking; we discuss these new experiments further below.

Minor points:

R2.1. The number of single B cells used for assessment of targeted VH gene usage is very small (13 and 22). What is the statistical certainty of the attributed values of 70 and 77% and what is the measure of Vk usage from such a small sample?

R2.1 Response: We agree with the reviewer that the number of cells in the initial analysis was insufficient for the conclusions we drew regarding Vk usage. We have revised these figures by analyzing additional cells, and now report n>1000 sequences acquired through a new 10x sequencing pipeline for each KI mouse characterization in **Figure 1E&H**, and **Supplementary Figure 2E–F**.

R2.2. Where are the data showing expression of the KI sequences among 'most' B cell populations (lines 162-164)? Which ones are absent?

R2.2 Response: This was infelicitous phrasing on our part; we intended to refer to the **Figure 1C** sequencing, (now expanded as described in the response above), to indicate that most assayed B cells expressed the KI HC, rather than to imply the existence of subpopulations in which the KI is absent. We appreciate the reviewer's correction and have rewritten the section in question to state, "Further, the KI sequences were expressed by the majority of the peripheral B cells assayed..." (line 181).

R2.3. Focussing on UCA rather than NGS-04 (lines 201-202) based on affinity of the whole original Abs is curious, as only the IgH chain is in the mice and now with a variety of IgK. Could the affinity of the serum Ab in the mice be a better measure of starting affinity? Even then, there is a difference between the surface of a B cell and a soluble IgG/IgM.

R2.3 response: The reviewer is correct: inferring the affinity from the heavy chain alone is inadequate. Based on our new 10x data from NGS-04, we selected 9 constructs to express antibodies to measure these affinities directly but were unfortunately unable to express these Fabs; this work is ongoing, and we will add it if possible. We therefore followed the reviewer’s suggestion of using the total sera, developing a new bead assay to characterize the binding in the sera of NGS-04^H mice and compared its binding to 10E8UCA^H and MPER HuGL18^H (**Reviewer Figure 1**).

While this confirmed that the sera of all three lines bind to 10E8-GT10.2, we could not estimate affinity due to the presence of both IgG and IgM. However, the level of binding for NGS-04^H does seem to be higher than for 10E8 UCA^H: we suspect this is due to the higher allelic exclusion in NGS-04^H (**Supplementary Figure 2E**) rather than reflecting a difference in affinity. We therefore attempted to measure the dissociation off rate, but were also unable to, likely due to the confounding presence of IgM in the serum. We therefore developed an assay for measuring BCR binding and dissociation to 10E8-GT10.2 probe by FACS. PBMCs containing naïve B cells from 10E8 UCA and NGS-04 mice were surface stained by 10E8-GT10.2 probes conjugated with SA-BV421. After final washing, stained cells were incubated at room temperature for 1, 2, or 3 hours to allow dissociation prior to FACS analysis. The observed dissociation rate of NGS-04 was faster than for 10E8 UCA, indicating that the affinity of NGS-04 is lower than 10E8UCA, and has been incorporated into the manuscript (**Supplementary Figure 5A**).

Furthermore, we also performed new immunization experiments in NGS-04^H KI mice (**Supplementary Figure 5B–D**). From our results we confirm that these cells could be activated *in vivo* and recruited to germinal centers (day 7), but that activation was extremely poor. This suggests that BCR affinity is likely lower than the activation floor, in line with our expectations from the H+L antibody measurement in humans reported in the co-submitted Schiffner et al. manuscript and in our **Supplemental Figure 3A**.

R2.4. A major observation of the work is the displacement of the targeted B cells in GC by endogenous B cells over time. This is investigated sequencing both to assess SHM and

subsequently affinity of expressed recombinant Abs (line 368) and crystal structure in binding to Ag (line 383). While endogenous B cells increased from undetected at d0 to 20nM by d7 (not d0 as stated line 376) compared to 580nM for targeted B cells. But this is not necessarily an explanation as affinity of soluble Ab need not reflect what happens to deposited Ag on FDC in the GC. The partial overlap from crystal structure is also incompletely investigated – was it a functional interference or not?

R2.4 Response: Thank you for noting our error at line 376! We have corrected it. As to the issue of soluble affinity, we entirely concur with Reviewer 2 that soluble affinity does not fully reflect the *in vivo* B cell receptor interaction; we were, indeed one of the first labs to study the importance of membrane antigen recognition, and we entirely agree that *in vivo*, 2D, or apparent affinity, is the right way. Unfortunately, the pool of light chains in these models renders measuring this infeasible. Therefore, we focused on biological assays, and have now also developed a bead assay to measure functional interference. This demonstrated competition by the WT CD45.1⁺ antibody *in vitro* (**Figure 5E–F**). Further, we also analyzed the *in vivo* effect of WT CD45.1⁺ antibody in blocking 10E8-GT10.2 immune responses by injecting a CD45.1⁺ mAb isolated 14 days after immunization into a naïve mouse after adoptive transfer but prior to immunization by 10E8-GT10.2. We confirmed that passive transfer of Day 14 CD45.1⁺ antibody was sufficient to inhibit the recruitment of 10E8 UCA^H B cells in response to 10E8-GT10.2 (**Figure 5G–I**); germinal center analysis 7 days post immunization showed no recruitment in competitor CD45.1⁺ antibody injected mice, despite GC sizes similar to the control groups (**Figure 5H&I**). These new experiments demonstrated that the partial overlap shown by the structures can provide a mechanism for inhibition. Similar effects have previously been observed in instances of identical or overlapping binding in other preclinical HIV models (Tas et al. *Immunity* 2022).

R2.5. Line 400-401 is a point of restating an inference as a conclusion, saying affinity difference was the driver of loss of targeted B cells. Likely but not certain.

R2.5 Response: The reviewer is entirely correct that this statement is too strong, and we have rewritten the sentence (now 428); however, in an earlier study using transgenic mice, it was demonstrated that competition from high affinity endogenous cells resulted in loss of low affinity B cells from GCs and that failure of sustenance of low affinity B cells is reflective of competition from high affinity BCRs rather than an intrinsic inability of low affinity B cells (Dal Porto et al., 2002); we now cover this study in the **Discussion** (lines 596–598). Furthermore, by changing (1) the antigen affinities and (2) the affinities of the BCRs in the models, we feel we have provided a quite robust empirical foundation for the importance of the affinity gap generated during GC maturation, as the mechanism underlying B cell persistence.

As we proposed with the editor, we have now performed additional direct competition experiments; we have confirmed via a competitive adoptive transfer experiment (**Figure 8**) that when NGS-04, 10E8-UCA, and MPERHuGL18 KI B cells are simultaneously transferred into the same murine host and immunized with clinically relevant 10E8-GT10.2, the higher affinity MPERHuGL18 B cells predominate, the middling 10E8 UCA are present, and the low affinity NGS-04 are excluded almost entirely at late timepoints, and that a greater fraction of MPER HuGL18 cells than 10E8 UCA are in the DZ. We believe that this additional data, and the additional analysis of affinity-enhancing vs. non-affinity-enhancing SHM in MPER HuGL and

10E8 UCA (**Figure 6I** and **Supplementary Figure 9F** for 10E8 UCA; **Supplementary Figure 10G and H** for MPER HuGL18), strengthens the connection between affinity and maintenance, though that connection is more likely a dynamic function of affinity differentials over time, which we now clarify in the revised text.

R2.6. How do the SHM frequency and L chain usage explain persistence of the 10E8 UCAH B cells in GC (lines 426-436)? Correlation not causation.

R.2.6 Response: We analyzed the LC usage and affinities of the 10E8UCA mAbs from day 21 sustained cells following GT10.3 immunization (**Supplementary Figure 9G**). We did not find a relationship between LC and observed affinity: though some pairings demonstrate ~1 log median difference in KD from others, measured variation within pairings is quite high. These data suggest that LC usage and enrichment are unlikely to be a driving force in the persistence of 10E8UCA bearing B cells in the germinal center; we now ameliorate our statements (lines 476–477), and we thank the reviewer for this clarifying question.

R2.7. Indeed, the data in lines 447-450 indicate that affinity alone is not the answer.

R2.8. Again SHM analysis on persisting B cells does not explain their persistence in and of itself (line 487).

R2.7–2.8 Response: The reviewer raises two excellent points here: can we directly connect SHM to affinity, and how does this relate to persistence? We analyzed the relationship between affinity and mutations in the 10E8UCA heavy chain after 10E8-GT10.3 immunization; as demonstrated in new analyses at **Figure 6I** and **Supplementary Figure 9F** for 10E8 UCA; **Supplementary Figure 10G and H** for MPER HuGL18, diversity generated by SHM may sometimes lead to a loss of affinity for the original antigen. This is in line with other articles showing that lower-affinity B cells found at later time points are the clonal descendants of higher affinity B cells which have undergone a detrimental mutation and lost monovalent affinity (as observed in Viant et al. *Cell* 2020). We have incorporated this improved understanding of the interactions between affinity and persistence in the text.

R2.9. While I agree with the observation of starting affinity and persistence, the ‘rule’ that can be derived from this is unclear. This remains to me as an elegant set of observations with relevance to this system.

R2.9 Response: As described above, we have now added significant additional experiments and analysis, and hope they will be more robustly make the mechanistic argument. However, we respectfully disagree that the original experiments, which interrogate antigen competition, could be accurately termed "observational," as they provided a strong mechanistic explanation in the form of the directly observed and, via antigen and BCR alteration, experimentally manipulated affinity gap.

R2.10. And, why do they authors think that the MPER HuGL18h B cells show more substantial gains in affinity for the same number of mutations? Could this be the basis for a more generalizable conclusion?

R2.10 Response: We analyzed heavy chain sequences of 10E8UCA and MPER HuGL18 cells sustained in the GC on 21 dpi in new phylogenetic trees that are labeled with affinity information; this suggests substantially increased affinity in a number of highly derived MPER HuGL18 descendants (**Supplementary Figure 10F**), while the highly derived 10E8 UCA descendants (**Figure 6I**) generally lost affinity. We suspect, but cannot confirm, that this may relate to the initial sequences of the IGHV genes for 10E8 UCA and MPER HuGL18. We agree with the reviewer that the varying quality of mutational gain in MPER HuGL18 and 10E8 UCA is relevant beyond this system: whenever several precursors are activated with specificity for the same antigen, the same number of mutations can lead to different maturation paths. We have refocused the third **Discussion** paragraph to address this issue.

R2.11. The conclusions listed in lines 568-570 are completely true and important but indicate the specialized nature of the conclusions that were reached.

R2.11 Response: We sincerely appreciate the reviewer's acknowledgement of the importance of this work, but we disagree that observations made first in a specific system are inherently uninteresting. Rather, observations found in extremis or under rare circumstance are the grit around which pearls form. However, we are not working in extremis, and our platform is highly relevant: This system and the immunogen development process described here are vital to the field of HIV research, and the tunable affinity gap effects observed provide new insights into GC function.

We would like to express our gratitude to both reviewers, whose recommendations have, we believe, substantially strengthened the quality of this manuscript and refined our thinking on the affinity gap. We hope our intervening efforts have matched the depth of thought and care they brought to their reviews of our work.

Decision Letter, first revision:

3rd Mar 2024

Dear Dr. Batista,

Thank you for submitting your revised manuscript "Elicitation of MPER-targeting antibodies in humanized mouse models: Are germinal center dynamics driven by precursor-competitor affinity gaps?" (NI-A36284A). It has now been seen by the original referees and their comments are below. The reviewers find that the paper has improved in revision, and therefore we'll be happy in principle to publish it in Nature Immunology, pending minor revisions to satisfy the referees' final requests and to comply with our editorial and formatting guidelines.

I will now pre-edit the current version of your paper. We will also perform detailed checks on your paper and will send you a checklist detailing our editorial and formatting requirements in about two weeks. Please do not upload the final materials and make any revisions until you receive this additional information from us.

In the meantime however, please deposit all omic and code data into public repositories so that the accession codes are readily available to be added in the revised manuscript. We cannot accept the paper without the codes. In addition, please check that the ORCID of ALL CORRESPONDING AUTHORS is linked to their Nature account, as this frequently causes delays at acceptance. Should you have any query or comments about ORCID, please do not hesitate to contact our editorial assistant at immunology@us.nature.com.

If you had not uploaded a Word file for the current version of the manuscript, we will need one before beginning the editing process; please email that to immunology@us.nature.com at your earliest convenience.

Thank you again for your interest in Nature Immunology. Please do not hesitate to contact me if you have any questions.

Sincerely,

Ioana Staicu, Ph.D.
Senior Editor
Nature Immunology

Tel: 212-726-9207
Fax: 212-696-9752
www.nature.com/ni

Reviewer #1 (Remarks to the Author):

The authors have performed additional experiments that have significantly strengthened the

conclusions and clarified the message, which is important and timely.

Reviewer #2 (Remarks to the Author):

I read with interest the revised submission from Ray et al. I found their thoughtful arguments in the rebuttal in response to the variety of comments from the referees persuasive, certainly in regard to the novelty of their findings on the basis of B cell persistence within GC. I agree that they make some exciting observations in that area, and I particularly appreciate the 3-way competition experiment in Fig 8. I think the work is of sufficient quality and impact to justify publication in Nat Imm.

My only suggestion is to reword the description of the experiment reported in Figure 5, which uses the terms CD45.1+ mAb and CD45.2+ mAb in what I found to be hard to follow ways. It may be me, but the phrase "a secondary fluorescently labeled CD45.2+ mAb" (line 412) is confusing. To me a secondary Ab refers usually to an antibody to detect the primary Ab. But I think here 'secondary' refers to a competition in binding in that the 'secondary mAb' is the same mAb as the primary, but unlabeled. There is the possibility for confusion in referring to the whole experiment as using CD45.1 Abs and CD45.2 Abs in vitro, without specifying that these are mAbs from CD45.1 or CD45.2 B cells, and to indicate how many were used and how many times this was done. Figure and text currently indicate once for Fig 5A-F, and not of how many the data are representative.

Author Rebuttal, first revision:

Response to Reviewers: NI-A36284A

Reviewer #1:

Remarks to the Author:

The authors have performed additional experiments that have significantly strengthened the conclusions and clarified the message, which is important and timely.

Response: We sincerely appreciate the Reviewer's recognition of our work, as well as their provisions of the initial suggestions and constructive critiques which led to that work. We are delighted that they find the additional experiments to have strengthened the manuscript.

Reviewer #2:

Remarks to the Author:

I read with interest the revised submission from Ray et al. I found their thoughtful arguments in the rebuttal in response to the variety of comments from the referees persuasive, certainly in regard to the novelty of their findings on the basis of B cell persistence within GC. I agree that they make some exciting observations in that area, and I particularly appreciate the 3-way

competition experiment in Fig 8. I think the work is of sufficient quality and impact to justify publication in Nat Imm.

My only suggestion is to reword the description of the experiment reported in Figure 5, which uses the terms CD45.1+ mAb and CD45.2+ mAb in what I found to be hard to follow ways. It may be me, but the phrase "a secondary fluorescently labeled CD45.2+ mAb" (line 412) is confusing. To me a secondary Ab refers usually to an antibody to detect the primary Ab. But I think here 'secondary' refers to a competition in binding in that the 'secondary mAb' is the same mAb as the primary, but unlabeled. There is the possibility for confusion in referring to the whole experiment as using CD45.1 Abs and CD45.2 Abs in vitro, without specifying that these are mAbs from CD45.1 or CD45.2 B cells, and to indicate how many were used and how many times this was done. Figure and text currently indicate once for Fig 5A-F, and not of how many the data are representative.

Response: We thank the reviewer for their excellent commentary on the first draft, and we agree that, in responding to their concerns in our last draft, our manuscript has been reshaped for the better. On re-reading our description of Figure 5, we found that the reviewer was entirely correct regarding the obscurity of our original description of the antibody blocking experiments. We have rewritten the results over all for clarity and conciseness, but we have paid special attention to this section and its legend, which we now hope more precisely and fully conveys the experimental details. We thank the reviewer for bringing this to our attention.

Final Decision Letter:

Dear Dr. Batista,

I am delighted to accept your manuscript entitled "Affinity gaps among B cells in germinal centers drive the selection of MPER precursors" for publication in an upcoming issue of Nature Immunology.

Over the next few weeks, your paper will be copyedited to ensure that it conforms to Nature Immunology style. Once your paper is typeset, you will receive an email with a link to choose the appropriate publishing options for your paper and our Author Services team will be in touch regarding any additional information that may be required.

Please note that *Nature Immunology* is a Transformative Journal (TJ). Authors may publish their research with us through the traditional subscription access route or make their paper immediately open access through payment of an article-processing charge (APC). Authors will not be required to make a final decision about access to their article until it has been accepted. Find out more about Transformative Journals.

Your paper will be published online soon after we receive your corrections and will appear in print in the next available issue.

Also, if you have any spectacular or outstanding figures or graphics associated with your manuscript - though not necessarily included with your submission - we'd be delighted to consider them as candidates for our cover. Simply send an electronic version (accompanied by a hard copy) to us with a possible cover caption enclosed.

To assist our authors in disseminating their research to the broader community, our SharedIt initiative provides you with a unique shareable link that will allow anyone (with or without a subscription) to read the published article. Recipients of the link with a subscription will also be able to download and

print the PDF.

If you have not already done so, we strongly recommend that you upload the step-by-step protocols used in this manuscript to the Protocol Exchange. Protocol Exchange is an open online resource that allows researchers to share their detailed experimental know-how. All uploaded protocols are made freely available, assigned DOIs for ease of citation and fully searchable through nature.com. Protocols can be linked to any publications in which they are used and will be linked to from your article. You can also establish a dedicated page to collect all your lab Protocols. By uploading your Protocols to Protocol Exchange, you are enabling researchers to more readily reproduce or adapt the methodology you use, as well as increasing the visibility of your protocols and papers. Upload your Protocols at www.nature.com/protocolexchange/. Further information can be found at www.nature.com/protocolexchange/about .

Please note that we encourage the authors to self-archive their manuscript (the accepted version before copy editing) in their institutional repository, and in their funders' archives, six months after publication. Nature Portfolio recognizes the efforts of funding bodies to increase access of the research they fund, and strongly encourages authors to participate in such efforts. For information about our editorial policy, including license agreement and author copyright, please visit www.nature.com/ni/about/ed_policies/index.html

Sincerely,

Ioana Staicu, Ph.D.
Senior Editor
Nature Immunology

Tel: 212-726-9207
Fax: 212-696-9752
www.nature.com/ni